# Toward Understanding Adversarial Distillation: Why Robust Teachers Fail

Hongsin Lee [1]    Hye Won Chung [1]

## Abstract

Adversarial Distillation aims to enhance student robustness by guiding the student with a robust teacher's soft labels within the min-max adversarial training framework, yet its success is notoriously inconsistent: a more robust teacher often fails to improve, or even harms, the student's robust generalization. In this paper, we identify a key mechanism of this teacher dependency: the misalignment between the teacher's supervisory confidence and the student's representational limitations on a consistent subset of training data—the Robustly Unlearnable Set. We present a theoretical framework analyzing the feature learning dynamics of a two-layer neural network, demonstrating that this mismatch creates a dichotomy in distillation outcomes. We prove that when a teacher provides confident supervision on unlearnable samples, it compels the student to memorize spurious noise patterns that eventually overpower the learned robust signal, thereby driving robust overfitting. Conversely, a teacher that exhibits high uncertainty on these samples effectively suppresses noise memorization, allowing the student to rely solely on the learnable signal for robust generalization. We empirically validate our theory across both synthetic simulations and real-image classification datasets, confirming that robust overfitting is driven by the teacher's interaction with unlearnable samples. Finally, we demonstrate that a teacher's predictive entropy on unlearnable samples serves as a strong indicator of student robustness, validating our theoretical framework and offering a principled guideline for robust teacher selection.

## 1. Introduction

Deep neural networks have achieved strong performance across various domains, yet they remain vulnerable to small adversarial perturbations that can reliably induce misclassification (Goodfellow et al., 2015; Madry et al., 2018; Carlini & Wagner, 2017), raising concerns for safety-critical deployment (Ma et al., 2021; Grigorescu et al., 2020). Adversarial training (AT) (Madry et al., 2018; Zhang et al., 2019) is currently the most effective empirical defense, utilizing a min-max optimization to minimize the maximum loss under adversarial attacks. However, AT often suffers from robust overfitting: robust test accuracy peaks at an intermediate epoch and then declines, even while robust training accuracy continues to improve (Rice et al., 2020; Yu et al., 2022).

A promising evolution of this paradigm is Adversarial Distillation (AD), which integrates knowledge distillation into the min-max adversarial training framework (Goldblum et al., 2020; Zhu et al., 2022; Zi et al., 2021; Huang et al., 2023a; Lee et al., 2025). Unlike standard AT that fits one-hot hard labels, AD trains a student model to match the softened probability distributions of a robust teacher on adversarially perturbed inputs. This richer supervisory signal is intended to improve generalization, and curb robust overfitting. Indeed, prior works indicate that distilling from an early-stage or well-generalized teacher can serve as an effective heuristic to mitigate performance degradation (Chen et al., 2020b).

However, the success of AD is notoriously unpredictable and highly dependent on the choice of teacher. A more robust teacher does not necessarily yield a more robust student; in some cases, distillation can even exacerbate overfitting and degrade student performance (Lee & Chung, 2026). This strong teacher dependence exposes a critical gap in our understanding. Although recent studies have identified empirical indicators of failure, such as the absence of transferable adversarial samples under which the teacher's guidance remains effective (Lee & Chung, 2026), these are symptoms rather than causes. They describe what happens when AD fails, but not the underlying reasons.

This paper aims to bridge the gap between empirical correlation in teacher-dependent student robustness and mechanistic understanding. We argue that the key lies in a subset of training data that is intrinsically hard to learn under adversarial constraints–hereafter referred to as the robustly

---

[1]School of Electrical Engineering, KAIST, Daejeon, Korea. Correspondence to: Hye Won Chung <hwchung@kaist.ac.kr>.

*Proceedings of the $43^{rd}$ International Conference on Machine Learning*, Seoul, South Korea. PMLR 306, 2026. Copyright 2026 by the author(s).

*unlearnable set.* Our investigation begins with a crucial empirical observation: a specific, consistent set of training samples is misclassified by non-overfitted adversarially trained models, regardless of the training method. We posit that these samples are the primary drivers of robust overfitting and, consequently, of failures in AD.

In this paper, we develop a rigorous theoretical framework that resolves the paradox in which distillation from a highly robust teacher exacerbates student overfitting. Our analysis identifies the unlearnable set as the pivotal factor: we show that confident supervision by the teacher on this set forces the student to memorize spurious noise patterns, which makes the student vulnerable to adversarial perturbations and breaks robust generalization. This framework not only clarifies the underlying mechanism but also precisely characterizes the teacher properties required to suppress, rather than exacerbate, such overfitting.

We empirically validate our theoretical framework through extensive experiments on both controlled synthetic settings and real-world image classification benchmarks. The observed learning dynamics closely match the predictions of our model and theoretical analysis. Moreover, our study yields a key practical principle for teacher selection in AD: a teacher's effectiveness is strongly tied to its predictive confidence on the unlearnable samples in the training set. This provides a simple, data-driven criterion for identifying robust teachers that can successfully guide student models, thereby bridging theory and practice. Our contributions are as follows:

- First, we empirically show that a consistent subset of the training data forms a robustly *unlearnable set* and serves as the primary driver of robust overfitting across different training paradigms (Section 3).

- Second, we develop a theoretical framework and characterize the learning dynamics of AD. Our analysis provides a first-principles explanation of how unlearnable samples induce robust generalization failure and resolves the teacher-dependency puzzle (Section 4).

- Finally, we validate our theory through extensive experiments and propose a practical criterion for selecting an effective robust teacher a priori, based on its predictive entropy on unlearnable samples (Section 5).

## 2. Related Work

We provide a detailed review of related work in Section C.

**Adversarial Training and Its Limitations.** Adversarial training (AT) is the most effective empirical defense and is commonly formulated as a min-max optimization (Madry et al., 2018). Although variants such as TRADES (Zhang

et al., 2019) improve the robustness-accuracy trade-off, AT still faces significant challenges: robust overfitting can degrade performance in later training stages (Rice et al., 2020), and robustness is highly sensitive to model capacity (Gowal et al., 2020; Rebuffi et al., 2021). These limitations motivate the use of adversarial distillation (AD) to transfer robustness from larger models to smaller, resource-constrained ones.

**Adversarial Distillation and Robust Saturation.** AD aligns a student's predictions with those of a robust teacher on adversarially perturbed inputs. Early methods such as ARD (Goldblum et al., 2020) demonstrated its potential, and later works refined the objective by leveraging teacher logits or confidence information (Zhu et al., 2022; Zi et al., 2021; Huang et al., 2023a; Lee et al., 2025). However, recent studies have revealed a paradox: a stronger teacher does not necessarily produce a stronger student. Zi et al. (2021) reported a robustness saturation effect, while Lee & Chung (2026) showed that overly strong teachers can even degrade student performance by inducing overfitting. In particular, Lee & Chung (2026) identified the scarcity of transferable adversarial samples (TAS) as a key factor leading to high adversarial variance and, consequently, robust overfitting. While TAS provides an important empirical signal, the underlying theoretical mechanism driving this failure remains an open question.

**Theoretical Analysis via Feature Learning.** Theoretical studies of robustness have largely focused on simplified linear models or the lazy training regime (Gao et al., 2019). However, Wang et al. (2022) showed that the lazy regime is insufficient to explain robustness, motivating a shift toward the feature learning regime (Allen-Zhu & Li, 2022). While this framework has recently been applied to standard AT (Li & Li, 2025a;b), the learning dynamics of AD remain unexplored. We fill this gap by extending feature learning theory to AD, rigorously showing how robust teacher supervision can, counterintuitively, distort student feature learning.

## 3. Empirical Motivation

In this section, we first highlight the central empirical puzzle motivating our work: the success of AD depends critically on the choice of robust teacher, which can either suppress or exacerbate robust overfitting. We then introduce our core hypothesis that this behavior is driven by a consistent subset of training samples that are intrinsically difficult to learn robustly, which we term the *unlearnable set*.

### 3.1. Teacher-Dependent Robust Overfitting

We first establish the baseline phenomenon of robust overfitting in standard AT. As shown in Figure 1a, robust test accuracy peaks at an intermediate epoch and then declines, even as robust training accuracy continues to increase, indi-

*Table 1.* **Identification of robustly unlearnable samples.** We categorize training samples based on the rigorous intersection of predictions from 60 models (spanning 6 training paradigms × 10 random seeds, including AD from 4 teachers detailed in Table 3), evaluated at their respective peak robust accuracy epochs (see Section A.3 and Algorithm 1). The 'Intersection' column reports the size of the consistent subsets: samples correctly classified by all models (Learnable) or misclassified by all models (Unlearnable).

| Model | Category | Adversarial Training | | Adversarial Distillation | | | | Intersection |
|---|---|---|---|---|---|---|---|---|
| | | PGD-AT | TRADES | Chen | Rebuffi | Bartoldson | Gowal | |
| MN-V2 | Unlearnable | 13,898 | 12,261 | 11,926 | 13,575 | 12,948 | 13,357 | 8,979 |
| | Learnable | 22,518 | 24,112 | 30,164 | 25,770 | 23,615 | 22,651 | 19,385 |
| ResNet-18 | Unlearnable | 8,360 | 10,217 | 10,464 | 8,627 | 7,511 | 8,047 | 5,217 |
| | Learnable | 25,927 | 26,903 | 34,007 | 32,714 | 27,598 | 26,323 | 21,899 |
| WRN-28-10 | Unlearnable | 2,816 | 5,084 | 9,744 | 6,958 | 3,270 | 2,713 | 1,697 |
| | Learnable | 24,003 | 31,519 | 36,405 | 38,401 | 28,533 | 25,706 | 19,610 |
| WRN-34-10 | Unlearnable | 2,608 | 4,511 | 9,682 | 6,861 | 3,112 | 2,579 | 1,559 |
| | Learnable | 21,277 | 32,727 | 36,567 | 38,508 | 20,188 | 25,932 | 16,397 |

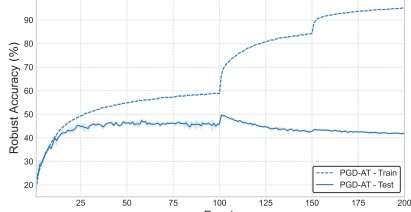

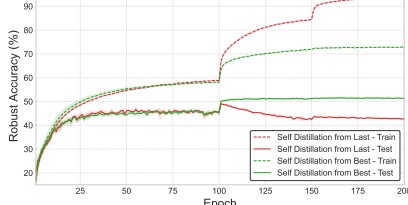

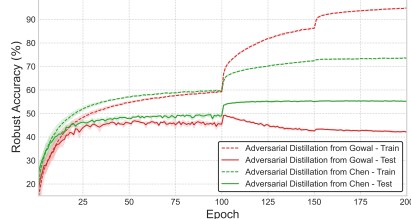

*(a)* **Standard PGD-AT:** Robust test accuracy degrades after an early peak, indicating robust overfitting.

*(b)* **Self-Distillation:** Distilling from a 'Best' teacher mitigates overfitting, whereas a 'Last' teacher amplifies it.

*(c)* **External Teachers:** Student robustness varies significantly depending on the teacher, even among strong robust models.

*Figure 1.* **Teacher-dependent dynamics of robust overfitting.** (a) Standard adversarial training suffers from robust overfitting. (b) In self-distillation, the outcome is strictly determined by the teacher's overfitting status. (c) Similarly, distillation from external teachers (specifically Gowal and Chen, detailed in Table 3) yields inconsistent results, indicating that teacher robustness alone is not a sufficient predictor of student success.

cating a breakdown in robust generalization.

This challenge is further compounded in AD, where the teacher's state plays a critical role. As seen in self-distillation (Figure 1b), a teacher from an early, well-generalized epoch can suppress student overfitting, whereas a teacher from a later, overfitted epoch instead exacerbates performance degradation. This puzzle persists even with external, stronger teachers (Figure 1c): high standalone robustness of a teacher model does not guarantee successful distillation and can still induce severe overfitting in the student, while another teacher model leads to robust generalization. These observations raise a fundamental question: *What underlying mechanism governs this teacher-dependent behavior in adversarial distillation?*

### 3.2. A Consistent Set of Unlearnable Samples

We hypothesize that robust overfitting is driven by a specific and consistent subset of training examples that are intrinsically difficult to learn robustly. To test this, we conduct

a systematic analysis across 10 random seeds and multiple robust training paradigms, including AT methods and AD with diverse teachers. As shown in Table 1, the training data consistently partitions into two groups: *learnable samples*, which are reliably classified, and *unlearnable samples*, which models persistently fail to learn. This partition is identified at the epoch of peak robust accuracy, indicating that unlearnability is an intrinsic property of the data-architecture pair rather than an effect of late-stage overfitting.

To probe the structure of this subset, we visualize internal representations of a ResNet-18 using feature inversion, as shown in Figure 2. Reconstructions of learnable samples preserve clear semantic content, whereas those of unlearnable samples exhibit severe representational collapse, degenerating into spurious noise or distorted artifacts. Quantitative results in Table 1 further show a strong dependence on model capacity: the size of the unlearnable set decreases markedly with larger architectures, from nearly 9,000 samples in MobileNet-V2 to about 1,500 in WRN-34-10. This inverse relationship confirms that unlearnability is not due to

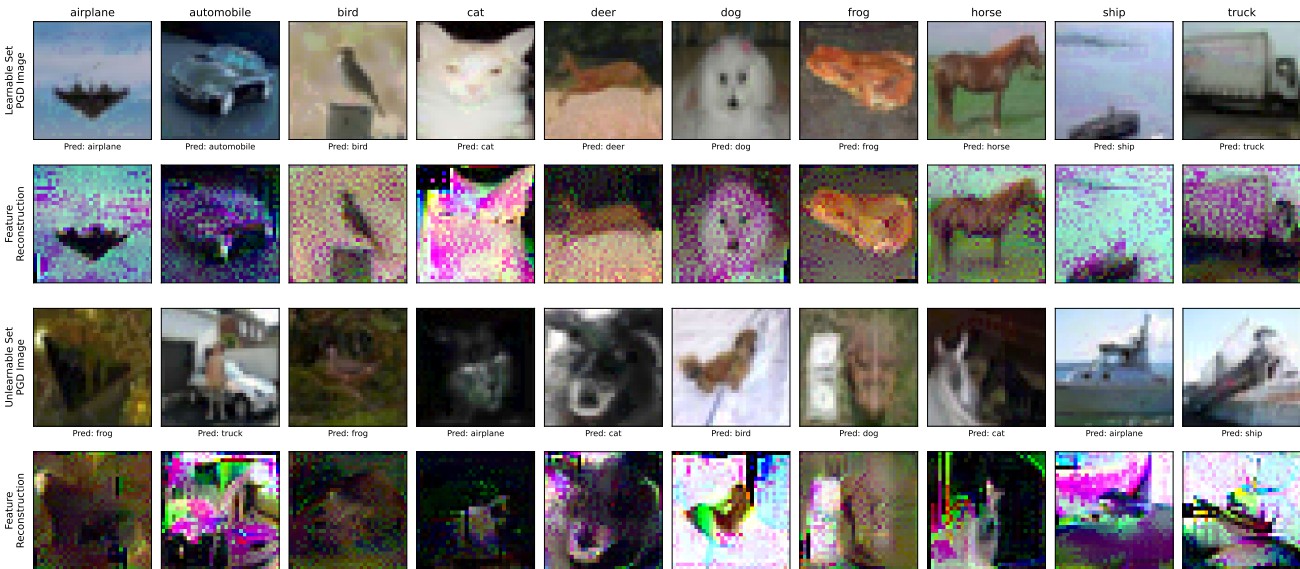

*Figure 2.* **Feature reconstruction analysis.** Using robust feature inversion (details in Section A.4), we visualize the internal representations of the student model. The distinct gap between the clear semantic recovery of learnable samples (top) and the distorted artifacts or spurious features of unlearnable samples (bottom) provides visual evidence of the model's inability to extract ground-truth aligned robust features from the unlearnable set.

inherent data corruption but reflects model capacity-limited representation.

Importantly, this pattern is consistent across training methods: larger models reduce the unlearnable set, but a non-trivial subset remains unlearnable. We observe the same trend on CIFAR-100 under the same evaluation protocol (Table 7). These results suggest that unlearnability is not an absolute property of the data, but is model-dependent: larger models can learn a wider portion of the training set robustly, whereas smaller students cannot, even at their peak robust accuracy.

### 3.3. From Empirical Evidence to a Theoretical Model

These results suggest that robust training can be limited by the model architecture: the same training sample can be robustly learnable for a larger model but unlearnable for a smaller one. A natural explanation is that robust classification depends on robust features whose utilization can vary with the model architecture. This observation aligns with established findings that adversarially robust models rely on robust features that differ from those used by standard models (Ilyas et al., 2019; Tsipras et al., 2019), and that robustness is strongly sensitive to model capacity and architecture (Gowal et al., 2020; Rebuffi et al., 2021).

Our experiments support this view at the sample level. We find a subset of training samples that smaller students cannot learn robustly, but larger models can, even under the same training protocol. This suggests that robustness is not equally easy to obtain for all samples: some samples rely

on robust features that smaller architectures cannot represent well under adversarial training. As we show next, this difference matters for adversarial distillation: the teacher can be confident based on robust features that the student cannot represent, which can drive the student to fit spurious patterns and induce robust overfitting.

## 4. Theoretical Analysis

To explain the teacher-dependent overfitting observed in Section 3, we present a theoretical framework capturing the representational mismatch between a capacity-constrained student and unlearnable robust features. The framework isolates how unlearnable samples can turn residual training gradients into noise memorization, and how teacher uncertainty can suppress this effect.

### 4.1. Problem Setup

We model the distillation dynamics using orthogonal feature subspaces and a capacity-constrained student; see Table 11 for notations and Appendix D for the complete formal setup. We use asymptotic notation, where $\tilde{O}(\cdot)$, $\tilde{\Omega}(\cdot)$, and $\tilde{\Theta}(\cdot)$ hide polylogarithmic factors.

**Definition 4.1** (Data Generating Process)**.** We consider a binary classification task with labels $y \in \{+1, -1\}$. Let the input $\mathbf{X} \in \mathbb{R}^{P \times d}$ be a concatenation of $P$ patches. We define two orthogonal robust features: a *learnable* feature $\mathbf{u} := \mathbf{e}_1$ and an *unlearnable* feature $\mathbf{v} := \mathbf{e}_d$. Let $\mathcal{F} := \text{span}\{\mathbf{u}, \mathbf{v}\}$ be the feature subspace. Fix a partition of the training indices $[N]$ into a learnable set $\mathcal{S}_L$ and an

unlearnable set $\mathcal{S}_U$ with ratio $p_{\mathrm{un}} := |\mathcal{S}_U|/N$. For each training index $i \in [N]$, the label $y_i$ and a signal patch index $s(\mathbf{X}_i) \in [P]$ are drawn uniformly. The signal patch is generated as:

$$\mathbf{x}_{i,s(\mathbf{X}_i)} = \begin{cases} \alpha y_i \mathbf{u}, & \text{if } i \in \mathcal{S}_L \quad \text{(learnable)}, \\ \alpha y_i \mathbf{v}, & \text{if } i \in \mathcal{S}_U \quad \text{(unlearnable)}. \end{cases} \quad (1)$$

Here, $\alpha > 0$ denotes the signal strength of the robust feature. The remaining patches ($p \neq s(\mathbf{X}_i)$) contain random noise orthogonal to the feature subspace: $\mathbf{x}_{i,p} \sim \mathcal{N}(\mathbf{0}, \sigma_n^2(\mathbf{I}_d - \Pi_\mathcal{F}))$, where $\Pi_\mathcal{F}$ is the projection matrix onto the feature subspace. We analyze two regimes: the learnable-only regime $p_{\mathrm{un}} = 0$ and the sparse-unlearnable regime $CN^{-1} \leq p_{\mathrm{un}} \leq C^{-1}N^{-1}\log d$.

For test-time evaluation, we consider the learnable-sample distribution $\mathcal{D}_L$, defined by the same generation rule with the learnable signal patch $\alpha y \mathbf{u}$ and independent Gaussian noise patches.

**Definition 4.2** (Student Model). The student model $f_\mathbf{W}$ is a two-layer neural network with $m$ filters and cubic activation function $\phi(z) = (\max\{0, z\})^3$. The output aggregates patch responses via

$$f_\mathbf{W}(\mathbf{X}) = \sum_{r=1}^m \sum_{p=1}^P [\phi(\langle \mathbf{w}_r, \mathbf{x}_p \rangle) - \phi(-\langle \mathbf{w}_r, \mathbf{x}_p \rangle)]. \quad (2)$$

To formally capture the representational mismatch, we enforce the weights to be orthogonal to the unlearnable feature $\mathbf{v}$ throughout training, i.e., $\langle \mathbf{w}_r, \mathbf{v} \rangle = 0$ for all $r$. Accordingly, we initialize $\mathbf{w}_r \sim \mathcal{N}(\mathbf{0}, \sigma_0^2\mathbf{I}_d)$ with the component parallel to $\mathbf{v}$ zeroed out, rendering the student structurally blind to the robust signal in $\mathcal{S}_U$.

**Definition 4.3** (Teacher Configurations). We analyze two robust teachers: a Good Teacher $f_{\mathbf{W}_G}$ and a Bad Teacher $f_{\mathbf{W}_B}$. When the distinction is irrelevant, we write $f_{\mathbf{W}_T}$ to denote either teacher. Both teachers generalize well on learnable samples and do not rely on sample-specific noise patches. In particular, for any $i \in \mathcal{S}_L$, either teacher satisfies the target-aligned margin condition

$$y_i f_{\mathbf{W}_T}(\mathbf{X}_i) \geq \Gamma, \quad (3)$$

where $\Gamma > 0$ is a sufficiently large teacher margin. They differ only in their behavior on the unlearnable feature $\mathbf{v}$:

1. **Good Teacher** ($f_{\mathbf{W}_G}$). The Good Teacher is orthogonal to $\mathbf{v}$, and therefore remains uncertain on unlearnable samples:

$$y_i f_{\mathbf{W}_G}(\mathbf{X}_i) = 0 \qquad (i \in \mathcal{S}_U). \quad (4)$$

2. **Bad Teacher** ($f_{\mathbf{W}_B}$). The Bad Teacher leverages $\mathbf{v}$, and therefore provides high-confidence supervision on unlearnable samples:

$$y_i f_{\mathbf{W}_B}(\mathbf{X}_i) \geq \Gamma \qquad (i \in \mathcal{S}_U). \quad (5)$$

Importantly, both $f_{\mathbf{W}_G}$ and $f_{\mathbf{W}_B}$ are robust models. The Bad Teacher is not "bad" in isolation; rather, it has access to the additional robust feature $\mathbf{v}$, which the capacity-constrained student cannot represent. The designation "Bad" refers only to its incompatibility as a teacher for the student, not to any deficiency in its standalone robustness or performance.

**Definition 4.4** (Training Adversarial Data Generation). We generate adversarial examples $\tilde{\mathbf{X}}$ by maximizing the logistic loss $\ell(z) := \log(1 + e^{-z})$ under perturbations restricted to the signal-patch direction:

$$\tilde{\mathbf{X}} = \operatorname*{argmax}_{\substack{\|\mathbf{X}'-\mathbf{X}\|_\infty \leq \epsilon \\ (\mathbf{X}'-\mathbf{X}) \in \mathrm{span}(\mathbf{x}_{s(\mathbf{X})})}} \ell\left(y f_\mathbf{W}(\mathbf{X}')\right). \quad (6)$$

Following Li & Li (2025b), we restrict training adversarial perturbations to the signal patch, excluding perturbations in the noise subspace. Observing AD failure even in this idealized setting shows that the failure is intrinsic to the teacher's supervision on unlearnable features that the student cannot represent.

**Definition 4.5** (Objectives and Optimization). We train the student either via adversarial training (AT) or adversarial distillation (AD). The AT objective is

$$\mathcal{L}_{\mathrm{AT}}(\mathbf{W}; \mathbf{X}, y) = \ell\left(y f_\mathbf{W}(\tilde{\mathbf{X}})\right). \quad (7)$$

The AD objective is the soft-target cross-entropy induced by the teacher:

$$\begin{aligned} \mathcal{L}_{\mathrm{AD}}(\mathbf{W}; \mathbf{X}, y) = {} & \sigma(y f_{\mathbf{W}_T}(\mathbf{X})) \ell\left(y f_\mathbf{W}(\tilde{\mathbf{X}})\right) \\ & + \sigma(-y f_{\mathbf{W}_T}(\mathbf{X})) \ell\left(-y f_\mathbf{W}(\tilde{\mathbf{X}})\right), \end{aligned} \quad (8)$$

where $\sigma(z) := (1 + e^{-z})^{-1}$ is the logistic sigmoid.

In both cases, we optimize the empirical objective by gradient descent. At each iteration $t$, adversarial examples $\{\tilde{\mathbf{X}}_i^{(t)}\}_{i=1}^N$ are generated using the current model $\mathbf{W}^{(t)}$, and the parameters are updated by

$$\mathbf{W}^{(t+1)} = \mathbf{W}^{(t)} - \frac{\eta}{N} \sum_{i \in [N]} \nabla_\mathbf{W} \mathcal{L}\left(\mathbf{W}^{(t)}; \mathbf{X}_i, y_i\right). \quad (9)$$

Here, $\mathcal{L} \in \{\mathcal{L}_{\mathrm{AT}}, \mathcal{L}_{\mathrm{AD}}\}$, $\eta > 0$ is the learning rate, and the student is trained for $T$ iterations.

*Remark* 4.6. We impose sufficient parameter and regime requirements in Condition D.5; the following is a high-level summary. The training horizon $T$ is long enough to cover both signal learning and possible noise memorization, with $T \geq \tilde{\Omega}(N(\eta\sigma_0\sigma_n^3 d^{3/2})^{-1})$. The dimension $d$ is sufficiently large, roughly $d \geq \tilde{\Omega}(N^2)$, so that the high-probability bounds and noise-interaction estimates used in the analysis hold uniformly. We also work in a signal-dominant

adversarial regime: the robust signal is stronger than the random noise scale, with $\alpha \geq \tilde{\Omega}(\sigma_n \sqrt{d}/N^{1/3})$, and the perturbation budget is nontrivial but does not erase the signal, summarized as $\sigma_n < \epsilon < \alpha$. Finally, the teacher margin is sufficiently large, $\Gamma \geq \tilde{\Omega}(d)$, so that confident teacher predictions lie in the saturated regime and induce nearly hard-label gradients.

## 4.2. Main Theoretical Results

We characterize when adversarial training and adversarial distillation exhibit robust overfitting. We define the robust test error on $\mathcal{D}_L$ as

$$\mathcal{L}_{\mathcal{D}_L}^{\text{rob}}(f_{\mathbf{W}}) := \Pr_{(\mathbf{X},y)\sim\mathcal{D}_L}\left[\min_{\|\boldsymbol{\delta}\|_\infty \leq \epsilon} y f_{\mathbf{W}}(\mathbf{X} + \boldsymbol{\delta}) < 0\right].$$
(10)

The core mechanism is that learning the robust feature $\mathbf{u}$ is necessary but not sufficient for small robust test error $\mathcal{L}_{\mathcal{D}_L}^{\text{rob}}$. Once this feature is learned, robust generalization is determined by how the training dynamics handle unlearnable samples. If their gradients remain controlled, the learned signal dominates and robust generalization succeeds; if the model instead fits them through sample-specific noise, robust overfitting occurs.

All results in the theoretical analysis are stated under the requirements in Condition D.5 and on the event $\mathcal{E}$ from Lemma E.1. For a sufficiently small failure probability $\delta \in (0, 1)$, this event occurs with probability at least $1 - \delta$; accordingly, we refer to statements on this event as holding "with high probability." Formal statements and proofs appear in Appendix E–H.

**Dichotomy of Standard AT.** We first demonstrate that robust overfitting in AT is not intrinsic to adversarial training itself, but is triggered by the presence of unlearnable samples.

**Theorem 4.7** (Dichotomy of AT). *Suppose a student is trained via AT. With high probability, the student successfully learns the robust feature* $\mathbf{u}$*, i.e., there exists a filter* $r \in [m]$ *such that* $w_{r,1}^{(T)} \geq \tilde{\Omega}(\alpha^{-1})$*. The robust generalization then diverges based on the data composition:*

1. ***Learnable-only:*** *All noise responses remain suppressed at their initialization scale:*

$$\max_{i\in[N], j\neq s(\mathbf{X}_i), r\in[m]}\left|\langle \mathbf{w}_r^{(T)}, \mathbf{x}_{i,j}\rangle\right| \leq \tilde{O}(\sigma_0\sigma_n\sqrt{d}).$$
(11)

*Consequently, the robust test error converges to zero, i.e.,* $\mathcal{L}_{\mathcal{D}_L}^{\text{rob}} \leq o(1)$*.*

2. ***Sparse-unlearnable:*** *The noise responses are amplified to memorize spurious noise on at least one unlearnable*

*sample: there exists* $i \in \mathcal{S}_U$ *such that*

$$\max_{r\in[m], j\neq s(\mathbf{X}_i)} y_i\langle \mathbf{w}_r^{(T)}, \mathbf{x}_{i,j}\rangle \geq \tilde{\Omega}(1).$$
(12)

*Consequently, the robust test error remains high, i.e.,* $\mathcal{L}_{\mathcal{D}_L}^{\text{rob}} \geq \frac{1}{2} - o(1)$*.*

Thus, unlearnable samples change the role of AT: instead of relying only on the learned signal, the model is driven to fit sample-specific noise.

**Teacher-Dependent Distillation Dynamics.** We next resolve the distillation puzzle by contrasting Good and Bad teachers. While the robust feature $\mathbf{u}$ is learned in both cases, the teacher's supervision on unlearnable samples determines what happens afterward: residual gradients are either suppressed by a Good Teacher or converted into memorized noise by a Bad Teacher. The Good Teacher guarantee is stated over the horizon $T \leq T_{\max}$ defined in (374), which already exceeds the noise-memorization timescale of AT and AD under a Bad Teacher.

**Theorem 4.8** (Dichotomy of Adversarial Distillation). *Suppose a student is trained via AD. With high probability, the student successfully learns the robust feature* $\mathbf{u}$*, i.e., there exists a filter* $r \in [m]$ *such that* $w_{r,1}^{(T)} \geq \tilde{\Omega}(\alpha^{-1})$*. The robust generalization then diverges based on the teacher type:*

1. ***Good Teacher*** $(f_{\mathbf{W}_G})$***:*** *For* $T \leq T_{\max}$*, high uncertainty on* $\mathcal{S}_U$ *suppresses the noise gradients, maintaining all noise responses at their initialization scale:*

$$\max_{i\in[N], j\neq s(\mathbf{X}_i), r\in[m]}\left|\langle \mathbf{w}_r^{(T)}, \mathbf{x}_{i,j}\rangle\right| \leq \tilde{O}(\sigma_0\sigma_n\sqrt{d}).$$
(13)

*Consequently, the robust test error converges to zero, i.e.,* $\mathcal{L}_{\mathcal{D}_L}^{\text{rob}} \leq o(1)$*.*

2. ***Bad Teacher*** $(f_{\mathbf{W}_B})$***:*** *Confident guidance on* $\mathbf{v}$ *generates persistent gradients, compelling the network to memorize spurious noise on at least one unlearnable sample: there exists* $i \in \mathcal{S}_U$ *such that*

$$\max_{r\in[m], j\neq s(\mathbf{X}_i)} y_i\langle \mathbf{w}_r^{(T)}, \mathbf{x}_{i,j}\rangle \geq \tilde{\Omega}(1).$$
(14)

*Consequently, the robust test error remains high, i.e.,* $\mathcal{L}_{\mathcal{D}_L}^{\text{rob}} \geq \frac{1}{2} - o(1)$*.*

In summary, effective distillation requires the teacher to be uncertain where the student is representationally limited. Otherwise, the student is compelled to match the teacher's confidence by memorizing sample-specific noise. Thus, although the robust feature $\mathbf{u}$ is successfully learned, confident supervision on such samples can still induce robust overfitting.

## 4.3. Proof Sketch: Learning Dynamics

We outline the key proof steps behind the two dichotomy theorems, deferring full details to Appendix F–H; see Appendix E.2 for the overall proof organization.

**Alignment with Learnable Feature.** Across all training regimes, the gradient component along the learnable feature $\mathbf{u}$ dominates the early optimization phase. We first establish that the student consistently learns this feature regardless of the method used.

**Lemma 4.9** (Signal Weight Growth). With high probability, for all $t \geq T_0 = \tilde{\Theta}((\eta\alpha^3\sigma_0)^{-1})$, at least one signal weight is amplified:

$$\max_{r\in[m]} w_{r,1}^{(t)} \geq \tilde{\Omega}(\alpha^{-1}). \tag{15}$$

As a direct consequence of this signal alignment, the margins of learnable samples grow and their loss gradients decay. In particular, after $T_0$, the cumulative gradient contribution from each learnable sample is bounded, so the learnable set no longer drives substantial updates. Thus, the learnable feature is captured early, and the remaining dynamics are governed by whether unlearnable samples induce persistent noise updates.

**AT without Unlearnable Samples.** We first consider the learnable-only regime. Since there are no unlearnable samples, the decay of learnable-sample gradients leaves no residual source of updates after $\mathbf{u}$ is learned. As a result, the noise components receive no substantial further updates and remain at their initialization scale.

**Lemma 4.10** (Noise Stability without Unlearnable Samples). Suppose the dataset contains only learnable samples. With high probability, for all iterations $t \leq T$, all noise responses remain bounded:

$$\max_{i\in[N],\, j\neq s(\mathbf{X}_i),\, r\in[m]} \left| \langle \mathbf{w}_r^{(t)}, \mathbf{x}_{i,j} \rangle \right| \leq \tilde{O}(\sigma_0\sigma_n\sqrt{d}). \tag{16}$$

Together with the signal alignment from Lemma 4.9, this uniform noise stability proves the learnable-only case of Theorem 4.7.

**AT with Unlearnable Samples.** We next consider the sparse-unlearnable regime. Unlike the learnable-only regime, even after the learnable feature $\mathbf{u}$ is learned, unlearnable samples still provide a residual source of updates because the student cannot represent $\mathbf{v}$. These residual gradients drive the model to fit the remaining error through noise components.

**Lemma 4.11** (Noise Memorization with Unlearnable Samples). In the presence of unlearnable samples, with high probability at iteration $T_1 = \tilde{\Theta}(N(\eta\sigma_0\sigma_n^3 d^{3/2})^{-1})$, the model memorizes spurious noise on at least one unlearnable sample:

$$\max_{i\in\mathcal{S}_U,\, j\neq s(\mathbf{X}_i),\, r} y_i \langle \mathbf{w}_r^{(T_1)}, \mathbf{x}_{i,j} \rangle \geq \tilde{\Omega}(1). \tag{17}$$

This memorized direction is used to construct an admissible adversarial perturbation that can overpower the learned signal on fresh test samples. Consequently, the classifier no longer generalizes robustly through $\mathbf{u}$, yielding $\mathcal{L}_{\mathcal{D}_L}^{\mathrm{rob}} \geq \frac{1}{2} - o(1)$, which establishes the second part of Theorem 4.7.

**Dynamics of Adversarial Distillation.** In Adversarial Distillation, the learnable-sample dynamics remain essentially the same as above: by the teacher margin condition, both teachers are confident on $\mathcal{S}_L$, so the AD gradient on learnable samples agrees with the AT gradient up to negligible terms. Hence, the student still learns the robust feature $\mathbf{u}$. The teacher-dependent behavior appears on unlearnable samples. For a Good Teacher, high predictive uncertainty on $\mathcal{S}_U$ prevents the residual updates from producing persistent one-sided noise growth.

**Lemma 4.12** (Noise Suppression with Good Teacher). Under a Good Teacher, the residual gradients on unlearnable samples do not create persistent one-sided noise growth. Hence, over the full training horizon $T \leq T_{\max} = \tilde{\Theta}\left(N(\eta\,mP\,\sigma_0^4\sigma_n^6 d^3)^{-1}\right)$, the noise responses remain at their initialization scale:

$$\max_{i\in[N],\, j\neq s(\mathbf{X}_i),\, r\in[m]} \left| \langle \mathbf{w}_r^{(T)}, \mathbf{x}_{i,j} \rangle \right| \leq \tilde{O}(\sigma_0\sigma_n\sqrt{d}). \tag{18}$$

This bound shows that the student can rely on the learnable feature $\mathbf{u}$ without entering the noise-memorization regime. Importantly, the horizon $T_{\max}$ is not merely an early stopping window: under the parameter regime in Condition D.5, it exceeds the noise-memorization time $T_1$ of standard AT and AD under a Bad Teacher. Thus, over a time scale long enough for noise memorization to occur in the bad cases, the Good Teacher still keeps the noise responses controlled. This establishes the first part of Theorem 4.8.

Conversely, a Bad Teacher provides high-confidence predictions on unlearnable samples, effectively mimicking the hard-label gradients of standard AT.

**Lemma 4.13** (Asymptotic Equivalence to Standard AT). Under a Bad Teacher, the AD gradient on unlearnable samples agrees with the corresponding AT gradient up to an exponentially small term:

$$\left| \frac{\partial\mathcal{L}_{\mathrm{AD}}}{\partial f_{\mathbf{W}}} - \frac{\partial\mathcal{L}_{\mathrm{AT}}}{\partial f_{\mathbf{W}}} \right| \leq e^{-\Omega(d)}. \tag{19}$$

This equivalence makes AD under a Bad Teacher follow the same unlearnable-sample noise-memorization mechanism as standard AT, establishing the second part of Theorem 4.8.

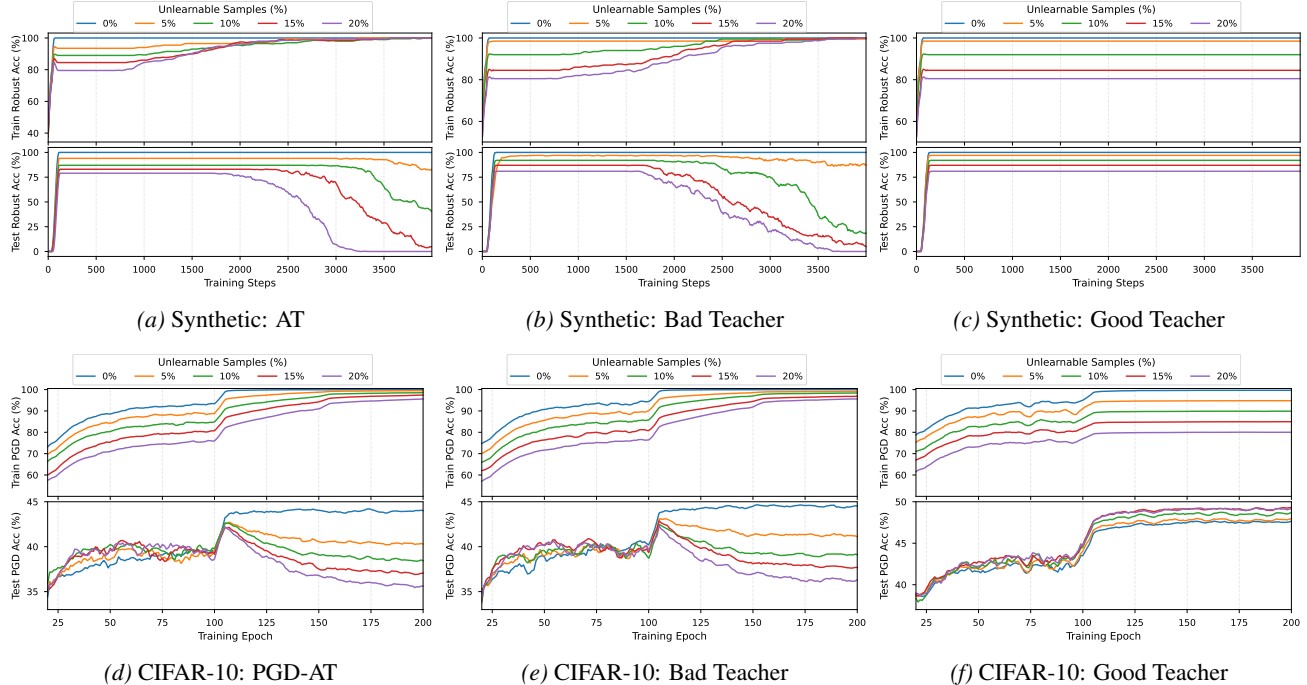

*(a)* Synthetic: AT        *(b)* Synthetic: Bad Teacher        *(c)* Synthetic: Good Teacher

*(d)* CIFAR-10: PGD-AT        *(e)* CIFAR-10: Bad Teacher        *(f)* CIFAR-10: Good Teacher

*Figure 3.* **Consistency between synthetic and real-world dynamics.** We compare the learning dynamics on synthetic data (top row) and CIFAR-10 (bottom row) under varying ratios of unlearnable samples. In both settings, Standard AT and distillation from a Bad Teacher suffer from severe robust overfitting as the unlearnable fraction increases. In contrast, the Good Teacher effectively suppresses noise memorization, maintaining high test robustness.

## 5. Empirical Validation

### 5.1. Simulations on Synthetic Data

To validate our theoretical analysis, we conduct simulations on synthetic data generated by the model in Section 4.1. We train a two-layer student network under the exact conditions of our theory, constraining its weights to be orthogonal to $\mathbf{v}$. This enforces a strict representational mismatch, making the unlearnable feature structurally inaccessible to the student. We compare three settings: standard AT, AD under a Bad Teacher (high confidence on $\mathbf{v}$), and AD under a Good Teacher (high uncertainty on $\mathbf{v}$). Full experimental details are provided in Section A.5.

The results, shown in the top row of Figure 3, support our theoretical framework. As the fraction of unlearnable samples increases, both AT and AD under a Bad Teacher display clear signatures of noise memorization: robust training accuracy saturates at 100%, while robust test accuracy collapses. This confirms that unlearnable samples drive robust overfitting, consistent with the mechanisms predicted by Theorem 4.7 and Theorem 4.8. In contrast, AD under a Good Teacher maintains high robust test accuracy across all unlearnable ratios, in line with Theorem 4.8. This is explained by the teacher's uncertainty on unlearnable samples, which suppresses the corresponding gradient signal and prevents the student from entering the noise-memorization regime.

### 5.2. Validation on Real-World Datasets

We extend our validation to real-world data by constructing controlled subsets of CIFAR-10. Using the consistent learnable ($\mathcal{S}_L$) and unlearnable ($\mathcal{S}_U$) sets identified in Table 1, we form training sets in which a fraction $p_{\mathrm{un}} \in \{0\%, \ldots, 20\%\}$ of learnable samples is replaced by unlearnable ones. We train ResNet-18 models using PGD-AT and AD. Following Lee & Chung (2026), we select representative teacher configurations listed in Table 3, using "Gowal" (Gowal et al., 2021) as the bad teacher and "Chen" (Chen & Lee, 2025) as the good teacher.

As shown in the bottom row of Figure 3, the real-data behavior closely mirrors the synthetic results. Both PGD-AT and the student distilled from the bad teacher exhibit pronounced performance degradation as the unlearnable fraction $p_{\mathrm{un}}$ increases, with a growing gap between robust training and test accuracy, confirming that $\mathcal{S}_U$ is the primary driver of robust overfitting. In contrast, the student distilled from the good teacher effectively suppresses this effect, maintaining stable robust test accuracy even at $p_{\mathrm{un}} = 20\%$. Notably, its robust training accuracy saturates below 100%, indicating that the student learns to ignore unlearnable samples, consistent with the noise-suppression mechanism predicted by our theory.

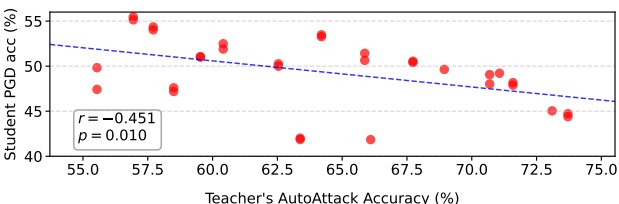

*(a)* Baseline: Teacher Robust Accuracy (AA).

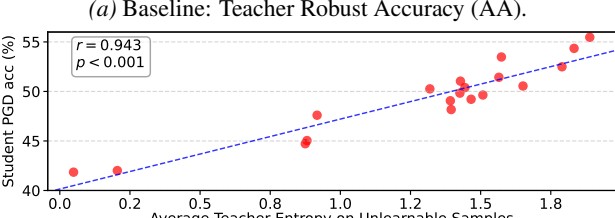

*(b)* **Ours**: Unlearnable Entropy (PGD-AT).

*Figure 4.* **Failure of conventional heuristics and success of our criterion.** (a) Teacher robust accuracy (AutoAttack) correlates negatively with student performance, failing as a selection metric. (b) In contrast, our Unlearnable-Entropy Criterion exhibits a strong positive correlation, effectively identifying high-quality teachers. The reported $r$ and $p$ values denote the Pearson correlation coefficient and its two-sided significance value, respectively. (See Figure 7 for additional baselines.)

### 5.3. Selecting Robust Teachers via Unlearnable-Entropy Criterion

Our theoretical and empirical results show that the effectiveness of AD hinges on the teacher's predictive confidence on unlearnable samples. A teacher that remains uncertain on these samples prevents the student from memorizing spurious noise features. Motivated by this insight, we propose the *Unlearnable-Entropy Criterion*: an effective teacher should exhibit high predictive entropy on the unlearnable set.

While the formal definition of $\mathcal{S}_U$ in Section 3 requires intersecting unlearnable sets across multiple models, directly constructing such a set can be computationally expensive. Instead, exploiting the strong overlap of unlearnable sets across methods observed in Table 1, we approximate $\mathcal{S}_U$ using a single reference model. Concretely, we define a proxy unlearnable set from a standard PGD-AT model at its peak robust accuracy, and evaluate each candidate teacher's average predictive entropy on this subset under PGD-10 attacks.

We apply this criterion to a wide collection of publicly available robust models (listed in Table 4). As shown in Figure 4, the teacher's entropy on the proxy unlearnable set strongly correlates with the student's final robust accuracy after distillation. This demonstrates that a proxy $\mathcal{S}_U$ derived from a single reference model suffices to distinguish good and bad teachers, making our criterion both theoretically principled and practically efficient.

Finally, we verify the consistency of this criterion across

*Table 2.* **Effect of teacher entropy on student robustness across five adversarial distillation methods.** We report the final Clean, PGD-20, and AutoAttack (AA) accuracies. **GenGap** denotes the generalization gap (Final Robust Train - Final Robust Test), and **Degrad.** measures the robust overfitting (Peak Robust Test - Final Robust Test). Values in parentheses denote the teacher's predictive entropy on unlearnable samples.

| Method | Clean | PGD-20 | AA | GenGap | Degrad. |
|---|---|---|---|---|---|
| AD from Low Entropy Teacher[†] (0.059) | | | | | |
| ARD | 84.26 | 41.51 | 40.07 | 53.38 | 8.04 |
| IAD | 84.41 | 42.01 | 40.49 | 53.17 | 7.62 |
| RSLAD | 84.37 | 40.26 | 38.40 | 57.09 | 7.42 |
| AdaAD | 84.10 | 42.20 | 40.51 | 53.04 | 8.03 |
| IGDM | 84.50 | 46.33 | 43.91 | 49.32 | 5.06 |
| **Average** | **84.33** | **42.46** | **40.68** | **53.20** | **7.23** |
| AD from High Entropy Teacher[‡] (1.882) | | | | | |
| ARD | 85.26 | 54.14 | 49.62 | 22.26 | 0.40 |
| IAD | 84.83 | 52.90 | 48.66 | 29.14 | 0.71 |
| RSLAD | 84.44 | 54.30 | 49.86 | 28.40 | 0.31 |
| AdaAD | 85.69 | 56.41 | 52.08 | 30.56 | 0.50 |
| IGDM | 85.26 | 57.09 | 52.73 | 30.28 | 0.71 |
| **Average** | **85.10** | **54.97** | **50.59** | **28.13** | **0.53** |

[†] Model ID: `Gowal2021Improving_70_16_ddpm_100m`
[‡] Model ID: `Chen2021LTD_WRN34_20` from Table 3

diverse distillation objectives in Table 2. Comparing a Low-Entropy teacher against a High-Entropy teacher reveals a striking contrast in student's learning dynamics. Regardless of the choice of distillation methods, the High-Entropy teacher consistently prevents robust overfitting, suppressing the 'Degrad.'—the performance drop from peak to final test accuracy—to negligible levels (avg. 0.53%). In contrast, the Low-Entropy teacher fails to prevent this collapse, inducing significant degradation (avg. 7.23%). This confirms that the teacher's predictive confidence on unlearnable samples is a primary factor governing robust generalization, outweighing the specific choice of distillation loss.

## 6. Conclusion

This work resolves the paradox in adversarial distillation whereby highly robust teachers can degrade student performance. Through a feature-learning analysis, we identify the root cause as a fundamental representational mismatch on the unlearnable set: when a teacher provides confident supervision on features the student cannot represent, it forces the student to memorize spurious noise, thereby destroying robust generalization. In contrast, a teacher that remains uncertain on this set acts as an implicit regularizer and prevents such overfitting. Translating this insight into practice, we propose the Unlearnable-Entropy Criterion, which efficiently identifies effective teachers using a single proxy model. Overall, our results establish a theoretically grounded principle for teacher selection in AD, replacing ad hoc heuristics with a principled metric that ensures stable robust generalization.

## Impact Statement

This work advances the understanding of adversarial distillation by explaining when teacher supervision improves student robustness and when it instead induces robust overfitting. We identify a consistent subset of training samples that are robustly unlearnable for capacity-limited students and show that these samples drive the teacher-dependent outcomes of distillation. Our theory and experiments indicate that overly confident supervision on this subset can lead the student to fit spurious patterns that are exploitable under adversarial attacks, degrading robust generalization, while higher teacher uncertainty on this subset mitigates the effect. These insights can support more reliable robust training by informing teacher selection and diagnostics, while also highlighting that robustness transfer is not automatic and must be carefully evaluated before deployment. We do not introduce new attack methods; however, understanding these failure modes could be misused to create brittle training setups, so we emphasize using our results to strengthen robustness through principled teacher selection and evaluation.

## Acknowledgement

This work was supported by the National Research Foundation of Korea (NRF) grant funded by the Korea government (MSIT) (No. RS-2024-00408003 and RS-2025-00516153) and the Institute for Information & communications Technology Planning & Evaluation (IITP) grant funded by the Korea government (MSIT) (No. RS-2024-00444862 and RS-2026-25522672).

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

# A. Experimental Analysis Details

## A.1. General Experimental Settings (Section 3, 5 and B)

We conduct our experimental analysis on standard benchmarks including CIFAR-10, CIFAR-100 (Krizhevsky et al., 2009), and Tiny-ImageNet (Le & Yang, 2015). We employ typical data augmentations such as random cropping with 4-pixel padding and random horizontal flipping. All student models are trained using SGD with a momentum of 0.9, weight decay of $2 \times 10^{-4}$, and a batch size of 128 for 200 epochs. The learning rate is initialized at 0.1 and decays by a factor of 10 at epochs 100 and 150. To analyze the mechanisms of robust overfitting, we employ PGD-AT (Madry et al., 2018) and TRADES (Zhang et al., 2019) as standard adversarial training baselines. For adversarial distillation, we adopt ARD (Goldblum et al., 2020) as the default method to investigate the unlearnable sample dynamics, while extending our evaluation to IAD (Zhu et al., 2022), RSLAD (Zi et al., 2021), AdaAD (Huang et al., 2023a), and IGDM (Lee et al., 2025) for universality verification. For method-specific hyperparameters, we adhere to the original configurations reported in their respective papers, with an adversarial budget of $\epsilon = 8/255$. For reproducibility, our code is available at https://github.com/HongsinLee/why-robust-teachers-fail.

We evaluate the robustness of trained models using three key metrics: Clean, PGD-20, and AutoAttack (AA) accuracy. Clean accuracy measures performance on the original test set, while PGD-20 and AA assess robustness against 20-step projected gradient descent (Madry et al., 2018) and the AutoAttack ensemble (Croce & Hein, 2020), respectively, under an adversarial budget of $\epsilon = 8/255$.

For the teacher models, we utilize pre-trained weights from the RobustBench model zoo (Croce et al., 2021), strictly selecting models whose robustness exceeds the performance achievable by the student architecture via standard PGD-AT or TRADES. This ensures that the teacher remains sufficiently more robust than the student throughout our analysis. First, for the CIFAR-10 experiments, as detailed in Table 3, we select four representative teacher models with diverse architectures and robustness levels. These models are primarily used to investigate the detailed dynamics of unlearnable samples. Notably, this selection includes 'Chen' (Chen & Lee, 2025), a standard teacher widely adopted in recent AD literature, and 'Bartoldson' (Bartoldson et al., 2024), representing the current state-of-the-art in robustness. Second, to validate the generalizability of our proposed criterion within CIFAR-10, we extend our evaluation to a broader array of teacher models in Section 5.3, whose comprehensive specifications are provided in Table 4. This expanded set is utilized to validate the correlation between our criterion and the resulting student robustness (Figure 4) and to demonstrate consistent performance trends across various distillation methods (Table 2). Finally, to verify the scalability of our approach on larger datasets, we employ additional robust teachers for CIFAR-100 and Tiny-ImageNet. The detailed specifications for these models are listed in Table 5.

*Table 3.* **Specifications of representative robust teachers trained on CIFAR-10.** We categorize the teachers into 'Good' and 'Bad' based on the classification established in Lee & Chung (2026). 'AA' denotes robust accuracy against AutoAttack.

| Shorthand | Reference | Teacher Type from Lee & Chung (2026) | Architecture | Clean | AA |
|---|---|---|---|---|---|
| Bartoldson | Bartoldson et al. (2024) | Bad Teacher | WRN-94-16 | 93.68 | 73.71 |
| Rebuffi | Rebuffi et al. (2021) | Good Teacher | WRN-70-16 | 88.54 | 64.20 |
| Gowal | Gowal et al. (2021) | Bad Teacher | WRN-28-10 | 87.50 | 63.38 |
| Chen | Chen & Lee (2025) | Good Teacher | WRN-34-10 | 85.21 | 56.94 |

## A.2. Self-Distillation Protocol (Section 3.1)

To investigate the impact of teacher overfitting in a controlled setting (Figure 1b), we employed a self-distillation method designed to isolate the teacher's quality as the sole variable. We first trained a standard ResNet-18 model using PGD-AT for 200 epochs. From this single training trajectory, we extracted two distinct checkpoints to serve as teachers: the *Best Self-Teacher*, selected at the epoch of highest robust test accuracy (typically epoch 100-120) to represent a model with generalizable robust features, and the *Last Self-Teacher*, taken from the final epoch to represent a model that has overfitted to the training set. We then trained new student models of the same architecture using these respective checkpoints via self-distillation (ARD with self-teacher).

*Table 4.* **Comprehensive list of robust teachers trained on CIFAR-10 used in Section 5.3.** This extended set is used to analyze the correlation between our criterion and the resulting student robustness.

| Reference | RobustBench Model ID | Architecture | Clean | AA |
|---|---|---|---|---|
| Bartoldson et al. (2024) | `Bartoldson2024Adversarial_WRN-94-16` | WRN-94-16 | 93.68 | 73.71 |
| Amini et al. (2024) | `Amini2024MeanSparse_S-WRN-94-16` | WRN-94-16 | 93.60 | 73.10 |
| Bartoldson et al. (2024) | `Bartoldson2024Adversarial_WRN-82-8` | WRN-82-8 | 93.11 | 71.59 |
| Peng et al. (2023) | `Peng2023Robust` | WRN-82-8 | 93.27 | 71.07 |
| Wang et al. (2023) | `Wang2023Better_WRN-70-16` | WRN-70-16 | 93.25 | 70.69 |
| Amini et al. (2024) | `Amini2024MeanSparse_Ra_WRN_70_16` | WRN-70-16 | 93.24 | 68.94 |
| Cui et al. (2024) | `Cui2023Decoupled_WRN-28-10` | WRN-28-10 | 92.16 | 67.73 |
| Gowal et al. (2021) | `Gowal2021Improving_70_16_ddpm_100m` | WRN-70-16 | 88.74 | 66.10 |
| Gowal et al. (2020) | `Gowal2020Uncovering_70_16_extra` | WRN-70-16 | 91.10 | 65.87 |
| Rebuffi et al. (2021) | `Rebuffi2021Fixing_70_16_cutmix_ddpm` | WRN-70-16 | 88.54 | 64.20 |
| Gowal et al. (2021) | `Gowal2021Improving_28_10_ddpm_100m` | WRN-28-10 | 87.50 | 63.38 |
| Huang et al. (2021) | `Huang2021Exploring_ema` | WRN-34-R | 91.23 | 62.54 |
| Dai et al. (2022) | `Dai2021Parameterizing` | WRN-28-10 | 87.02 | 61.55 |
| Sridhar et al. (2022) | `Sridhar2021Robust_34_15` | WRN-34-15 | 86.53 | 60.41 |
| Carmon et al. (2019) | `Carmon2019Unlabeled` | WRN-28-10 | 89.69 | 59.53 |
| Gowal et al. (2021) | `Gowal2021Improving_R18_ddpm_100m` | PreActRN-18 | 87.35 | 58.50 |
| Chen & Lee (2025) | `Chen2021LTD_WRN34_20` | WRN-34-20 | 86.03 | 57.71 |
| Chen & Lee (2025) | `Chen2021LTD_WRN34_10` | WRN-34-10 | 85.21 | 56.94 |
| Sehwag et al. (2022) | `Sehwag2021Proxy_R18` | RN-18 | 84.59 | 55.54 |

*Table 5.* **Specifications of robust teachers trained on CIFAR-100 and Tiny-ImageNet.** These models are employed to verify the scalability of our approach across larger datasets. 'AA' denotes robust accuracy against AutoAttack.

| Shorthand | Reference | RobustBench Model ID | Architecture | Clean | AA |
|---|---|---|---|---|---|
| **CIFAR-100** | | | | | |
| **Chen** | Chen & Lee (2025) | `Chen2021LTD_WRN34_10` | WRN-34-10 | 64.07 | 30.59 |
| **Wang28** | Wang et al. (2023) | `Wang2023Better_WRN-28-10` | WRN-28-10 | 72.58 | 38.77 |
| **Wang70** | Wang et al. (2023) | `Wang2023Better_WRN-70-16` | WRN-70-16 | 75.22 | 42.66 |
| **Gowal** | Gowal et al. (2020) | `Gowal2020Uncovering_extra` | WRN-70-16 | 69.15 | 36.88 |
| **Tiny-ImageNet** | | | | | |
| **Wang** | Wang et al. (2023) | Official Repository[†] | WRN-28-10 | 65.19 | 31.30 |

[†] Pre-trained weights obtained from `https://github.com/wzekai99/DM-Improves-AT`

## A.3. Identification of Learnable and Unlearnable Sets (Section 3.2)

To construct the consistent learnable ($\mathcal{S}_L$) and unlearnable ($\mathcal{S}_U$) subsets, we strictly followed the intersection protocol outlined in Algorithm 1.

We compiled an ensemble of models trained via diverse methodologies ($\mathcal{P}$), comprising standard adversarial training baselines (PGD-AT and TRADES) and ARD. For CIFAR-10 and CIFAR-100, we utilized ARD configured with the four distinct robust teachers detailed in Table 3 and Table 5, resulting in six paradigms ($|\mathcal{P}| = 6$) and a total ensemble of 60 candidate models ($|\mathcal{M}| = 60$) across $K = 10$ random seeds. For Tiny-ImageNet, we adopted a representative subset comprising PGD-AT, TRADES, and ARD with the "Wang" teacher, yielding three paradigms ($|\mathcal{P}| = 3$) and a total of 30 candidate models ($|\mathcal{M}| = 30$).

The identification process was conducted on the full training dataset $\mathcal{D}$ for each benchmark ($N = 50,000$ for CIFAR-10/100; $N = 100,000$ for Tiny-ImageNet). To isolate intrinsic data hardness from the transient degradation of late-stage overfitting, every model in $\mathcal{M}$ was evaluated specifically at its epoch of peak robust test accuracy.

Based on the predictions at these peak-performance checkpoints, we categorized the samples using a strict intersection criterion. A sample is defined as learnable ($\mathcal{S}_L$) if and only if it is correctly classified by all models in the ensemble (i.e., all 60 models for CIFAR or 30 for Tiny-ImageNet). Conversely, a sample is designated as unlearnable ($\mathcal{S}_U$) if it is consistently misclassified by every model.

This rigorous selection ensures that $\mathcal{S}_U$ captures samples that are universally difficult regardless of the loss function. As

---

**Algorithm 1** Identification of Learnable ($\mathcal{S}_L$) and Unlearnable ($\mathcal{S}_U$) Sets

---

1: **Input:** Training dataset $\mathcal{D} = \{(x_i, y_i)\}_{i=1}^N$
2: **Input:** Set of training paradigms $\mathcal{P}$
3: **Input:** Number of random seeds $K$ per paradigm
4: **Output:** Learnable set $\mathcal{S}_L$, Unlearnable set $\mathcal{S}_U$
5: Initialize $\mathcal{S}_L \leftarrow \emptyset$, $\mathcal{S}_U \leftarrow \emptyset$
6: $\mathcal{M} \leftarrow \emptyset$                                          # Collection of all trained models
7: **for** each paradigm $p \in \mathcal{P}$ **do**
8:     **for** seed $k = 1$ to $K$ **do**
9:         Train model $\theta_{p,k}$ using method $p$
10:        Identify epoch $t^*$ achieving peak robust test accuracy
11:        Save model snapshot $f_{p,k} \leftarrow \theta_{p,k}^{(t^*)}$
12:        $\mathcal{M} \leftarrow \mathcal{M} \cup \{f_{p,k}\}$
13:    **end for**
14: **end for**
15: **for** each sample $(x_i, y_i) \in \mathcal{D}$ **do**
16:    $C_i \leftarrow 0$                                          # Count of correct robust predictions
17:    **for** each model $f \in \mathcal{M}$ **do**
18:        **if** $f(x_i + \delta) = y_i$ under PGD-10 attack **then**
19:            $C_i \leftarrow C_i + 1$
20:        **end if**
21:    **end for**
22:    **if** $C_i = |\mathcal{M}|$ **then**
23:        $\mathcal{S}_L \leftarrow \mathcal{S}_L \cup \{i\}$                                          # Consistent Learnable
24:    **else if** $C_i = 0$ **then**
25:        $\mathcal{S}_U \leftarrow \mathcal{S}_U \cup \{i\}$                                          # Consistent Unlearnable
26:    **end if**
27: **end for**
28: **return** $\mathcal{S}_L, \mathcal{S}_U$

---

shown in Tables 1 and 7, the size of $\mathcal{S}_U$ varies inversely with model capacity. This empirical evidence supports our theoretical premise that unlearnability is a relative property arising from the representational limits of the student architecture.

### A.4. Feature Reconstruction Methodology (Section 3.2)

To visualize the internal representations learned by the student model, we employ the robust feature inversion technique (Engstrom et al., 2019). The objective is to synthesize an input image $\mathbf{x}_{\text{recon}}$ that elicits the same feature activations in the model's penultimate layer as the target image $\mathbf{x}_{\text{target}}$. Formally, letting $h(\cdot)$ denote the feature extractor (i.e., the network up to the penultimate layer), we solve the following optimization problem:

$$\mathbf{x}_{\text{recon}} = \underset{\mathbf{x}' \in [0,1]^d}{\arg\min} \|h(\mathbf{x}') - h(\mathbf{x}_{\text{target}})\|_2^2. \tag{20}$$

The optimization initializes $\mathbf{x}'$ as random noise sampled from a uniform distribution $\mathcal{U}(0, 1)$ and iteratively updates it to minimize the Mean Squared Error in the feature space.

This reconstruction process serves as a diagnostic tool to analyze the semantic quality of learned features. As illustrated in Figure 2, the results exhibit a sharp contrast correlated with the learnability of the samples. For samples in the learnable set (top rows), the reconstructions retain high visual fidelity, preserving clear object shapes and semantic structures. This indicates that the model relies on robust, human-aligned features. Conversely, for samples in the unlearnable set (bottom rows), the reconstructions degrade into unstructured, high-frequency noise patterns or distorted artifacts. This visual evidence confirms that the student model fails to extract coherent features from these samples.

As described in Algorithm 2, we perform the inversion using the Adam optimizer. For our experiments, we set the number of steps $T = 5,000$ and the learning rate $\eta = 0.01$. At each step, pixel values are clipped to the valid range $[0, 1]$. We

---

**Algorithm 2** Robust Feature Reconstruction

---

1: **Input:** Target image $\mathbf{x}_{\text{target}}$, Feature extractor $h$, Number of steps $T$, Learning rate $\eta$
2: **Output:** Reconstructed image $\mathbf{x}_{\text{recon}}$
3: $\mathbf{z}_{\text{target}} \leftarrow h(\mathbf{x}_{\text{target}}).\text{detach}()$        # Extract target features
4: $\mathbf{x}_{\text{recon}} \leftarrow \mathbf{n}$, where $\mathbf{n} \sim \mathcal{U}(0, 1)$        # Initialize with random noise
5: Initialize Adam optimizer with parameters $\mathbf{x}_{\text{recon}}$ and lr $\eta$
6: **for** $t = 1$ to $T$ **do**
7:     $\mathbf{z}_{\text{current}} \leftarrow h(\mathbf{x}_{\text{recon}})$
8:     $\mathcal{L} \leftarrow \|\mathbf{z}_{\text{current}} - \mathbf{z}_{\text{target}}\|_2^2$        # MSE Loss in feature space
9:     Update $\mathbf{x}_{\text{recon}}$ using $\nabla_{\mathbf{x}_{\text{recon}}} \mathcal{L}$
10:     $\mathbf{x}_{\text{recon}} \leftarrow \text{clip}(\mathbf{x}_{\text{recon}}, 0, 1)$        # Enforce valid pixel range
11: **end for**
12: **return** $\mathbf{x}_{\text{recon}}$

---

apply this reconstruction to PGD-perturbed images to analyze the stability of the learned representations under adversarial perturbations.

### A.5. Simulations on Synthetic Data (Section 5.1)

To corroborate our theoretical analysis, we perform a controlled simulation mirroring the setting in Definition 4.1. We construct a synthetic dataset with dimension $d = 100$, sample size $N = 200$, and $P = 4$ patches. The signal strength is set to $\alpha = 5$, and the noise component is sampled with $\sigma_n = 0.4$. The teacher margin is set to $\Gamma = 10$, and the student model is a two-layer neural network with $m = 80$ and $\sigma_0 = 0.01$. To strictly enforce the representational mismatch, we constrain the student's weights to be orthogonal to $\mathbf{v}$, thereby rendering the unlearnable feature structurally invisible to the model. We train the network for $T = 4,000$ steps with a learning rate of $\eta = 0.01$ and an adversarial perturbation budget of $\epsilon = 0.5$. For robust evaluation, we report robust test accuracy against a 20-steps PGD attack on 100 unseen samples sampled from the same distribution as the training set.

### A.6. Calculation of Unlearnable Entropy (Section 5.3)

We quantify a teacher's predictive uncertainty using the Shannon entropy of its softmax output. Given a teacher $f_{\mathbf{W}_T}$ and an unlearnable set $\mathcal{S}_U$ identified for a student model $f_{\mathbf{W}}$, we compute the teacher's average entropy on adversarial examples of samples in $\mathcal{S}_U$. Specifically, for each training sample $(\mathbf{X}_i, y_i) \in \mathcal{S}_U$, we generate a PGD-10 adversarial example $\tilde{\mathbf{X}}_i$ using the student model $f_{\mathbf{W}}$, and evaluate the teacher on $\tilde{\mathbf{X}}_i$. We then compute:

$$H(\mathcal{S}_U) = \frac{1}{|\mathcal{S}_U|} \sum_{i \in \mathcal{S}_U} \left( -\sum_{c=1}^{C} p_T(\tilde{\mathbf{X}}_i)_c \log p_T(\tilde{\mathbf{X}}_i)_c \right), \tag{21}$$

where $p_T(\tilde{\mathbf{X}}_i)_c$ is the predicted probability for class $c$.

## B. Additional Experiments

### B.1. Verifying the Mechanism of Robust Overfitting: A Random Label Test

We hypothesize that robust overfitting is primarily caused by the model memorizing noise patterns in the unlearnable set $\mathcal{S}_U$. To verify this, we conducted a random label test. The rationale is simple: if the model learns meaningful features from $\mathcal{S}_U$, randomizing the labels should disrupt the learning process. However, if the model minimizes the loss on $\mathcal{S}_U$ merely by memorizing noise, the overfitting pattern should remain similar even with random labels, as the model can fit noise to any label given sufficient capacity.

We constructed a training set where the labels of $\mathcal{S}_U$ were randomly shuffled, while the labels of $\mathcal{S}_L$ remained correct. We then trained a ResNet-18 model using Standard PGD-AT. The results are shown in Figure 5. First, looking at Figure 5a, the model achieves nearly 100% robust training accuracy on the randomized $\mathcal{S}_U$. This confirms that the model memorize these samples even when the labels are meaningless. More importantly, the robust test accuracy shows the same degradation

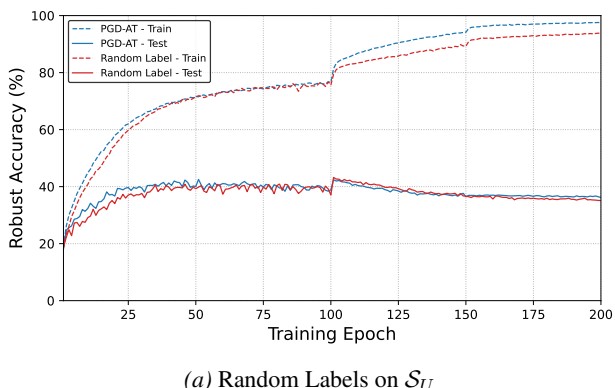

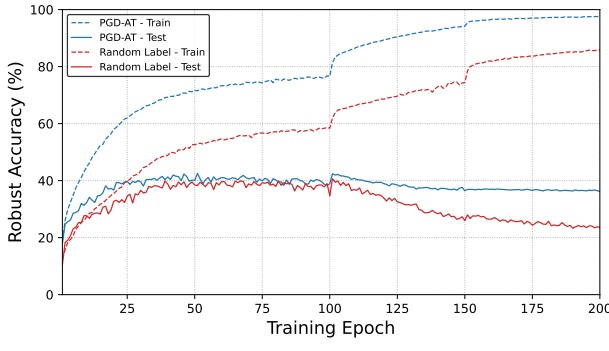

*(a)* Random Labels on $\mathcal{S}_U$          *(b)* Random Labels on $\mathcal{S}_L$ (Partial)

*Figure 5.* **Validation of the robust overfitting mechanism via the Random Label Test.** (a) Randomizing labels within the unlearnable set ($\mathcal{S}_U$) yields a robust test accuracy trajectory indistinguishable from the baseline Standard AT. This implies that the model memorizes $\mathcal{S}_U$ as arbitrary noise patterns regardless of the semantic labels. (b) Conversely, randomizing an equivalent number of learnable samples ($\mathcal{S}_L$) severely degrades generalization, confirming that $\mathcal{S}_L$ contains the necessary robust features.

pattern as the standard training scenario where $\mathcal{S}_U$ has correct labels. This indicates that fitting $\mathcal{S}_U$ is practically equivalent to fitting random noise.

In contrast, we performed a counter-experiment on the learnable set $\mathcal{S}_L$. To ensure a fair comparison, we randomly shuffled the labels of a subset of $\mathcal{S}_L$ equal in size to $\mathcal{S}_U$, while keeping the rest unchanged. As shown in Figure 5b, this corruption significantly degraded the model's generalization performance compared to the baseline. This sharp contrast confirms that $\mathcal{S}_L$ contains valid robust signals necessary for generalization, whereas $\mathcal{S}_U$ acts as noise that induces overfitting regardless of the label correctness.

## B.2. Noise Overfitting and Robustness Decay

To understand the mechanism of robust overfitting, we analyze the robustness changes of test samples from the peak robustness checkpoint (Best) to the overfitted checkpoint (Last). Using ResNet-18 models trained via both Standard PGD-AT and TRADES, we define the Forgotten Set as the subset of test samples correctly classified under a PGD-20 attack by the Best model but misclassified by the Last model.

We first quantify the prevalence of this phenomenon. As reported in Table 6, the Forgotten Set constitutes a significant portion of the test data ($10.21\%$ for PGD-AT and $7.32\%$ for TRADES). This confirms that the robustness degradation is not limited to rare outliers but affects a substantial fraction of the data distribution. To analyze the nature of this degradation, we evaluate the Last model across perturbation budgets ranging from 0 to $8/255$ on these subsets. The results, illustrated in Figure 6, show a consistent trend across both training methods regarding the Forgotten Set (Red line). With no perturbation ($\epsilon = 0$), the Last model achieves high accuracy ($> 95\%$) on the Forgotten Set, comparable to the Consistent Learnable Set (Blue line). This indicates that the model retains its generalization capability for the semantic features of these samples. However, as the perturbation size increases, the accuracy of the Forgotten Set drops sharply, reaching near zero at the maximum budget.

This distinct decay profile suggests that robust overfitting reduces the robustness margin of test samples without losing their semantic features. While the model correctly classifies clean data, overfitting to training noise distorts the decision boundary, making these specific samples highly sensitive to perturbations. Consequently, they become vulnerable to adversarial attacks despite being correctly classified in clean settings. This empirical behavior mirrors our theoretical findings in Theorem 4.7:

*Table 6.* **Distribution of test sample subsets based on training dynamics (Best vs. Last).** We report the percentage of samples in each category (mean $\pm$ std over 10 seeds) for PGD-AT and TRADES. The 'Forgotten' subset represents samples that were robust at the peak epoch but lost robustness due to overfitting.

| Model | Consistent Learnable | Forgotten | Newly Learned | Consistent Unlearnable |
|-------|---------------------|-----------|---------------|------------------------|
| PGD-AT | $34.36 \pm 0.46$ | $10.21 \pm 0.57$ | $3.30 \pm 0.25$ | $52.13 \pm 0.39$ |
| TRADES | $40.24 \pm 0.44$ | $7.32 \pm 0.36$ | $3.72 \pm 0.18$ | $48.72 \pm 0.59$ |

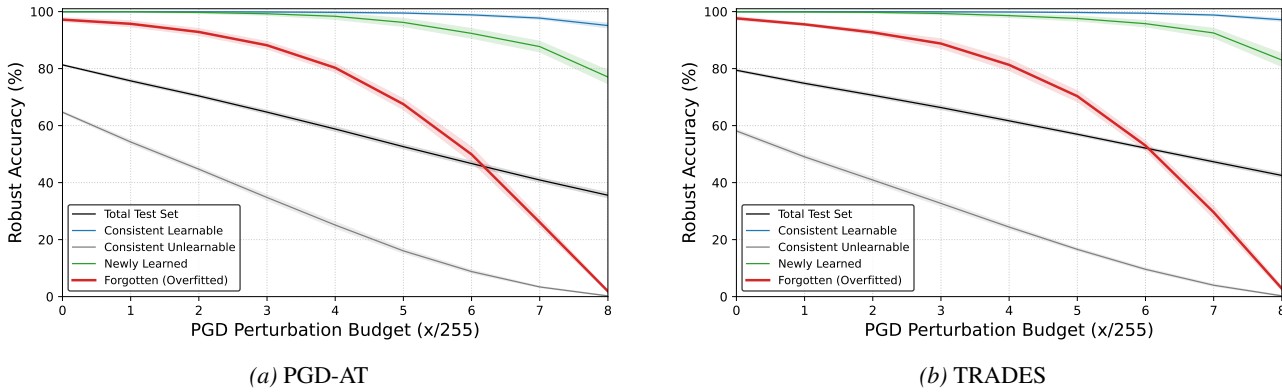

*(a)* PGD-AT            *(b)* TRADES

*Figure 6.* **Robustness decay profiles across perturbation budgets.** We trace the accuracy of different sample subsets as the attack strength ($\epsilon$) increases. The *Consistent Learnable* set (Blue) maintains stability, indicating robust feature learning. Crucially, the *Forgotten (Overfitted)* set (Red) shows high accuracy at $\epsilon = 0$ but suffers a catastrophic drop as perturbation increases. This confirms that robust overfitting does not erase clean signal features but rather compromises the robust margin, likely by overfitting to non-robust noise.

the model successfully learns the robust feature $\mathbf{u}$ (high clean accuracy) but simultaneously memorizes spurious noise, which the adversary exploits to override the robust prediction.

### B.3. Unlearnable Sample in Different Dataset

To verify the universality of our findings, we extend the identification of robustly unlearnable samples to CIFAR-100 and Tiny-ImageNet. As reported in Table 7 and Table 8, a consistent subset of unlearnable samples persists across all datasets and architectures. These results establish that the existence of $\mathcal{S}_U$ is not an artifact of specific settings, but a fundamental and pervasive characteristic of robust learning tasks.

*Table 7.* **Identification of robustly unlearnable samples on CIFAR-100.** We categorize CIFAR-100 training samples based on the rigorous intersection of predictions from 60 models (spanning 6 training paradigms × 10 random seeds, including AD from 4 teachers detailed in Table 5), evaluated at their respective peak robust accuracy epochs (see Section A.3 and Algorithm 1). The 'Intersection' reveals a persistent core of samples that remain unlearnable regardless of the defense method employed.

| Model | Category | Adversarial Training | | Adversarial Distillation | | | | Intersection |
|---|---|---|---|---|---|---|---|---|
| | | PGD-AT | TRADES | Chen | Wang28 | Wang70 | Gowal | |
| MN-V2 | Unlearnable | 22,184 | 17,948 | 19,720 | 22,115 | 21,973 | 21,986 | 14,727 |
| | Learnable | 13,816 | 14,625 | 22,054 | 16,988 | 14,845 | 14,993 | 10,971 |
| ResNet-18 | Unlearnable | 8,942 | 10,422 | 17,241 | 12,485 | 10,727 | 12,003 | 6,867 |
| | Learnable | 19,178 | 19,000 | 25,172 | 26,626 | 22,258 | 23,117 | 15,251 |
| WRN-28-10 | Unlearnable | 3,315 | 4,468 | 15,394 | 8,771 | 3,010 | 5,383 | 1,652 |
| | Learnable | 30,795 | 27,328 | 28,213 | 33,267 | 33,497 | 40,178 | 15,614 |
| WRN-34-10 | Unlearnable | 2,668 | 3,343 | 15,119 | 8,529 | 2,849 | 5,240 | 1,559 |
| | Learnable | 33,046 | 30,086 | 28,448 | 32,891 | 40,747 | 40,234 | 16,397 |

*Table 8.* **Identification of robustly unlearnable samples on Tiny-ImageNet.** We categorize training samples based on the intersection of predictions from PreActResNet-18 models, evaluated at their respective peak robust accuracy epochs. The 'Intersection' reveals a persistent core of samples that remain unlearnable regardless of the defense method employed.

| Category | Adversarial Training | | AD | Intersection |
|---|---|---|---|---|
| | PGD-AT | TRADES | Wang | |
| Unlearnable | 42,204 | 29,389 | 44,372 | 25,391 |
| Learnable | 23,997 | 21,839 | 39,936 | 17,896 |

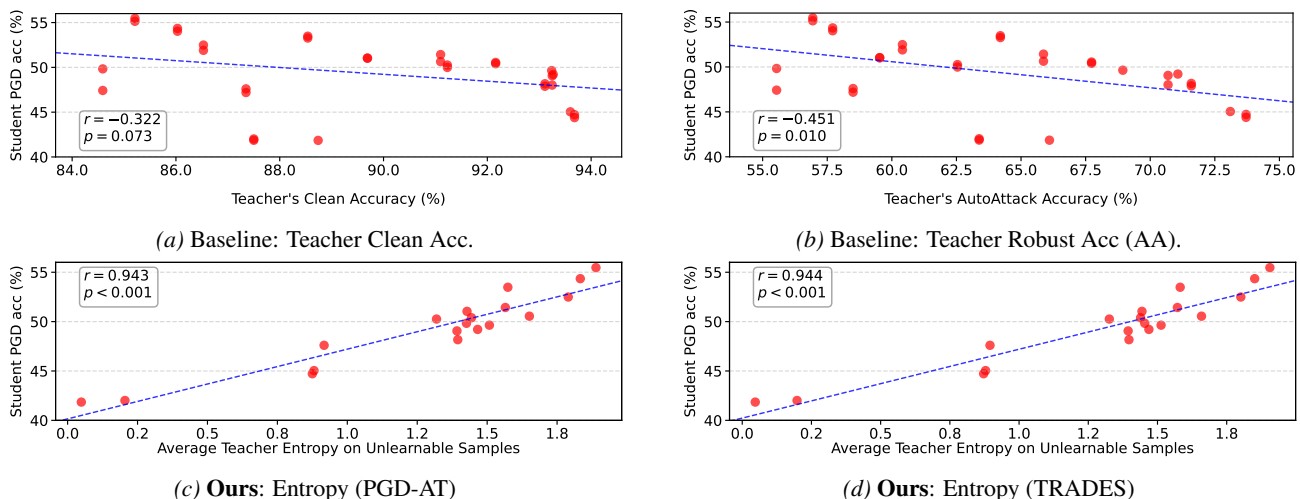

*Figure 7.* **Comparison of teacher selection criteria.** Top row: Existing selection heuristics based on (a) Clean Accuracy and (b) Robust Accuracy (AutoAttack) show no correlation with student robustness. Bottom row: Our proposed *Unlearnable-Confidence Criterion* consistently exhibits a strong correlation ($r > 0.9$) regardless of the reference model used for identification ((c) PGD-AT or (d) TRADES). This confirms that teacher uncertainty on unlearnable samples is the dominant predictor of distillation success.

### B.4. Further Validation of the Unlearnable-Entropy Criterion

We further substantiate the validity of our criterion by comparing it against conventional teacher selection heuristics in Figure 7. As shown in the top row, standard metrics such as Teacher Clean Accuracy and Robust Accuracy fail to predict student performance, exhibiting weak or negligible correlations. In contrast, our Unlearnable-Confidence Criterion (bottom row) demonstrates a strong positive correlation with student robustness. Crucially, this high predictive power remains consistent regardless of whether the unlearnable set is approximated via PGD-AT or TRADES, confirming that the teacher's uncertainty on unlearnable samples is the decisive factor for successful distillation.

### B.5. Sensitivity to Perturbation Bounds

We investigate the sensitivity of the robustly unlearnable set ($\mathcal{S}_U$) to varying perturbation budgets ($\epsilon$) and evaluate the stability of our proposed teacher-selection metric under these changing constraints. As reported in Table 9, the size of the unlearnable intersection changes substantially depending on the attack budget. When the adversarial constraint is relaxed, the number of universally unlearnable samples strictly decreases. This dynamic expansion confirms that unlearnability is not an intrinsic property of the data alone, but a relative phenomenon dynamically induced by the interplay between the student architecture and the adversarial constraint.

Furthermore, Table 10 supports the generalizability of our metric: for standard bounds ($\epsilon \geq 4/255$), student robust accuracy aligns with the teacher's predictive entropy ordering. The boundary case of $\epsilon = 2/255$ is also consistent with our theoretical framework; at this minimal budget, $\mathcal{S}_U$ nearly vanishes (Table 9), mitigating the representational mismatch. With the reduction of $\mathcal{S}_U$, the degradation typically caused by low-entropy teachers is alleviated, and the performance gap among distinct teachers narrows. This behavior suggests that the presence of $\mathcal{S}_U$ is a primary factor driving teacher-induced failure modes in adversarial distillation.

## C. Full Related Works

### C.1. Adversarial Vulnerability of Deep Neural Networks

Deep neural networks are inherently vulnerable to adversarial examples—indistinguishable perturbations designed to maximize the classification error (Szegedy et al., 2014; Goodfellow et al., 2015). The generation of such examples is generally categorized based on the adversary's access to the victim model: white-box and black-box settings. In the white-box setting, the adversary has full access to the model parameters and gradients. This allows for the use of iterative gradient-based methods, such as PGD (Madry et al., 2018), or more sophisticated attacks like C&W (Carlini & Wagner, 2017) and AutoAttack (Croce & Hein, 2020), to directly maximize the loss. In contrast, the black-box setting assumes

*Table 9.* **Sensitivity of robustly unlearnable samples to varying perturbation bounds.** Unlike the baseline subset identification performed at a fixed attack budget, this experiment evaluates how the learnable and unlearnable sets evolve under different training and test bounds ($\epsilon$). At each $\epsilon$ level, we categorize training samples based on the rigorous intersection of predictions from 60 models. The 'Intersection' column reports the size of the consistent subsets: samples correctly classified by all models (Learnable) or misclassified by all models (Unlearnable). Crucially, the unlearnable intersection expands significantly as $\epsilon$ increases.

| Epsilon | Category | Adversarial Training | | Adversarial Distillation | | | | Intersection |
|---|---|---|---|---|---|---|---|---|
| | | PGD-AT | TRADES | Chen | Rebuffi | Bartoldson | Gowal | |
| 2/255 | Unlearnable | 106 | 2842 | 3,898 | 2,890 | 419 | 231 | 38 |
| | Learnable | 44,361 | 39,811 | 43,274 | 44,024 | 46,036 | 45,593 | 36,654 |
| 4/255 | Unlearnable | 1,130 | 4,796 | 5,707 | 4,188 | 1,472 | 1,413 | 642 |
| | Learnable | 38.299 | 35,667 | 40,576 | 41,617 | 39,620 | 37,592 | 31,025 |
| 6/255 | Unlearnable | 4,215 | 6,842 | 7,885 | 5,981 | 3,395 | 4,225 | 2,518 |
| | Learnable | 31,395 | 31,567 | 37,419 | 38,836 | 33,066 | 31,369 | 26,242 |
| 8/255 | Unlearnable | 8,360 | 10,217 | 10,464 | 8,627 | 7,511 | 8,047 | 5,217 |
| | Learnable | 25,927 | 26,903 | 34,007 | 32,714 | 27,598 | 26,323 | 21,899 |
| 10/255 | Unlearnable | 12,323 | 13,910 | 13,343 | 10,298 | 12,025 | 12,442 | 7,865 |
| | Learnable | 22,485 | 22,096 | 30,466 | 32,562 | 23,189 | 22,466 | 18,357 |

*Table 10.* **Robust test accuracy across varying perturbation budgets** ($\epsilon$). We evaluate the adversarial distillation performance of a ResNet-18 student using four different teachers. Values in parentheses denote the teacher's predictive entropy on the unlearnable subset.

| Epsilon | Teacher (Entropy) | | | |
|---|---|---|---|---|
| | **Chen** (1.906) | **Rebuffi** (1.582) | **Bartoldson** (0.872) | **Gowal** (0.198) |
| 2/255 | 80.18 | 81.42 | 79.89 | 79.46 |
| 4/255 | 72.03 | 71.13 | 66.70 | 65.47 |
| 6/255 | 63.37 | 62.12 | 55.05 | 52.34 |
| 8/255 | 55.47 | 53.48 | 44.39 | 41.85 |
| 10/255 | 48.11 | 44.46 | 37.27 | 35.31 |

the adversary has no access to the internal gradients, relying solely on the model's output. Black-box attacks are further divided into two main categories: query-based and transfer-based attacks. Query-based attacks estimate gradients or search for decision boundaries by repeatedly querying the model outputs, using techniques such as score-based methods (Uesato et al., 2018; Andriushchenko et al., 2020) or decision-based boundary attacks (Chen & Gu, 2020; Chen et al., 2020a; 2021b). Transfer-based attacks exploit the phenomenon of adversarial transferability, where adversarial examples generated on a surrogate model are used to attack the target model without direct access (Szegedy et al., 2014; Papernot et al., 2016; Liu et al., 2017; Papernot et al., 2017; Tramèr et al., 2017; Mahmood et al., 2021a)

While the above methodologies have been primarily developed and evaluated in the context of convolutional image classifiers, adversarial vulnerability itself is not limited to specific architectures or modalities; it appears to be a pervasive property of deep learning models across vision, language, and speech (Carlini & Wagner, 2018; Jin et al., 2020; Bhojanapalli et al., 2021; Mahmood et al., 2021b). Furthermore, this threat extends to large-scale foundation models: Vision-Language Models like CLIP can be misled by multimodal attacks (Goh et al., 2021; Mao et al., 2023), and Large Language Models are vulnerable to jailbreaking attacks that bypass safety alignments (Wei et al., 2023; Yu et al., 2023; Zou et al., 2023; Liu et al., 2024; Mehrotra et al., 2024; Chao et al., 2025). This pervasive nature underscores the fundamental necessity of developing robust learning mechanisms.

### C.2. Empirical Advances in Adversarial Training

Adversarial training (AT), typically formulated as a min–max robust optimization problem (Madry et al., 2018), is widely used in practice and can withstand strong adaptive attacks (Athalye et al., 2018). To better balance natural and robust accuracy, TRADES (Zhang et al., 2019) adds a KL-divergence regularizer to smooth the decision boundary, while MART (Wang

et al., 2020) explicitly emphasizes misclassified examples via a margin-based loss. Building on these instance-reweighting principles, subsequent work further shapes the robust objective through geometry-aware sampling and related reweighting schemes (Moosavi-Dezfooli et al., 2019; Bai et al., 2021; Liu et al., 2021; Zhang et al., 2021; Jin et al., 2022), and another line leverages an auxiliary standard model to counteract excessive margins (Rade & Moosavi-Dezfooli, 2022). Beyond objective design, extensive research has focused on optimization dynamics and data diversity, showing that carefully tuned training hyperparameters (Pang et al., 2021), weight-space perturbations (Wu et al., 2020; Zhang et al., 2024), and strong augmentations (Rebuffi et al., 2021; Wang et al., 2023) improve robustness.

Beyond conventional image classifiers, AT has also been extended to large-scale foundation models: adversarially fine-tuned vision–language models aim to improve robustness against multimodal perturbations and zero-shot robustness (Mao et al., 2023; Schlarmann et al., 2024; Wang et al., 2024; Yu et al., 2024; Dong et al., 2025; 2026; Wang et al., 2026), and large language models are adversarially trained to withstand jailbreak prompts, thereby strengthening safety alignment (Mazeika et al., 2024; Xhonneux et al., 2024; Casper et al., 2025; Sheshadri et al., 2025). This cross-modal trend highlights AT as a key mechanism for achieving robustness across architectures and modalities.

### C.3. Difficulties in Adversarial Training

Despite these advances, adversarial training remains difficult to deploy reliably in practice. A central failure mode is robust overfitting, where robust accuracy degrades sharply in the later stages of training (Rice et al., 2020), and empirical studies point to sharp loss landscapes and memorization of non-robust examples as key factors (Rice et al., 2020; Dong et al., 2022; Yu et al., 2022; Li & Spratling, 2023). Mitigation strategies such as early stopping and weight averaging have been shown to partially alleviate these issues (Rice et al., 2020; Izmailov et al., 2018; Gowal et al., 2020; Jia et al., 2024). Robust performance is also much more sensitive to model capacity and architecture than standard accuracy: larger, carefully designed networks typically achieve substantially higher robust accuracy than compact models under the same adversarial training protocol (Gowal et al., 2020; Huang et al., 2021; Rebuffi et al., 2021). Together, these issues make it difficult to obtain practical robust models under realistic resource constraints and motivate mechanisms that decouple robustness from a specific architecture and training schedule. Adversarial distillation is one such approach, transferring robustness from high-capacity teachers to smaller, more deployable students.

### C.4. Empirical Works on Adversarial Distillation

Adversarial distillation (AD) transfers robustness from a teacher to a student by training the student to match teacher signals on adversarially perturbed inputs (Goldblum et al., 2020; Zhu et al., 2022; Maroto et al., 2022; Zi et al., 2021; Huang et al., 2023a; Kuang et al., 2023; Lee et al., 2025). In contrast to standard knowledge distillation, which aligns clean predictions (Hinton et al., 2015), AD integrates the adversarial inner loop into the distillation objective so that robustness is preserved in the student. Early work such as ARD (Goldblum et al., 2020) showed that robust teachers can substantially improve student robustness when both teacher and student are trained under adversarial settings. Subsequent methods refine how teacher information is used: RSLAD (Zi et al., 2021) guides adversarial example generation with teacher logits and reports a robust saturation effect, where student robustness improves with teacher strength only up to a point before degrading; AKD (Maroto et al., 2022) and IAD (Zhu et al., 2022) adapt the distillation signal based on teacher confidence or agreement; AdaAD (Huang et al., 2023a) and IGDM (Lee et al., 2025) further involve the teacher in the inner maximization or gradient matching to better align student updates with robust teacher behavior. Empirically, distilling from early-stage or well-generalized teachers has been observed to mitigate robust overfitting and stabilize AD (Chen et al., 2020b; Zi et al., 2021).

Beyond vanilla robustness, AD has been extended to class-imbalanced and long-tailed regimes (Yue et al., 2023; Zhao et al., 2024; Cho et al., 2025b), incremental and continual learning (Cho et al., 2025a), and self-distillation settings (Jung et al., 2024), indicating that robustness transfer via a teacher–student interface is a flexible tool across tasks and data regimes. A complementary line of work seeks robustness transfer without an explicit adversarial inner loop, replacing PGD-based examples with gradient or feature matching on clean inputs (Chan et al., 2020; Shafahi et al., 2020; Awais et al., 2021a;b; Chen et al., 2021a; Shao et al., 2021; Vaishnavi et al., 2022). These non–AT-based approaches primarily target computational efficiency and typically trade off some robustness, and are therefore orthogonal to our focus on understanding the limitations and failure modes of AT-based adversarial distillation.

Recently, SAAD (Lee & Chung, 2026) challenged the capacity gap hypothesis (Zi et al., 2021), attributing robust saturation instead to the scarcity of Transferable Adversarial Samples (TAS)–inputs where student perturbations successfully degrade

the teacher's confidence. They empirically showed that robust teachers remain overconfident on non-transferable samples, inducing high adversarial variance and triggering robust overfitting. However, while SAAD identified TAS as a key empirical indicator, they did not explain the theoretical origin of this failure. In this work, we bridge this gap by proving that the failure arises when the robust teacher enforces supervision on unlearnable structures. We demonstrate that this guidance compels the student–who is incapable of robustly acquiring these features–to instead memorize noise to match the teacher's predictions.

### C.5. Theoretical Understanding of Adversarial Training

Theoretical understanding of adversarial training is difficult in general due to the non-convex min–max objective, so many analyses focus on simplified settings. A series of works studies adversarial training of linear models, where the inner maximization is tractable and one can obtain robust risk guarantees under stylized data assumptions (Li et al., 2020; Zou et al., 2021; Javanmard & Soltanolkotabi, 2022; Chen et al., 2023). Another line considers neural networks in a lazy regime to make the dynamics analyzable (Gao et al., 2019; Zhang et al., 2020; Li & Telgarsky, 2023). However, recent work suggests a fundamental conflict in this regime; Wang et al. (2022) proved that neural networks trained in the lazy regime remain vulnerable to adversarial attacks, implying that robustness may be incompatible with lazy training in general. This limitation necessitates a shift towards understanding the optimization dynamics beyond the lazy regime.

### C.6. Feature Learning Theory of Deep Learning

The feature learning theory (Wen & Li, 2021; Allen-Zhu & Li, 2022; Cao et al., 2022; Jelassi & Li, 2022; Allen-Zhu & Li, 2023a;b; Chidambaram et al., 2023; Huang et al., 2023b; Kou et al., 2023; Lu et al., 2024; Oh & Yun, 2024; Oh et al., 2026) explores how representations are dynamically acquired in deep learning tasks. By employing patch-wise structured data distributions, this theoretical paradigm establishes a versatile framework for studying optimization dynamics, successfully characterizing diverse phenomena such as benign overfitting, optimizer convergence, the impact of data augmentation, and weak-to-strong generalization.

Recent studies have applied this framework to standard adversarial training, elucidating mechanisms of robust feature learning (Li & Li, 2025a) and the memorization-induced robust generalization gap (Li & Li, 2025b). However, the theoretical mechanisms of adversarial distillation remain largely unexplored. We bridge this gap by establishing a feature learning framework specifically for AD. Distinct from prior AT-focused analyses, we investigate the failure modes of distillation. We theoretically elucidate how teacher supervision can detrimentally alter the student's feature learning dynamics, providing a rigorous explanation for the counter-intuitive phenomenon where highly robust teachers fail to improve student robustness.

# D. Formal Problem Setup

### D.1. Notations

We summarize the notations used throughout this paper in Table 11.

### D.2. Data and Feature Model

We study a patch-structured binary classification model in which each sample contains one label-dependent signal patch and $P - 1$ label-independent noise patches.

**Definition D.1** (Data Generating Process). This definition formalizes the data model introduced in Definition 4.1. We define the training samples $\{(\mathbf{X}_i, y_i)\}_{i=1}^N$ with $\mathbf{X}_i = [\mathbf{x}_{i,1}; \ldots; \mathbf{x}_{i,P}] \in \mathbb{R}^{P \times d}$ and $y_i \in \{+1, -1\}$ as follows.

1. Fix two orthogonal robust features: the learnable feature $\mathbf{u} := \mathbf{e}_1$ and the unlearnable feature $\mathbf{v} := \mathbf{e}_d$. Let $\mathcal{F} := \mathrm{span}\{\mathbf{u}, \mathbf{v}\}$, and let $\Pi_{\mathcal{F}}$ be the projection matrix onto this subspace. Also fix a partition of the training indices $[N]$ into $\mathcal{S}_L$ and $\mathcal{S}_U$, with $|\mathcal{S}_L| = (1 - p_{\mathrm{un}})N$ and $|\mathcal{S}_U| = p_{\mathrm{un}}N$.

2. For each $i \in [N]$, draw the label $y_i$ uniformly from $\{+1, -1\}$, and draw a signal patch index $s(\mathbf{X}_i)$ uniformly from $[P]$. The signal patch is generated by

$$\mathbf{x}_{i,s(\mathbf{X}_i)} = \begin{cases} \alpha y_i \mathbf{u}, & \text{if } i \in \mathcal{S}_L \quad (\text{learnable}), \\ \alpha y_i \mathbf{v}, & \text{if } i \in \mathcal{S}_U \quad (\text{unlearnable}). \end{cases} \tag{22}$$

*Table 11.* **Summary of notations used in the theoretical analysis.**

| Notation | Description |
|---|---|
| *General Mathematical Notations* | |
| $[N]$ | The set of integers $\{1, 2, \ldots, N\}$. |
| $\|\mathbf{x}\|_2, \|\mathbf{x}\|_\infty$ | Euclidean norm and infinity norm. |
| $\mathbb{E}[\cdot], \Pr[\cdot]$ | Expectation and probability operators. |
| $\tilde{O}(\cdot), \tilde{\Omega}(\cdot), \tilde{\Theta}(\cdot)$ | Asymptotic notations hiding polylogarithmic factors. |
| $\delta, \mathcal{E}$ | Failure probability and global concentration event from Lemma E.1. |
| *Data Model & Features* | |
| $N, P, d$ | Number of training samples, number of patches, and dimension of each patch. |
| $\mathbf{X} \in \mathbb{R}^{P \times d}, \mathbf{x}_{i,p}$ | Input with $P$ patches, and the $p$-th patch of sample $i$. |
| $y \in \{+1, -1\}, s(\mathbf{X}) \in [P]$ | Binary label and signal-patch index. |
| $\mathcal{S}_L, \mathcal{S}_U$ | Training index sets of learnable and unlearnable samples. |
| $p_{\text{un}}$ | Ratio of unlearnable samples in the training dataset, $|\mathcal{S}_U|/N$. |
| $\mathbf{u}, \mathbf{v}$ | Learnable robust feature ($\mathbf{e}_1$) and unlearnable robust feature ($\mathbf{e}_d$). |
| $\alpha, \sigma_n$ | Robust signal strength and noise standard deviation. |
| $\mathcal{D}_L, \mathcal{L}^{\text{rob}}_{\mathcal{D}_L}$ | Learnable-sample test distribution and its robust test error. |
| *Neural Networks & Training* | |
| $f_\mathbf{W}, f_{\mathbf{W}_{\text{T}}}$ | Student model and generic teacher model. |
| $f_{\mathbf{W}_{\text{G}}}, f_{\mathbf{W}_{\text{B}}}$ | Good Teacher and Bad Teacher. |
| $\mathbf{W}, \mathbf{w}_r$ | Student parameters and the weight vector of the $r$-th filter. |
| $m, \sigma_0, \eta, \epsilon$ | Number of filters, initialization scale, learning rate, and adversarial perturbation budget. |
| $\tilde{\mathbf{X}}^{(t)}, \tilde{\mathbf{X}}_i^{(t)}$ | Adversarial example generated at iteration $t$. |
| $\rho_{i,j,r}^{(t)}$ | Noise coefficient for sample $i$, patch $j$, and filter $r$ at iteration $t$. |
| $T, T_0, T_1, T_{\max}$ | Training horizon, signal learning time, noise memorization time, and Good Teacher horizon. |
| $\Gamma$ | Teacher margin parameter. |
| $\phi(z), \ell(z)$ | Activation function $\phi(z) = (\max\{0, z\})^3$ and logistic loss $\ell(z) = \log(1 + e^{-z})$. |
| $\sigma(z), \psi(z)$ | Logistic sigmoid $\sigma(z) = (1 + e^{-z})^{-1}$ and negative sigmoid $\psi(z) = \sigma(-z)$. |
| $\mathcal{L}_{\text{AT}}, \mathcal{L}_{\text{AD}}$ | Adversarial training loss and adversarial distillation loss. |

3. For each non-signal patch $p \neq s(\mathbf{X}_i)$, draw independent Gaussian noise orthogonal to the feature subspace:

$$\mathbf{x}_{i,p} \sim \mathcal{N}\left(\mathbf{0}, \sigma_n^2(\mathbf{I}_d - \Pi_\mathcal{F})\right). \tag{23}$$

We analyze two regimes of the unlearnable ratio $p_{\text{un}}$: (i) the learnable-only regime $p_{\text{un}} = 0$ (so $\mathcal{S}_U = \emptyset$), and (ii) the sparse-unlearnable regime $CN^{-1} \leq p_{\text{un}} \leq C^{-1}N^{-1} \log d$ (so $C \leq |\mathcal{S}_U| \leq C^{-1} \log d$). These two regimes lead to different adversarial training dynamics, which are treated separately in our theorems.

For test-time evaluation, we consider the learnable-sample distribution $\mathcal{D}_L$, defined by the same generation rule with the learnable signal patch $\alpha y \mathbf{u}$ and independent Gaussian noise patches. We focus on this distribution because unlearnable test samples contain label information only in $\mathbf{v}$, which is orthogonal to all student weights, making their robust error trivially high.

### D.3. Network Architecture

We next specify the student architecture and the structural constraint that makes the unlearnable feature inaccessible to the student.

**Definition D.2** (Student Model). This definition formalizes the student model introduced in Definition 4.2. We define the student model $f_\mathbf{W}$ as a two-layer neural network with $m$ filters. To facilitate theoretical analysis, the network is modeled with anti-symmetric weights, utilizing a cubic activation function $\phi(z) := (\max\{0, z\})^3$. The final network output $f_\mathbf{W}(\mathbf{X})$

aggregates the response across all patches:

$$f_{\mathbf{W}}(\mathbf{X}) = \sum_{r=1}^{m} \sum_{p=1}^{P} \left[ \phi(\langle \mathbf{w}_r, \mathbf{x}_p \rangle) - \phi(-\langle \mathbf{w}_r, \mathbf{x}_p \rangle) \right]. \tag{24}$$

Here, $\mathbf{W} = \{\mathbf{w}_r\}_{r=1}^{m}$ denotes the set of student weight vectors.

To formalize the limited capacity of the student, we enforce the weight space to be orthogonal to the unlearnable feature $\mathbf{v}$ at initialization and throughout training:

$$\langle \mathbf{w}_r, \mathbf{v} \rangle = 0, \qquad \forall r \in [m]. \tag{25}$$

Given our data model where $\mathbf{v} = \mathbf{e}_d$, this constraint is equivalent to fixing the $d$-th coordinate of all weights to zero, i.e., $w_{r,d} = 0$. The remaining coordinates are initialized i.i.d. from a Gaussian distribution, $w_{r,j} \sim \mathcal{N}(0, \sigma_0^2)$ for $j \in [d-1]$.

### D.4. Teacher Configurations and Properties

For the analysis of adversarial distillation, we consider two fixed robust teacher models: a Good Teacher $f_{\mathbf{W}_{\mathrm{G}}}$ and a Bad Teacher $f_{\mathbf{W}_{\mathrm{B}}}$. When a statement applies to either teacher, we write $f_{\mathbf{W}_{\mathrm{T}}}$ for a generic teacher.

**Definition D.3** (Teacher Configurations). This definition formalizes the teacher configurations introduced in Definition 4.3. We define the teacher configurations as follows:

1. Both teachers generalize well on learnable samples. In particular, for any $i \in \mathcal{S}_L$, the generic teacher produces a large logit in the direction of the target label:

$$y_i f_{\mathbf{W}_{\mathrm{T}}}(\mathbf{X}_i) \geq \Gamma. \tag{26}$$

   Additionally, both teachers have weights orthogonal to noise patches. Their distinction lies only in their behavior on the unlearnable feature $\mathbf{v}$.

2. **Good Teacher** $(f_{\mathbf{W}_{\mathrm{G}}})$. This teacher relies only on the learnable feature $\mathbf{u}$ and is orthogonal to the unlearnable feature $\mathbf{v}$. Consequently, for any unlearnable training sample indexed by $i \in \mathcal{S}_U$,

$$y_i f_{\mathbf{W}_{\mathrm{G}}}(\mathbf{X}_i) = 0. \tag{27}$$

3. **Bad Teacher** $(f_{\mathbf{W}_{\mathrm{B}}})$. This teacher is aligned with the unlearnable feature $\mathbf{v}$. Consequently, for any unlearnable training sample indexed by $i \in \mathcal{S}_U$,

$$y_i f_{\mathbf{W}_{\mathrm{B}}}(\mathbf{X}_i) \geq \Gamma. \tag{28}$$

The terms Good Teacher and Bad Teacher refer to their compatibility with the capacity-constrained student. Both teachers are robust models. The Bad Teacher is called "bad" only because its confident supervision on the unlearnable feature $\mathbf{v}$ cannot be represented by the student and therefore drives the student toward noise memorization.

### D.5. Training Objectives and Optimization

We consider a binary classifier $f_{\mathbf{W}}(\mathbf{X})$ trained using the logistic loss $\ell(z) := \log(1 + e^{-z})$. We analyze two training paradigms: standard adversarial training (AT) and adversarial distillation (AD).

**Definition D.4** (Training Objectives and Optimization). This definition formalizes the adversarial data generation in Definition 4.4 and the training objectives and optimization dynamics in Definition 4.5. We define the adversarial examples, training objectives, and gradient-descent dynamics as follows.

1. Following Li & Li (2025b), for a sample $(\mathbf{X}, y)$, we define the training adversarial example $\tilde{\mathbf{X}}$ as the solution to

$$\tilde{\mathbf{X}} = \operatorname*{argmax}_{\mathbf{X}'} \ell\big(y f_{\mathbf{W}}(\mathbf{X}')\big) \quad \text{s.t.} \quad \|\mathbf{X}' - \mathbf{X}\|_{\infty} \leq \epsilon, \ \mathbf{X}' - \mathbf{X} \in \operatorname{span}\big(\mathbf{x}_{s(\mathbf{X})}\big). \tag{29}$$

   Thus, the adversary perturbs only the signal patch along its original direction. In particular, for a learnable sample with signal patch $\mathbf{x}_{s(\mathbf{X})} = y\alpha\mathbf{u}$, the adversarially perturbed signal patch takes the form

$$\tilde{\mathbf{x}}_{s(\mathbf{X})} = y(\alpha - \epsilon)\mathbf{u}. \tag{30}$$

2. Standard AT minimizes the logistic loss on the generated adversarial example:

$$\mathcal{L}_{\mathrm{AT}}(\mathbf{W}; \mathbf{X}, y) = \ell\big(y\, f_{\mathbf{W}}(\tilde{\mathbf{X}})\big). \tag{31}$$

3. In AD, we leverage a fixed robust teacher network $f_{\mathbf{W}_{\mathrm{T}}}$ to provide soft targets for the student. Following the standard AD setup (e.g., RSLAD (Zi et al., 2021)), the student matches the teacher's output distribution. In the binary margin-based formulation, this yields the weighted logistic loss:

$$\mathcal{L}_{\mathrm{AD}}(\mathbf{W}; \mathbf{X}, y) = \sigma\big(yf_{\mathbf{W}_{\mathrm{T}}}(\mathbf{X})\big)\, \ell\big(yf_{\mathbf{W}}(\tilde{\mathbf{X}})\big) + \sigma\big(-yf_{\mathbf{W}_{\mathrm{T}}}(\mathbf{X})\big)\, \ell\big(-yf_{\mathbf{W}}(\tilde{\mathbf{X}})\big). \tag{32}$$

4. We optimize the empirical risk by gradient descent. At each iteration $t$, the adversarial examples $\{\tilde{\mathbf{X}}_i^{(t)}\}_{i=1}^N$ are generated using the current model $\mathbf{W}^{(t)}$, and the parameters are updated by

$$\mathbf{W}^{(t+1)} = \mathbf{W}^{(t)} - \frac{\eta}{N} \sum_{i\in[N]} \nabla_{\mathbf{W}} \mathcal{L}\big(\mathbf{W}^{(t)}; \mathbf{X}_i, y_i\big), \tag{33}$$

where $\mathcal{L} \in \{\mathcal{L}_{\mathrm{AT}}, \mathcal{L}_{\mathrm{AD}}\}$ and $\eta > 0$ is the learning rate. We analyze the student after $T$ iterations.

### D.6. Parameter and Regime Conditions

**Condition D.5.** There exists a sufficiently large universal constant $C > 0$ such that the following parameter and regime conditions hold:

(C1) Training horizon $T$ is sufficiently large to cover the signal-learning and noise-memorization time scales:

$$T \geq \frac{CN}{\eta\sigma_0\sigma_n^3 d^{3/2}}.$$

(C2) Data dimension is sufficiently large:

$$d \geq Cm^2 P^2 N^2 \log^4\big(TNmP\delta^{-1}\big).$$

(C3) Signal strength is sufficiently large:

$$\alpha \geq \frac{C\sigma_n\sqrt{d}\,\log\big(TNmP\delta^{-1}\big)}{N^{1/3}(1-p_{\mathrm{un}})^{1/3}}.$$

(C4) Initialization scale is sufficiently small:

$$\sigma_0 \leq C^{-1}\min\left\{\frac{1}{m^{2/3}P^{2/3}\sigma_n\sqrt{d}}, \frac{1}{\alpha m^{2/3}}, \frac{1}{\sigma_n m^2 P^{1/2}d}\right\}\frac{1}{\log\big(TdNmP\delta^{-1}\big)}.$$

(C5) Learning rate is sufficiently small:

$$\eta \leq C^{-1}\min\left\{\frac{1}{\alpha^3\sigma_0}, \frac{1}{\sigma_n^2 d}, \frac{1}{\alpha^2}\right\}\frac{1}{\log\big(TNmP\delta^{-1}\big)}.$$

(C6) Adversarial budget lies in the required regime:

$$C\,\sigma_n m P^{1/2}\log\big(TdNP\delta^{-1}\big) \leq \epsilon < C^{-1}\min\left\{\alpha, \frac{1}{m\sigma_0 d}\right\}.$$

(C7) In the sparse-unlearnable regime, unlearnable samples are sufficiently sparse but nontrivial:

$$\frac{C}{N} \leq p_{\mathrm{un}} \leq \frac{C^{-1}\log d}{N}, \qquad N \geq C\,mP\log\big(TdNmP\delta^{-1}\big).$$

(C8) The model width is controlled at a polylogarithmic scale, and the number of patches is bounded:

$$C \log \left( N P \delta^{-1} \right) \le m \le C \log^C d, \qquad 2 \le P \le C.$$

(C9) Teacher margin is sufficiently large to be in the saturated regime:

$$\Gamma \ge Cd.$$

*Remark* D.6 (Interpretation of the parameter conditions). The conditions in Condition D.5 are sufficient conditions under which the training dynamics can be cleanly separated into signal learning and noise memorization. The logarithmic terms are technical factors needed to ensure that all high-probability bounds hold uniformly over $N$ samples, $P$ patches, $m$ filters, and $T$ iterations. They arise from applying tail bounds and taking union bounds over these indices. Thus, the main constraints should be read from the polynomial dependences on $N, d, m, P, \alpha, \sigma_0, \sigma_n, \epsilon$, and $\eta$, rather than from the exact logarithmic powers.

Conditions (C1) and (C5) control the time scale of gradient descent. The lower bound on $T$ ensures that the training horizon is long enough to contain both the signal-learning phase and the noise-memorization phase. The upper bound on $\eta$ prevents one-step updates from overshooting the signal and noise scales tracked in the proof. Thus, each gradient step remains small relative to the quantities being controlled, which allows the discrete-time induction arguments to close.

Conditions (C2) and (C8) allow us to apply concentration inequalities uniformly over samples, patches, and filters. They ensure that the training data and the initial model parameters satisfy the desirable high-probability properties, such as controlled noise norms, small noise correlations, and bounded initialization-dependent quantities.

Conditions (C3), (C4), and (C6) fix the separation between signal, noise, initialization, and adversarial perturbation. The lower bound on $\alpha$ makes the robust signal strength large enough compared to the random noise scale, so that signal-driven updates can be separated from the contributions of noise patches. The upper bound on $\sigma_0$ ensures that the dynamics are not already determined by initialization-scale correlations. The condition on $\epsilon$ places adversarial perturbations in a nontrivial regime: they are large enough relative to the noise scale, but not large enough to erase the signal. Informally, this corresponds to the hierarchy $\sigma_n < \epsilon < \alpha$.

Condition (C7) formalizes the sparse-unlearnable regime in Section D.2; in the learnable-only regime it is replaced by $p_{\mathrm{un}} = 0$. Since $p_{\mathrm{un}}$ scales inversely with $N$, the accompanying lower bound on $N$ ensures that the sample size is large enough relative to the model and patch complexity. Finally, condition (C9) places the teacher in the saturated regime on the samples where it is confident. This makes the teacher-induced gradient match the corresponding hard-label gradient up to exponentially small terms, allowing us to compare adversarial distillation with adversarial training.

# E. Proof Preliminaries

In this section, we establish the probabilistic preliminaries used throughout the theoretical analysis and outline the organization of the subsequent proofs.

## E.1. Properties of Data Sampling

To decouple the randomness of data generation and network initialization from the deterministic optimization dynamics, we define a global high-probability event $\mathcal{E}$ where all necessary concentration bounds hold simultaneously.

**Lemma E.1** (Definition and Probability of Concentration Event $\mathcal{E}$). Suppose $\delta \in (0, 1)$. Under condition (C2) and condition (C8), which ensure the sufficient conditions $d \ge C \log(NmP/\delta)$ and $m \ge C \log(NP/\delta)$ are satisfied for a large constant $C > 0$, let $\mathcal{E}$ denote the event that, for all $r \in [m]$, $i, k \in [N]$, and noise patches $j \ne s(\mathbf{X}_i)$ and $q \ne s(\mathbf{X}_k)$ with $(i, j) \ne (k, q)$, the following hold:

1. Uniform bounds on noise patches:

$$\frac{1}{2}\sigma_n^2 d \le \|\mathbf{x}_{i,j}\|_2^2 \le \frac{3}{2}\sigma_n^2 d. \tag{P1}$$

$$|\langle \mathbf{x}_{i,j}, \mathbf{x}_{k,q} \rangle| \le 2\sigma_n^2 \sqrt{d \log\left(\frac{16N^2P^2}{\delta}\right)}. \tag{P2}$$

$$\|\mathbf{x}_{i,j}\|_\infty \leq \sigma_n \sqrt{2 \log\left(\frac{16dNP}{\delta}\right)}. \tag{P3}$$

2. Uniform bounds on initialization:

$$\|\mathbf{w}_r^{(0)}\|_2 \leq 2\sigma_0\sqrt{d}. \tag{P4}$$

$$\left|\langle \mathbf{w}_r^{(0)}, \mathbf{e}_1 \rangle\right| \leq \sigma_0 \sqrt{2 \log\left(\frac{16m}{\delta}\right)}. \tag{P5}$$

$$\left|\langle \mathbf{w}_r^{(0)}, \mathbf{x}_{i,j} \rangle\right| \leq 2\sigma_0\sigma_n \sqrt{d \log\left(\frac{16NmP}{\delta}\right)}. \tag{P6}$$

3. Maximum initialization margins:

$$\frac{1}{2}\sigma_0 \leq \max_{r \in [m]} \langle \mathbf{w}_r^{(0)}, \mathbf{e}_1 \rangle \leq \sigma_0 \sqrt{2 \log\left(\frac{16m}{\delta}\right)}. \tag{P7}$$

$$\frac{1}{4}\sigma_0\sigma_n\sqrt{d} \leq \max_{r \in [m]} y_i \langle \mathbf{w}_r^{(0)}, \mathbf{x}_{i,j} \rangle \leq 2\sigma_0\sigma_n \sqrt{d \log\left(\frac{16NmP}{\delta}\right)}. \tag{P8}$$

Then, the event $\mathcal{E}$ occurs with probability at least $1 - \delta$.

*Proof of Lemma E.1.* We prove that (P1)–(P6) and the lower bounds in (P7) and (P8) each fail with probability at most $\delta/8$. The upper bounds in (P7) and (P8) are exactly (P5) and (P6), respectively, so the claim then follows by a union bound. We prove the statements one by one, marking the completion of each proof with a ■.

For (P1), fix $i \in [N]$ and $j \neq s(\mathbf{X}_i)$. Since $\mathbf{x}_{i,j}$ is Gaussian in the $(d-2)$-dimensional subspace orthogonal to $\text{span}\{\mathbf{e}_1, \mathbf{e}_d\}$, there exist i.i.d. standard Gaussian random variables $g_1, \ldots, g_{d-2}$ such that

$$\|\mathbf{x}_{i,j}\|_2^2 = \sigma_n^2 \sum_{\ell=1}^{d-2} g_\ell^2. \tag{34}$$

Applying Lemma I.4 with $n = d - 2$ and $\gamma = d/4$, we obtain

$$\Pr\left(\left|\sum_{\ell=1}^{d-2} g_\ell^2 - (d-2)\right| \geq \frac{d}{4}\right) \leq 2\exp(-cd) \leq \frac{\delta}{16NP}, \tag{35}$$

where the last inequality holds under $d \geq C \log(NmP/\delta)$. Hence, with probability at least $1 - \delta/(8NP)$,

$$\left|\|\mathbf{x}_{i,j}\|_2^2 - \sigma_n^2(d-2)\right| \leq \frac{1}{4}\sigma_n^2 d. \tag{36}$$

Since $d$ is sufficiently large, this implies (P1) for the fixed pair $(i, j)$. Taking a union bound over at most $NP$ noise patches, we conclude that (P1) holds for all $i \in [N]$ and $j \neq s(\mathbf{X}_i)$ with probability at least $1 - \delta/8$. ■

For (P2), fix distinct pairs $(i, j) \neq (k, q)$ with $j \neq s(\mathbf{X}_i)$ and $q \neq s(\mathbf{X}_k)$. By independence of noise patches, there exist two independent collections of i.i.d. standard Gaussian random variables $g_1, \ldots, g_{d-2}$ and $z_1, \ldots, z_{d-2}$ such that

$$\langle \mathbf{x}_{i,j}, \mathbf{x}_{k,q} \rangle = \sigma_n^2 \sum_{\ell=1}^{d-2} g_\ell z_\ell. \tag{37}$$

Applying Lemma I.5 with $n = d - 2$ and $\gamma = 2\sqrt{d \log\left(\frac{16N^2P^2}{\delta}\right)}$, we obtain

$$\Pr\left(|\langle \mathbf{x}_{i,j}, \mathbf{x}_{k,q}\rangle| \geq 2\sigma_n^2\sqrt{d \log\left(\frac{16N^2P^2}{\delta}\right)}\right) \leq 2\exp\left(-\frac{1}{4}\min\left\{\frac{4d\log\left(\frac{16N^2P^2}{\delta}\right)}{d-2}, \ 2\sqrt{d\log\left(\frac{16N^2P^2}{\delta}\right)}\right\}\right)$$

$$\leq 2\exp\left(-\log\left(\frac{16N^2P^2}{\delta}\right)\right)$$

$$= \frac{\delta}{8N^2P^2},$$

(38)

where the second inequality holds under $d \geq C\log(NmP/\delta)$ for a sufficiently large constant $C$. Thus, (P2) holds for the fixed pair $(i,j) \neq (k,q)$ with probability at least $1 - \delta/(8N^2P^2)$. Taking a union bound over at most $N^2P^2$ ordered pairs of noise patches, we conclude that (P2) holds uniformly for all distinct pairs $(i,j) \neq (k,q)$ with $j \neq s(\mathbf{X}_i)$ and $q \neq s(\mathbf{X}_k)$ with probability at least $1 - \delta/8$. ∎

For (P3), fix $i \in [N]$, $j \neq s(\mathbf{X}_i)$, and $\ell \in [d]$. Each coordinate of each noise patch is a centered Gaussian random variable with variance at most $\sigma_n^2$. Hence, by Lemma I.3,

$$\Pr\left(|(\mathbf{x}_{i,j})_\ell| \geq \sigma_n\sqrt{2\log\left(\frac{16dNP}{\delta}\right)}\right) \leq \frac{\delta}{8dNP}$$

(39)

Thus, the desired coordinate bound holds for this fixed triple $(i,j,\ell)$ with probability at least $1 - \delta/(8dNP)$. Taking a union bound over at most $dNP$ such triples, we conclude that (P3) holds with probability at least $1 - \delta/8$. ∎

For (P4), fix $r \in [m]$. Since $\mathbf{w}_r^{(0)}$ is Gaussian in the $(d-1)$-dimensional subspace orthogonal to $\mathbf{e}_d$, there exist i.i.d. standard Gaussian random variables $g_1, \ldots, g_{d-1}$ such that

$$\|\mathbf{w}_r^{(0)}\|_2^2 = \sigma_0^2 \sum_{\ell=1}^{d-1} g_\ell^2.$$

(40)

Applying Lemma I.4 with $n = d - 1$ and $\gamma = d/2$, we obtain

$$\Pr\left(\left|\sum_{\ell=1}^{d-1} g_\ell^2 - (d-1)\right| \geq \frac{d}{2}\right) \leq 2\exp(-cd) \leq \frac{\delta}{8m},$$

(41)

where the last inequality holds under $d \geq C\log(NmP/\delta)$. Hence, with probability at least $1 - \delta/(8m)$,

$$\left|\|\mathbf{w}_r^{(0)}\|_2^2 - \sigma_0^2(d-1)\right| \leq \frac{1}{2}\sigma_0^2 d.$$

(42)

Since $d$ is sufficiently large, this implies (P4) for the fixed filter $r$. Taking a union bound over $r \in [m]$, we conclude that (P4) holds for all $r \in [m]$ with probability at least $1 - \delta/8$. ∎

For (P5), fix $r \in [m]$. Since $\langle \mathbf{w}_r^{(0)}, \mathbf{e}_1\rangle \sim \mathcal{N}(0, \sigma_0^2)$, Lemma I.3 gives

$$\Pr\left(\left|\langle \mathbf{w}_r^{(0)}, \mathbf{e}_1\rangle\right| \geq \sigma_0\sqrt{2\log\left(\frac{16m}{\delta}\right)}\right) \leq \frac{\delta}{8m}.$$

(43)

Thus, (P5) holds for the fixed filter $r$ with probability at least $1 - \delta/(8m)$. Taking a union bound over $r \in [m]$, we conclude that (P5) holds for all $r \in [m]$ with probability at least $1 - \delta/8$. ∎

For (P6), fix $r \in [m]$, $i \in [N]$, and $j \neq s(\mathbf{X}_i)$. Since $\mathbf{x}_{i,j}$ lies in the $(d-2)$-dimensional subspace orthogonal to $\mathrm{span}\{\mathbf{e}_1, \mathbf{e}_d\}$, only the corresponding $(d-2)$ coordinates of $\mathbf{w}_r^{(0)}$ contribute to the inner product. Hence there exist two independent collections of i.i.d. standard Gaussian random variables $g_1, \ldots, g_{d-2}$ and $z_1, \ldots, z_{d-2}$ such that

$$\langle \mathbf{w}_r^{(0)}, \mathbf{x}_{i,j}\rangle = \sigma_0\sigma_n \sum_{\ell=1}^{d-2} g_\ell z_\ell.$$

(44)

Applying Lemma I.5 with $n = d - 2$ and $\gamma = 2\sqrt{d \log\left(\frac{16NmP}{\delta}\right)}$, we obtain

$$\Pr\left(\left|\langle \mathbf{w}_r^{(0)}, \mathbf{x}_{i,j}\rangle\right| \geq 2\sigma_0 \sigma_n \sqrt{d \log\left(\frac{16NmP}{\delta}\right)}\right) \leq 2\exp\left(-\frac{1}{4}\min\left\{\frac{4d\log\left(\frac{16NmP}{\delta}\right)}{d-2}, \ 2\sqrt{d\log\left(\frac{16NmP}{\delta}\right)}\right\}\right)$$

$$\leq 2\exp\left(-\log\left(\frac{16NmP}{\delta}\right)\right)$$

$$= \frac{\delta}{8NmP},$$

(45)

where the second inequality holds under $d \geq C\log(NmP/\delta)$ for a sufficiently large constant $C$. Thus, (P6) holds for the fixed triple $(r, i, j)$ with probability at least $1 - \delta/(8NmP)$. Taking a union bound over at most $NmP$ such triples, we conclude that (P6) holds uniformly for all $r \in [m]$, $i \in [N]$, and $j \neq s(\mathbf{X}_i)$ with probability at least $1 - \delta/8$. ∎

For (P7), the random variables

$$\langle \mathbf{w}_1^{(0)}, \mathbf{e}_1\rangle, \ldots, \langle \mathbf{w}_m^{(0)}, \mathbf{e}_1\rangle$$

(46)

are i.i.d. $\mathcal{N}(0, \sigma_0^2)$. Therefore, applying Lemma I.6 with $n = m$ and $\sigma = \sigma_0$, we obtain

$$\Pr\left(\max_{r\in[m]}\langle \mathbf{w}_r^{(0)}, \mathbf{e}_1\rangle < \frac{1}{2}\sigma_0\right) \leq e^{-cm} \leq \frac{\delta}{8},$$

(47)

where the last inequality holds under $m \geq C\log(NP/\delta)$ for a sufficiently large constant $C$. Thus, the lower bound in (P7) holds with probability at least $1 - \delta/8$. The upper bound in (P7) is already guaranteed by (P5), so no additional probability loss is needed. ∎

For (P8), fix $i \in [N]$ and $j \neq s(\mathbf{X}_i)$. Conditional on $\mathbf{x}_{i,j}$, the random variables

$$y_i\langle \mathbf{w}_1^{(0)}, \mathbf{x}_{i,j}\rangle, \ldots, y_i\langle \mathbf{w}_m^{(0)}, \mathbf{x}_{i,j}\rangle$$

(48)

are i.i.d. $\mathcal{N}(0, \sigma_0^2\|\mathbf{x}_{i,j}\|_2^2)$. On the event that (P1) holds,

$$\|\mathbf{x}_{i,j}\|_2 \geq \sigma_n\sqrt{\frac{d}{2}},$$

(49)

and therefore

$$\frac{1}{4}\sigma_0\sigma_n\sqrt{d} \leq \frac{1}{2}\sigma_0\|\mathbf{x}_{i,j}\|_2.$$

(50)

Hence, still conditioning on $\mathbf{x}_{i,j}$ and applying Lemma I.6 with $n = m$ and $\sigma = \sigma_0\|\mathbf{x}_{i,j}\|_2$, we obtain

$$\Pr\left(\max_{r\in[m]} y_i\langle \mathbf{w}_r^{(0)}, \mathbf{x}_{i,j}\rangle < \frac{1}{4}\sigma_0\sigma_n\sqrt{d} \ \middle| \ \mathbf{x}_{i,j}\right) \leq e^{-cm}$$

(51)

whenever (P1) holds. Therefore,

$$\Pr\left(\max_{r\in[m]} y_i\langle \mathbf{w}_r^{(0)}, \mathbf{x}_{i,j}\rangle < \frac{1}{4}\sigma_0\sigma_n\sqrt{d}\right) \leq e^{-cm} + \Pr\left(\|\mathbf{x}_{i,j}\|_2 < \sigma_n\sqrt{\frac{d}{2}}\right)$$

$$\leq e^{-cm} + \frac{\delta}{16NP} \leq \frac{\delta}{8NP},$$

(52)

where the second inequality follows from (35), and the last inequality holds under $m \geq C\log(NP/\delta)$ for a sufficiently large constant $C$. Thus, the lower bound in (P8) holds for the fixed pair $(i, j)$ with probability at least $1 - \delta/(8NP)$. Taking a union bound over at most $NP$ such pairs, we conclude that the lower bound in (P8) holds uniformly with probability at least $1 - \delta/8$. The upper bound in (P8) is already guaranteed by (P6), so no additional probability loss is needed. ∎

Each of the eight required bounds has failure probability at most $\delta/8$. Applying a union bound over (P1)–(P6) and the lower bounds in (P7) and (P8) shows that $\mathcal{E}$ holds with probability at least $1 - \delta$, completing the proof. □

*Table 12.* **Hierarchy of lemmas and dependencies.** All results are stated on the event $\mathcal{E}$ from Lemma E.1 and under the parameter and regime conditions in Condition D.5; these common assumptions are omitted from the dependencies column.

| Lemma/Theorem | Description | Dependencies |
| --- | --- | --- |
| *Section F: Learning Dynamics of Adversarial Training* | | |
| Lemma F.1 | Signal weight update bounds | - |
| Lemma F.2 | Noise coefficient update rule | - |
| Hypothesis F.3 | Noise stability on learnable samples | - |
| Lemma F.4 | Gradient bound on learnable samples | Hypothesis F.3 |
| Lemma F.5 | Signal learning time | Hypothesis F.3 and Lemmas F.1 and F.4 |
| Lemma F.6 | Early-phase noise coefficient bound | Lemmas F.2 and F.5 |
| Lemma F.7 | Signal weight bound | Hypothesis F.3 and Lemma F.1 |
| Lemma F.8 | Gradient decay on learnable samples | Hypothesis F.3 and Lemmas F.1, F.5 and F.7 |
| Lemma F.9 | Uniform noise coefficient bound | Lemma F.2 |
| Lemma F.10 | Verification of noise stability on learnable samples | Lemmas F.2, F.6, F.8 and F.9 |
| *Section G: The Dichotomy of Adversarial Training* | | |
| Lemma G.1 | Noise stability without unlearnable samples | Lemmas F.2, F.6 and F.8 |
| Theorem G.2 | AT without unlearnable samples | Lemmas F.1, F.5, F.8 and G.1 |
| Lemma G.3 | Gradient bound on unlearnable samples | Lemmas F.2 and F.10 |
| Lemma G.4 | Dynamics of the maximum shifted noise coefficient | Lemmas F.2, F.10 and G.3 |
| Lemma G.5 | Noise memorization time | Lemma G.4 |
| Theorem G.6 | AT with unlearnable samples | Lemmas F.5, F.7, F.8, F.9 and G.5 |
| *Section H: The Dichotomy of Adversarial Distillation* | | |
| Lemma H.1 | Gradient approximation on learnable samples | - |
| Lemma H.2 | Gradient approximation under a bad teacher | - |
| Theorem H.3 | AD under a bad teacher | Theorem G.6 and Lemmas H.1 and H.2 |
| Hypothesis H.4 | Noise stability under a good teacher | - |
| Lemma H.5 | Inner product bound under a good teacher | Lemma F.2 and Hypothesis H.4 |
| Lemma H.6 | Inherited AT estimates under a good teacher | Lemmas F.4, F.5, F.6, F.7, F.8, H.1 and H.5 |
| Lemma H.7 | Verification of noise stability under a good teacher | Lemmas F.2, H.5 and H.6 |
| Theorem H.8 | AD under a good teacher | Lemmas H.5, H.6 and H.7 |

### E.2. Proof Overview

Before entering the detailed dynamics, we summarize the mechanism behind the proof. Under Condition D.5, we first condition on the event $\mathcal{E}$, which holds with probability at least $1 - \delta$ by Lemma E.1. Here $\delta$ appears in the conditions only through logarithmic factors, so it can be chosen polynomially small in $d$; this is the high-probability sense used in the main text. On this event, the random noise patches have controlled norms and small pairwise correlations, while the initialized weights have uniformly bounded correlations and favorable maximum alignments with the signal and noise directions. From this point on, the proof is essentially deterministic: we compare the signal and noise updates over time. The proof first analyzes how AT learns the signal and noise components, and then uses this decomposition to study how teacher supervision changes the same dynamics in AD. Table 12 summarizes the formal dependency structure of the lemmas.

**Why the learnable signal grows first.** The first key step is to show that, in the early stage of training, the learnable feature $\mathbf{u}$ is amplified before the noise components become large. For learnable samples, the adversarial perturbation changes the signal patch from $\alpha y \mathbf{u}$ to $(\alpha - \epsilon) y \mathbf{u}$, so a positive learnable signal remains. In the early stage, the corresponding margin is still small, and hence the loss gradient is not yet saturated; this yields a persistent positive update along $\mathbf{u}$ (Lemmas F.1 and F.4). At the same time, on the event $\mathcal{E}$, the noise patches are nearly orthogonal to one another, so their contributions to the learnable-sample logits remain controlled (Hypothesis F.3 and Lemma F.10). This yields the signal-learning time $T_0$, at which at least one filter satisfies $w_{r,1}^{(T_0)} \geq \tilde{\Omega}(\alpha^{-1})$ (Lemma F.5). Once the signal coordinate reaches this scale, the margins of learnable samples increase, so their loss gradients become small. More precisely, over the remaining training horizon, the cumulative gradient contribution of each learnable sample is controlled (Lemma F.8). Consequently, after the signal

is learned, learnable samples no longer drive substantial updates, and the remaining dynamics are determined by how the training procedure handles unlearnable samples and noise components.

**Why standard adversarial training overfits only with unlearnable samples.** The dichotomy for adversarial training is determined by whether any substantial gradient remains after the learnable signal has been acquired. If $p_{\text{un}} = 0$, every training sample is learnable. Hence, once $\mathbf{u}$ is learned, the learnable-sample gradients become small, and the noise responses stay near their initialization scale (Lemma G.1). This yields robust generalization (Theorem G.2). In contrast, when sparse unlearnable samples are present, the student cannot represent their robust feature $\mathbf{v}$. Thus, even after $\mathbf{u}$ is learned, these samples continue to produce non-negligible loss gradients (Lemma G.3). Under standard adversarial training, the resulting updates have a persistent one-sided effect on the sample-specific noise coefficients (Lemma G.4). Over the longer timescale $T_1$, this one-sided growth produces a memorized noise response of constant order on at least one unlearnable sample (Lemma G.5). This memorized noise direction can then be exploited by a test-time adversary, leading to robust overfitting (Theorem G.6).

**How the teacher determines the distillation dynamics.** The adversarial distillation dynamics are governed by how the teacher's soft targets affect the student gradients on learnable and unlearnable samples. On learnable samples, both teachers are saturated in the correct direction, so the AD gradient differs from the AT gradient only by an exponentially small term. Thus, the same signal-learning mechanism remains valid (Lemmas H.1 and H.6). The key distinction appears on unlearnable samples. A Bad Teacher is confident on the unlearnable feature $\mathbf{v}$, so its soft target is effectively the hard label. Consequently, the AD update behaves like the AT update and produces the same one-sided noise growth (Lemma H.2 and Theorem H.3). A Good Teacher, in contrast, has zero margin on unlearnable samples. Its soft target is therefore exactly balanced between the two labels, so the unlearnable-sample update no longer creates a persistent one-sided drift. The resulting noise responses remain controlled over the horizon $T \leq T_{\max}$ (Hypothesis H.4 and Lemma H.7). Thus, the Good Teacher keeps the signal-learning part of AD intact while preventing unlearnable samples from driving noise memorization, yielding low robust training and test error (Theorem H.8).

## F. Learning Dynamics of Adversarial Training

In this section, we analyze the dynamics of adversarial training by tracking the evolution of the signal and noise coordinates. We first derive the local update rules for the signal weights and the noise coefficients. We then show that the learnable signal is amplified in the early stage of training, while the noise coefficients remain controlled. After the signal has reached the target scale, we prove that the signal weights stay bounded and that the gradients on learnable samples decay over time. The time at which the signal reaches this scale is denoted by $T_0$. These estimates are used at the end of the section to verify the noise stability hypothesis on learnable samples.

### F.1. Local Dynamics

We begin by deriving the one-step update formulas that will be used throughout the analysis.

**Lemma F.1** (Signal Weight Update Bounds)**.** For each $r \in [m]$ and any $t \geq 0$, the signal weights $w_{r,1}^{(t)}$ are monotonically non-decreasing and satisfy

$$w_{r,1}^{(t+1)} - w_{r,1}^{(t)} \geq \frac{3\eta(\alpha - \epsilon)^3}{N} \left(w_{r,1}^{(t)}\right)^2 \sum_{i \in \mathcal{S}_L} \psi\left(y_i f_{\mathbf{W}^{(t)}}(\tilde{\mathbf{X}}_i^{(t)})\right), \tag{53}$$

$$w_{r,1}^{(t+1)} - w_{r,1}^{(t)} \leq \frac{3\eta\alpha^3}{N} \left(w_{r,1}^{(t)}\right)^2 \sum_{i \in \mathcal{S}_L} \psi\left(y_i f_{\mathbf{W}^{(t)}}(\tilde{\mathbf{X}}_i^{(t)})\right). \tag{54}$$

*Proof of Lemma F.1.* For sample $i$ at iteration $t$, the student output satisfies

$$f_{\mathbf{W}^{(t)}}(\tilde{\mathbf{X}}_i^{(t)}) = \sum_{r=1}^{m} \sum_{j=1}^{P} \left(\phi\left(\langle \mathbf{w}_r^{(t)}, \tilde{\mathbf{x}}_{i,j}^{(t)} \rangle\right) - \phi\left(-\langle \mathbf{w}_r^{(t)}, \tilde{\mathbf{x}}_{i,j}^{(t)} \rangle\right)\right). \tag{55}$$

Since $\phi(z) - \phi(-z) = z|z|^2$, differentiating the loss $\ell(z) = \log(1 + e^{-z})$ with respect to $w_{r,1}^{(t)}$ gives

$$
\frac{\partial}{\partial w_{r,1}^{(t)}} \ell\left(y_i f_{\mathbf{W}^{(t)}}(\tilde{\mathbf{X}}_i^{(t)})\right) = -y_i \psi\left(y_i f_{\mathbf{W}^{(t)}}(\tilde{\mathbf{X}}_i^{(t)})\right) \frac{\partial}{\partial w_{r,1}^{(t)}} f_{\mathbf{W}^{(t)}}(\tilde{\mathbf{X}}_i^{(t)})
$$
$$
= -y_i \psi\left(y_i f_{\mathbf{W}^{(t)}}(\tilde{\mathbf{X}}_i^{(t)})\right) \sum_{j=1}^P 3\left\langle \mathbf{w}_r^{(t)}, \tilde{\mathbf{x}}_{i,j}^{(t)} \right\rangle^2 \left\langle \mathbf{e}_1, \tilde{\mathbf{x}}_{i,j}^{(t)} \right\rangle. \tag{56}
$$

Therefore, the one-step update satisfies

$$
w_{r,1}^{(t+1)} - w_{r,1}^{(t)} = -\frac{\eta}{N} \sum_{i=1}^N \frac{\partial}{\partial w_{r,1}^{(t)}} \ell\left(y_i f_{\mathbf{W}^{(t)}}(\tilde{\mathbf{X}}_i^{(t)})\right)
$$
$$
= \frac{3\eta}{N} \sum_{i=1}^N \psi\left(y_i f_{\mathbf{W}^{(t)}}(\tilde{\mathbf{X}}_i^{(t)})\right) \sum_{j=1}^P y_i \left\langle \mathbf{w}_r^{(t)}, \tilde{\mathbf{x}}_{i,j}^{(t)} \right\rangle^2 \left\langle \mathbf{e}_1, \tilde{\mathbf{x}}_{i,j}^{(t)} \right\rangle. \tag{57}
$$

For every noise patch $j \neq s(\mathbf{X}_i)$, we have

$$
\left\langle \mathbf{e}_1, \tilde{\mathbf{x}}_{i,j}^{(t)} \right\rangle = 0. \tag{58}
$$

Moreover, if $i \in \mathcal{S}_U$, then the signal patch is aligned with $\mathbf{v} = \mathbf{e}_d$, and hence

$$
\left\langle \mathbf{e}_1, \tilde{\mathbf{x}}_{i,s(\mathbf{X}_i)}^{(t)} \right\rangle = 0. \tag{59}
$$

Therefore,

$$
w_{r,1}^{(t+1)} - w_{r,1}^{(t)} = \frac{3\eta}{N} \sum_{i \in \mathcal{S}_L} \psi\left(y_i f_{\mathbf{W}^{(t)}}(\tilde{\mathbf{X}}_i^{(t)})\right) y_i \left\langle \mathbf{w}_r^{(t)}, \tilde{\mathbf{x}}_{i,s(\mathbf{X}_i)}^{(t)} \right\rangle^2 \left\langle \mathbf{e}_1, \tilde{\mathbf{x}}_{i,s(\mathbf{X}_i)}^{(t)} \right\rangle. \tag{60}
$$

Since $\psi(\cdot) > 0$ and $y_i \langle \mathbf{e}_1, \tilde{\mathbf{x}}_{i,s(\mathbf{X}_i)}^{(t)} \rangle \geq 0$, the signal weight $w_{r,1}^{(t)}$ is non-decreasing.

For the signal patch of a learnable sample, the perturbation constraint gives

$$
\alpha - \epsilon \leq y_i \left\langle \mathbf{e}_1, \tilde{\mathbf{x}}_{i,s(\mathbf{X}_i)}^{(t)} \right\rangle \leq \alpha. \tag{61}
$$

Since $\tilde{\mathbf{x}}_{i,s(\mathbf{X}_i)}^{(t)}$ is a scalar multiple of $\mathbf{e}_1$, we also have

$$
\left| \left\langle \mathbf{w}_r^{(t)}, \tilde{\mathbf{x}}_{i,s(\mathbf{X}_i)}^{(t)} \right\rangle \right| = |w_{r,1}^{(t)}| \left| \left\langle \mathbf{e}_1, \tilde{\mathbf{x}}_{i,s(\mathbf{X}_i)}^{(t)} \right\rangle \right|. \tag{62}
$$

Hence,

$$
(\alpha - \epsilon)^3 \left(w_{r,1}^{(t)}\right)^2 \leq y_i \left\langle \mathbf{w}_r^{(t)}, \tilde{\mathbf{x}}_{i,s(\mathbf{X}_i)}^{(t)} \right\rangle^2 \left\langle \mathbf{e}_1, \tilde{\mathbf{x}}_{i,s(\mathbf{X}_i)}^{(t)} \right\rangle \leq \alpha^3 \left(w_{r,1}^{(t)}\right)^2. \tag{63}
$$

Substituting these bounds into the update gives the two claimed inequalities. $\qquad\square$

**Lemma F.2** (Noise Coefficient Update Rule). For each filter $r \in [m]$ and each noise patch $(i,j)$ with $i \in [N]$ and $j \neq s(\mathbf{X}_i)$, define the coefficients recursively by

$$
\rho_{i,j,r}^{(0)} = 0, \tag{64}
$$

and

$$
\rho_{i,j,r}^{(t+1)} - \rho_{i,j,r}^{(t)} = \frac{3\eta}{N} \psi\left(y_i f_{\mathbf{W}^{(t)}}(\tilde{\mathbf{X}}_i^{(t)})\right) \langle \mathbf{w}_r^{(t)}, \mathbf{x}_{i,j} \rangle^2 \tag{65}
$$

for every iteration $t \geq 0$. Then the coefficients $\rho_{i,j,r}^{(t)}$ are monotonically non-decreasing. Moreover, for every $t \geq 0$,

$$
\langle \mathbf{w}_r^{(t)}, \mathbf{x}_{i,j} \rangle = \langle \mathbf{w}_r^{(0)}, \mathbf{x}_{i,j} \rangle + y_i \rho_{i,j,r}^{(t)} \|\mathbf{x}_{i,j}\|_2^2 + \sum_{\substack{k,q \\ (k,q) \neq (i,j)}} y_k \rho_{k,q,r}^{(t)} \langle \mathbf{x}_{k,q}, \mathbf{x}_{i,j} \rangle \tag{66}
$$

for every $i \in [N]$ and $j \neq s(\mathbf{X}_i)$.

*Proof of Lemma F.2.* For a sample $k$ at iteration $t$, the gradient with respect to $\mathbf{w}_r^{(t)}$ is

$$\nabla_{\mathbf{w}_r^{(t)}} \ell\left(y_k f_{\mathbf{W}^{(t)}}(\tilde{\mathbf{X}}_k^{(t)})\right) = -y_k \psi\left(y_k f_{\mathbf{W}^{(t)}}(\tilde{\mathbf{X}}_k^{(t)})\right) \sum_{q=1}^{P} 3\left\langle \mathbf{w}_r^{(t)}, \tilde{\mathbf{x}}_{k,q}^{(t)} \right\rangle^2 \tilde{\mathbf{x}}_{k,q}^{(t)}. \tag{67}$$

Hence,

$$
\begin{aligned}
\mathbf{w}_r^{(t+1)} - \mathbf{w}_r^{(t)} &= \frac{3\eta}{N} \sum_{k=1}^{N} \psi\left(y_k f_{\mathbf{W}^{(t)}}(\tilde{\mathbf{X}}_k^{(t)})\right) y_k \left\langle \mathbf{w}_r^{(t)}, \tilde{\mathbf{x}}_{k,s(\mathbf{X}_k)}^{(t)} \right\rangle^2 \tilde{\mathbf{x}}_{k,s(\mathbf{X}_k)}^{(t)} \\
&\quad + \frac{3\eta}{N} \sum_{k=1}^{N} \sum_{q \neq s(\mathbf{X}_k)} \psi\left(y_k f_{\mathbf{W}^{(t)}}(\tilde{\mathbf{X}}_k^{(t)})\right) \langle \mathbf{w}_r^{(t)}, \mathbf{x}_{k,q}\rangle^2 (y_k \mathbf{x}_{k,q}),
\end{aligned}
\tag{68}
$$

where we used $\tilde{\mathbf{x}}_{k,q}^{(t)} = \mathbf{x}_{k,q}$ for all $q \neq s(\mathbf{X}_k)$. By definition, the increment of $\rho_{i,j,r}^{(t)}$ is exactly the coefficient of $y_i \mathbf{x}_{i,j}$ in the second summation. Since $\psi(\cdot) > 0$ and the inner product is squared,

$$\rho_{i,j,r}^{(t+1)} - \rho_{i,j,r}^{(t)} \geq 0, \tag{69}$$

so $\rho_{i,j,r}^{(t)}$ is monotonically non-decreasing.

Now fix $i \in [N]$ and $j \neq s(\mathbf{X}_i)$. Since $\mathbf{x}_{i,j}$ is orthogonal to $\mathrm{span}\{\mathbf{e}_1, \mathbf{e}_d\}$, while each signal patch $\tilde{\mathbf{x}}_{k,s(\mathbf{X}_k)}^{(t)}$ lies in $\mathrm{span}\{\mathbf{e}_1, \mathbf{e}_d\}$, the signal-patch term is orthogonal to $\mathbf{x}_{i,j}$. Taking inner product with $\mathbf{x}_{i,j}$ therefore yields

$$\langle \mathbf{w}_r^{(t+1)}, \mathbf{x}_{i,j} \rangle = \langle \mathbf{w}_r^{(t)}, \mathbf{x}_{i,j} \rangle + \frac{3\eta}{N} \sum_{k=1}^{N} \sum_{q \neq s(\mathbf{X}_k)} \psi\left(y_k f_{\mathbf{W}^{(t)}}(\tilde{\mathbf{X}}_k^{(t)})\right) \langle \mathbf{w}_r^{(t)}, \mathbf{x}_{k,q}\rangle^2 \left(y_k \langle \mathbf{x}_{k,q}, \mathbf{x}_{i,j}\rangle\right). \tag{70}$$

Using the definition of the coefficient increments, this becomes

$$\langle \mathbf{w}_r^{(t+1)}, \mathbf{x}_{i,j} \rangle = \langle \mathbf{w}_r^{(t)}, \mathbf{x}_{i,j} \rangle + y_i\left(\rho_{i,j,r}^{(t+1)} - \rho_{i,j,r}^{(t)}\right)\|\mathbf{x}_{i,j}\|_2^2 + \sum_{\substack{k,q \\ (k,q) \neq (i,j)}} y_k\left(\rho_{k,q,r}^{(t+1)} - \rho_{k,q,r}^{(t)}\right)\langle \mathbf{x}_{k,q}, \mathbf{x}_{i,j}\rangle. \tag{71}$$

Summing this identity from $0$ to $t-1$ and using $\rho_{i,j,r}^{(0)} = 0$ gives

$$\langle \mathbf{w}_r^{(t)}, \mathbf{x}_{i,j} \rangle = \langle \mathbf{w}_r^{(0)}, \mathbf{x}_{i,j} \rangle + y_i\rho_{i,j,r}^{(t)}\|\mathbf{x}_{i,j}\|_2^2 + \sum_{\substack{k,q \\ (k,q) \neq (i,j)}} y_k\rho_{k,q,r}^{(t)}\langle \mathbf{x}_{k,q}, \mathbf{x}_{i,j}\rangle. \tag{72}$$

This proves the coefficient update rule, the monotonicity of the coefficients, and the stated inner-product decomposition. $\quad\square$

### F.2. Noise Stability Hypothesis

We state a noise stability hypothesis for the learnable samples. It will be used in the subsequent analysis and verified at the end of the section.

**Hypothesis F.3** (Noise Stability on Learnable Samples). On the event $\mathcal{E}$ defined in Lemma E.1, for every filter $r \in [m]$ and every noise patch $(i, j)$ with $i \in \mathcal{S}_L$ and $j \neq s(\mathbf{X}_i)$, the following bound holds for all $t \leq T$:

$$\left|\langle \mathbf{w}_r^{(t)}, \mathbf{x}_{i,j}\rangle\right| \leq 3\sigma_0 \sigma_n \sqrt{d \log\left(\frac{16NmP}{\delta}\right)} + \frac{21 p_{\mathrm{un}} NP \log^{1/3} T}{\sqrt{d}} \sqrt{\log\left(\frac{16N^2P^2}{\delta}\right)}. \tag{73}$$

See Lemma F.10 for the verification.

### F.3. Early-Phase Signal Learning

We analyze the phase before the signal learning time $T_0$. During this phase, the gradients on learnable samples remain non-negligible, allowing the signal coordinate to grow while the noise coefficients stay controlled.

**Lemma F.4** (Gradient Bound on Learnable Samples). On the event $\mathcal{E}$ defined in Lemma E.1, suppose that the conclusion of Hypothesis F.3 holds at iteration $t$. Then, for any learnable sample $i \in \mathcal{S}_L$, provided that the signal weights satisfy $w_{r,1}^{(t)} \leq \frac{C_0}{\alpha m^{1/3}}$ for all filters $r \in [m]$ and some absolute constant $C_0 > 0$, the negative sigmoid function satisfies:

$$\frac{1}{1 + \exp(2C_0^3)} \leq \psi\left(y_i f_{\mathbf{W}^{(t)}}(\tilde{\mathbf{X}}_i^{(t)})\right) \leq 1. \tag{74}$$

*Proof of Lemma F.4.* Fix any learnable sample $i \in \mathcal{S}_L$. We decompose the margin into the signal-patch contribution and the noise-patch residual:

$$y_i f_{\mathbf{W}^{(t)}}(\tilde{\mathbf{X}}_i^{(t)}) = \left(y_i \langle \mathbf{e}_1, \tilde{\mathbf{x}}_{i,s(\mathbf{X}_i)}^{(t)}\rangle\right)^3 \sum_{r\in[m]} (w_{r,1}^{(t)})^3 + \sum_{r\in[m]} \sum_{j \neq s(\mathbf{X}_i)} y_i \langle \mathbf{w}_r^{(t)}, \mathbf{x}_{i,j}\rangle^3. \tag{75}$$

By the perturbation constraint, $0 \leq y_i \langle \mathbf{e}_1, \tilde{\mathbf{x}}_{i,s(\mathbf{X}_i)}^{(t)}\rangle \leq \alpha$. Using the assumption $w_{r,1}^{(t)} \leq C_0/(\alpha m^{1/3})$ for all $r \in [m]$, we obtain

$$\left(y_i \langle \mathbf{e}_1, \tilde{\mathbf{x}}_{i,s(\mathbf{X}_i)}^{(t)}\rangle\right)^3 \sum_{r\in[m]} (w_{r,1}^{(t)})^3 \leq \alpha^3 \cdot m \cdot \left(\frac{C_0}{\alpha m^{1/3}}\right)^3 = C_0^3. \tag{76}$$

For the noise residual, Hypothesis F.3 guarantees that

$$\left|\langle \mathbf{w}_r^{(t)}, \mathbf{x}_{i,j}\rangle\right| \leq 3\sigma_0 \sigma_n \sqrt{d \log\left(\frac{16NmP}{\delta}\right)} + \frac{21 p_{\mathrm{un}} NP \log^{1/3} T}{\sqrt{d}} \sqrt{\log\left(\frac{16N^2P^2}{\delta}\right)} \tag{77}$$

for every $r \in [m]$ and $j \neq s(\mathbf{X}_i)$. Therefore,

$$\left|\sum_{r\in[m]} \sum_{j \neq s(\mathbf{X}_i)} y_i \langle \mathbf{w}_r^{(t)}, \mathbf{x}_{i,j}\rangle^3\right| \leq mP\left(3\sigma_0 \sigma_n \sqrt{d \log\left(\frac{16NmP}{\delta}\right)} + \frac{21 p_{\mathrm{un}} NP \log^{1/3} T}{\sqrt{d}} \sqrt{\log\left(\frac{16N^2P^2}{\delta}\right)}\right)^3. \tag{78}$$

Using $(a+b)^3 \leq 4(a^3 + b^3)$, we obtain

$$\left|\sum_{r\in[m]} \sum_{j \neq s(\mathbf{X}_i)} y_i \langle \mathbf{w}_r^{(t)}, \mathbf{x}_{i,j}\rangle^3\right| \leq 108\, mP\, \sigma_0^3 \sigma_n^3 d^{3/2} \log^{3/2}\left(\frac{16NmP}{\delta}\right)$$
$$+ 4 \cdot 21^3\, mP\, \frac{p_{\mathrm{un}}^3 N^3 P^3 \log T}{d^{3/2}} \log^{3/2}\left(\frac{16N^2P^2}{\delta}\right) \tag{79}$$
$$\leq C^{-1} + C^{-1} mP\, p_{\mathrm{un}}^3 \log T$$
$$\leq 2C^{-1} \leq C_0^3,$$

where the second inequality applies condition (C4) to the first term and condition (C2) to the second, and the third inequality follows from condition (C7). The final step holds because $C$ is chosen to be a sufficiently large constant in Condition D.5. Hence, the total margin in (75) is bounded from above by

$$y_i f_{\mathbf{W}^{(t)}}(\tilde{\mathbf{X}}_i^{(t)}) \leq C_0^3 + C_0^3 = 2C_0^3. \tag{80}$$

Since the negative sigmoid function is monotonically decreasing, this upper bound on the margin directly yields

$$\psi\left(y_i f_{\mathbf{W}^{(t)}}(\tilde{\mathbf{X}}_i^{(t)})\right) \geq \frac{1}{1 + \exp(2C_0^3)}. \tag{81}$$

Together with the trivial upper bound $\psi(\cdot) \leq 1$, this proves the claimed gradient bound on learnable samples. $\square$

**Lemma F.5** (Signal Learning Time). We formalize the signal growth statement in Lemma 4.9. Let $T_0$ be the first hitting time at which the maximum signal component reaches the threshold $C_0/(\alpha m^{1/3})$:

$$T_0 := \min \left\{ t \geq 0 \;\middle|\; \max_{r \in [m]} w_{r,1}^{(t)} > \frac{C_0}{\alpha m^{1/3}} \right\}. \tag{82}$$

On the event $\mathcal{E}$ defined in Lemma E.1, suppose that Hypothesis F.3 holds for all iterations $\tau < T_0$. Then, this time satisfies:

$$\frac{1}{12\sqrt{2\log\left(\frac{16m}{\delta}\right)}\eta\alpha^3\sigma_0} \leq T_0 \leq \frac{5\exp(2C_0^3)}{\eta(\alpha-\epsilon)^3(1-p_{\mathrm{un}})\sigma_0}. \tag{83}$$

*Proof of Lemma F.5.* Invoking Lemma F.1 and observing that the sigmoid function is strictly upper-bounded by 1, we obtain the upper bound on the update:

$$w_{r,1}^{(t+1)} - w_{r,1}^{(t)} \leq \frac{3\eta\alpha^3}{N}\left(w_{r,1}^{(t)}\right)^2 \sum_{i \in \mathcal{S}_L} \psi\left(y_i f_{\mathbf{W}^{(t)}}(\tilde{\mathbf{X}}_i^{(t)})\right) \leq 3\eta\alpha^3\left(w_{r,1}^{(t)}\right)^2. \tag{84}$$

For the lower bound, we combine Lemma F.1 with Lemma F.4:

$$
\begin{aligned}
w_{r,1}^{(t+1)} - w_{r,1}^{(t)} &\geq \frac{3\eta(\alpha-\epsilon)^3}{N}\left(w_{r,1}^{(t)}\right)^2 \sum_{i \in \mathcal{S}_L} \psi\left(y_i f_{\mathbf{W}^{(t)}}(\tilde{\mathbf{X}}_i^{(t)})\right) \\
&\geq \frac{3\eta(\alpha-\epsilon)^3}{N}\left(w_{r,1}^{(t)}\right)^2 \frac{|\mathcal{S}_L|}{1+\exp(2C_0^3)} \\
&= \frac{3\eta(\alpha-\epsilon)^3(1-p_{\mathrm{un}})}{1+\exp(2C_0^3)}\left(w_{r,1}^{(t)}\right)^2.
\end{aligned}
\tag{85}
$$

Let $A$ and $B$ be defined by

$$A := 3\eta\alpha^3, \qquad B := \frac{3\eta(\alpha-\epsilon)^3(1-p_{\mathrm{un}})}{1+\exp(2C_0^3)}. \tag{86}$$

This leads to the following recursive bounds:

$$w_{r,1}^{(t+1)} \leq w_{r,1}^{(t)} + A(w_{r,1}^{(t)})^2, \qquad w_{r,1}^{(t+1)} \geq w_{r,1}^{(t)} + B(w_{r,1}^{(t)})^2. \tag{87}$$

On the event $\mathcal{E}$, the upper bound in (P7) with condition (C4) guarantees that the initial point is small enough compared to the target threshold , $\max_{r \in [m]} w_{r,1}^{(0)} \leq \sigma_0\sqrt{2\log\left(\frac{16m}{\delta}\right)} \leq \frac{1}{2}\frac{C_0}{\alpha m^{1/3}}$, and the lower bound in (P7) guarantees the existence of a filter $r \in [m]$ with $w_{r,1}^{(0)} \geq \frac{1}{2}\sigma_0$. We track the evolution of this filter.

Applying Lemma I.1 with the target threshold $\frac{C_0}{\alpha m^{1/3}}$, we obtain

$$T_0 \leq \frac{3}{Bw_{r,1}^{(0)}} + \frac{8A}{B}\left\lceil \frac{\log\left(\frac{C_0}{\alpha m^{1/3}}/w_{r,1}^{(0)}\right)}{\log 2} \right\rceil. \tag{88}$$

Substituting $w_{r,1}^{(0)} \geq \frac{1}{2}\sigma_0$ and the definitions of $A$ and $B$, we expand the bound:

$$
\begin{aligned}
T_0 &\leq \frac{2(1+\exp(2C_0^3))}{\eta(\alpha-\epsilon)^3(1-p_{\mathrm{un}})\sigma_0} + \frac{8(1+\exp(2C_0^3))\alpha^3}{(\alpha-\epsilon)^3(1-p_{\mathrm{un}})}\left\lceil \log_2\left(\frac{2C_0}{\alpha\sigma_0 m^{1/3}}\right) \right\rceil \\
&= \frac{2(1+\exp(2C_0^3))}{(\alpha-\epsilon)^3(1-p_{\mathrm{un}})}\left(\frac{1}{\eta\sigma_0} + 4\alpha^3\left\lceil \log_2\left(\frac{2C_0}{\alpha\sigma_0 m^{1/3}}\right) \right\rceil\right) \\
&\leq \frac{2(1+\exp(2C_0^3))}{(\alpha-\epsilon)^3(1-p_{\mathrm{un}})}\left(\frac{2}{\eta\sigma_0}\right) \\
&\leq \frac{5\exp(2C_0^3)}{\eta(\alpha-\epsilon)^3(1-p_{\mathrm{un}})\sigma_0},
\end{aligned}
\tag{89}
$$

where the second inequality follows from condition (C5), and the final inequality holds for a sufficiently large absolute constant $C_0$.

For the lower bound on the hitting time, consider the minimum time $T_{\text{dbl}}$ required simply to double the initial weight $w_{r,1}^{(0)}$. For any step $t$ before this doubling occurs, the weight satisfies $w_{r,1}^{(t)} \leq 2w_{r,1}^{(0)}$. Consequently, the update magnitude is strictly upper-bounded by:

$$w_{r,1}^{(t+1)} - w_{r,1}^{(t)} \leq A(2w_{r,1}^{(0)})^2 = 4A(w_{r,1}^{(0)})^2. \tag{90}$$

To traverse the distance of $w_{r,1}^{(0)}$ (from $w_{r,1}^{(0)}$ to $2w_{r,1}^{(0)}$) subject to this maximum growth rate, the required steps must satisfy:

$$T_0 \geq T_{\text{dbl}} \geq \frac{w_{r,1}^{(0)}}{4A(w_{r,1}^{(0)})^2} = \frac{1}{4Aw_{r,1}^{(0)}}. \tag{91}$$

On the event $\mathcal{E}$, the upper bound in (P7) gives $w_{r,1}^{(0)} \leq \sigma_0 \sqrt{2\log\left(\frac{16m}{\delta}\right)}$. Substituting this along with the definition $A = 3\eta\alpha^3$, we obtain:

$$T_0 \geq \frac{1}{12\sqrt{2}\eta\alpha^3\sigma_0\sqrt{\log\left(\frac{16m}{\delta}\right)}}. \tag{92}$$

Combining the upper and lower bounds proves the claimed estimates for $T_0$. $\qquad\square$

**Lemma F.6** (Early-Phase Noise Coefficient Bound). Let

$$\hat{T}_0 := \frac{5\exp(2C_0^3)}{\eta(\alpha - \epsilon)^3(1 - p_{\text{un}})\sigma_0}. \tag{93}$$

On the event $\mathcal{E}$ defined in Lemma E.1, for any iteration $t \leq \hat{T}_0$, the noise coefficient satisfies

$$\max_{i,j,r} \rho_{i,j,r}^{(t)} \leq 100\exp(2C_0^3)\log\left(\frac{16NmP}{\delta}\right) \cdot \frac{\sigma_0\sigma_n^2 d}{N(\alpha - \epsilon)^3(1 - p_{\text{un}})}. \tag{94}$$

In particular, if $T_0$ denotes the hitting time from Lemma F.5, then $T_0 \leq \hat{T}_0$, so the same bound holds for all $t \leq T_0$.

*Proof of Lemma F.6.* Define the threshold

$$\rho_{\text{th}} := 100\exp(2C_0^3)\log\left(\frac{16NmP}{\delta}\right) \cdot \frac{\sigma_0\sigma_n^2 d}{N(\alpha - \epsilon)^3(1 - p_{\text{un}})}. \tag{95}$$

Assume for contradiction that there exists an iteration $t \leq \hat{T}_0$ such that $t$ is the first iteration where

$$\max_{i,j,r} \rho_{i,j,r}^{(t)} > \rho_{\text{th}}. \tag{96}$$

Then, by the minimality of $t$, we have $\rho_{i,j,r}^{(\tau)} \leq \rho_{\text{th}}$ for all $\tau < t$ and all $i, j, r$.

Fix any iteration $\tau < t$. By the decomposition in Lemma F.2, the triangle inequality and the nonnegativity of the noise coefficients give

$$\left|\langle \mathbf{w}_r^{(\tau)}, \mathbf{x}_{i,j}\rangle\right| \leq \left|\langle \mathbf{w}_r^{(0)}, \mathbf{x}_{i,j}\rangle\right| + \rho_{i,j,r}^{(\tau)}\|\mathbf{x}_{i,j}\|_2^2 + \sum_{\substack{k,q \\ (k,q)\neq(i,j)}} \rho_{k,q,r}^{(\tau)}\left|\langle \mathbf{x}_{k,q}, \mathbf{x}_{i,j}\rangle\right|. \tag{97}$$

On the event $\mathcal{E}$, the bounds (P1), (P2), and (P6) imply

$$\|\mathbf{x}_{i,j}\|_2^2 \leq \frac{3}{2}\sigma_n^2 d, \quad \left|\langle \mathbf{x}_{k,q}, \mathbf{x}_{i,j}\rangle\right| \leq 2\sigma_n^2\sqrt{d\log\left(\frac{16N^2P^2}{\delta}\right)}, \quad \text{and} \quad \left|\langle \mathbf{w}_r^{(0)}, \mathbf{x}_{i,j}\rangle\right| \leq 2\sigma_0\sigma_n\sqrt{d\log\left(\frac{16NmP}{\delta}\right)}. \tag{98}$$

Substituting these bounds together with the assumed upper bound $\rho_{i,j,r}^{(\tau)} \leq \rho_{\text{th}}$, we obtain

$$
\begin{aligned}
\left| \langle \mathbf{w}_r^{(\tau)}, \mathbf{x}_{i,j} \rangle \right| &\leq 2\sigma_0 \sigma_n \sqrt{d \log\left(\frac{16NmP}{\delta}\right)} + \rho_{\text{th}} \cdot \frac{3}{2}\sigma_n^2 d + 2NP\rho_{\text{th}}\sigma_n^2 \sqrt{d \log\left(\frac{16N^2P^2}{\delta}\right)} \\
&= 2\sigma_0 \sigma_n \sqrt{d \log\left(\frac{16NmP}{\delta}\right)} + \rho_{\text{th}}\sigma_n^2 d \left(\frac{3}{2} + \frac{2NP}{\sqrt{d}}\sqrt{\log\left(\frac{16N^2P^2}{\delta}\right)}\right) \\
&\leq 2\sigma_0 \sigma_n \sqrt{d \log\left(\frac{16NmP}{\delta}\right)} + \left(100\exp(2C_0^3)\log\left(\frac{16NmP}{\delta}\right) \cdot \frac{\sigma_0 \sigma_n^2 d}{N(\alpha-\epsilon)^3(1-p_{\text{un}})}\right) \cdot 2\sigma_n^2 d \\
&\leq \sqrt{\frac{20}{3}}\sigma_0 \sigma_n \sqrt{d \log\left(\frac{16NmP}{\delta}\right)},
\end{aligned}
\tag{99}
$$

where the second inequality follows from condition (C2), and the final inequality holds under conditions (C3) and (C6).

Summing the update rule from Lemma F.2 over $\tau = 0, \ldots, t-1$ and using $\psi\left(y_i f_{\mathbf{W}^{(\tau)}}(\tilde{\mathbf{X}}_i^{(\tau)})\right) \leq 1$, we obtain

$$
\rho_{i,j,r}^{(t)} = \sum_{\tau=0}^{t-1} \frac{3\eta}{N}\psi\left(y_i f_{\mathbf{W}^{(\tau)}}(\tilde{\mathbf{X}}_i^{(\tau)})\right) \langle \mathbf{w}_r^{(\tau)}, \mathbf{x}_{i,j} \rangle^2 \leq t \cdot \frac{20\eta}{N}\sigma_0^2 \sigma_n^2 d \log\left(\frac{16NmP}{\delta}\right).
\tag{100}
$$

Since $t \leq \hat{T}_0$, we have

$$
\rho_{i,j,r}^{(t)} \leq \hat{T}_0 \cdot \frac{20\eta}{N}\sigma_0^2 \sigma_n^2 d \log\left(\frac{16NmP}{\delta}\right) = \frac{5\exp(2C_0^3)}{\eta(\alpha-\epsilon)^3(1-p_{\text{un}})\sigma_0} \cdot \frac{20\eta}{N}\sigma_0^2 \sigma_n^2 d \log\left(\frac{16NmP}{\delta}\right) = \rho_{\text{th}}.
\tag{101}
$$

This contradicts the choice of $t$. Therefore, for all $i, j, r$ and all $t \leq \hat{T}_0$,

$$
\rho_{i,j,r}^{(t)} \leq 100\exp(2C_0^3)\log\left(\frac{16NmP}{\delta}\right) \cdot \frac{\sigma_0 \sigma_n^2 d}{N(\alpha-\epsilon)^3(1-p_{\text{un}})}.
\tag{102}
$$

$\square$

## F.4. Post-$T_0$ Control on Learnable Samples

Once the signal reaches its target scale at time $T_0$, learnable samples no longer drive large updates. We show that the signal weights remain bounded and that the cumulative gradient contribution from learnable samples is controlled.

**Lemma F.7** (Signal Weight Bound). On the event $\mathcal{E}$ defined in Lemma E.1, suppose that Hypothesis F.3 holds for all iterations $\tau \leq t$. Then, for any iteration $t \leq T$ and all filters $r \in [m]$, the signal component is uniformly bounded by

$$
w_{r,1}^{(t)} \leq \frac{3\log^{1/3} T}{\alpha}.
\tag{103}
$$

*Proof of Lemma F.7.* Fix any iteration $t \leq T$ and assume that Hypothesis F.3 holds for all iterations up to $t$. Define the threshold

$$
W_{\text{th}} := \frac{3\log^{1/3} T}{\alpha}.
\tag{104}
$$

Assume for contradiction that there exists an iteration $t \leq T$ such that $t$ is the first iteration where

$$
w_{r,1}^{(t)} > W_{\text{th}}
\tag{105}
$$

for some filter $r \in [m]$. Then, by the minimality of $t$, we have $w_{r',1}^{(\tau)} \leq W_{\text{th}}$ for all $\tau < t$ and all $r' \in [m]$.

On the event $\mathcal{E}$, the upper bound in (P7) gives $w_{r,1}^{(0)} \leq \sigma_0 \sqrt{2\log\left(\frac{16m}{\delta}\right)}$. By condition (C4), this is strictly less than $0.5W_{\text{th}}$. Thus, there exists a last iteration $t_0 < t$ such that $w_{r,1}^{(t_0)} \leq 0.5W_{\text{th}}$. Consequently, for all $\tau \in [t_0+1, t-1]$,

$$
w_{r,1}^{(\tau)} > 0.5W_{\text{th}}.
\tag{106}
$$

At iteration $t_0$, Lemma F.1 yields

$$w_{r,1}^{(t_0+1)} - w_{r,1}^{(t_0)} \le 3\eta\alpha^3 \left(w_{r,1}^{(t_0)}\right)^2. \tag{107}$$

Using $w_{r,1}^{(t_0)} \le 0.5W_{\text{th}}$, we obtain

$$w_{r,1}^{(t_0+1)} - w_{r,1}^{(t_0)} \le 3\eta\alpha^3 \left(0.5W_{\text{th}}\right)^2 = \frac{27}{4}\eta\alpha \log^{2/3} T. \tag{108}$$

By condition (C5), the right-hand side is strictly bounded by $0.1W_{\text{th}}$. Therefore,

$$w_{r,1}^{(t_0+1)} \le 0.5W_{\text{th}} + 0.1W_{\text{th}} = 0.6W_{\text{th}}. \tag{109}$$

Now fix any $\tau \in [t_0 + 1, t - 1]$ and any learnable sample $i \in \mathcal{S}_L$. Since $w_{r,1}^{(\tau)} > 0.5W_{\text{th}}$, we lower-bound the margin $y_i f_{\mathbf{W}^{(\tau)}}(\tilde{\mathbf{X}}_i^{(\tau)})$. By Lemma F.1, the signal coordinates are monotonically non-decreasing, so $w_{r',1}^{(\tau)} \ge w_{r',1}^{(0)}$ for all $r' \in [m]$. On the event $\mathcal{E}$, (P5) implies

$$w_{r',1}^{(0)} \ge -\sigma_0\sqrt{2\log\left(\frac{16m}{\delta}\right)} \tag{110}$$

for all $r' \in [m]$.

Moreover, by Hypothesis F.3, we have

$$\left|\langle \mathbf{w}_{r'}^{(\tau)}, \mathbf{x}_{i,j}\rangle\right| \le 3\sigma_0\sigma_n\sqrt{d\log\left(\frac{16NmP}{\delta}\right)} + \frac{21p_{\text{un}}NP\log^{1/3} T}{\sqrt{d}}\sqrt{\log\left(\frac{16N^2P^2}{\delta}\right)} \tag{111}$$

for every $r' \in [m]$ and every $j \neq s(\mathbf{X}_i)$. Therefore,

$$\left|\sum_{r'\in[m]}\sum_{j\neq s(\mathbf{X}_i)} y_i\langle \mathbf{w}_{r'}^{(\tau)}, \mathbf{x}_{i,j}\rangle^3\right| \le mP\left(3\sigma_0\sigma_n\sqrt{d\log\left(\frac{16NmP}{\delta}\right)} + \frac{21p_{\text{un}}NP\log^{1/3} T}{\sqrt{d}}\sqrt{\log\left(\frac{16N^2P^2}{\delta}\right)}\right)^3. \tag{112}$$

Using $(a + b)^3 \le 4(a^3 + b^3)$, we obtain

$$\left|\sum_{r'\in[m]}\sum_{j\neq s(\mathbf{X}_i)} y_i\langle \mathbf{w}_{r'}^{(\tau)}, \mathbf{x}_{i,j}\rangle^3\right| \le 108\, mP\, \sigma_0^3\sigma_n^3 d^{3/2}\log^{3/2}\left(\frac{16NmP}{\delta}\right)$$
$$+ 4\cdot 21^3\, mP\, \frac{p_{\text{un}}^3 N^3 P^3\log T}{d^{3/2}}\log^{3/2}\left(\frac{16N^2P^2}{\delta}\right) \tag{113}$$
$$\le C^{-1} + C^{-1}mP\, p_{\text{un}}^3\log T$$
$$\le 2C^{-1},$$

where the second inequality applies condition (C4) to the first term and condition (C2) to the second, and the third inequality follows from condition (C7).

Therefore, the total margin is bounded from below by

$$y_i f_{\mathbf{W}^{(\tau)}}(\tilde{\mathbf{X}}_i^{(\tau)}) \ge (\alpha - \epsilon)^3(w_{r,1}^{(\tau)})^3 + (\alpha - \epsilon)^3\sum_{r'\neq r}(w_{r',1}^{(\tau)})^3 + \sum_{r'\in[m]}\sum_{j\neq s(\mathbf{X}_i)} y_i\left\langle \mathbf{w}_{r'}^{(\tau)}, \mathbf{x}_{i,j}\right\rangle^3$$
$$\ge (\alpha - \epsilon)^3(w_{r,1}^{(\tau)})^3 - (\alpha - \epsilon)^3 m\sigma_0^3\left(2\log\left(\frac{16m}{\delta}\right)\right)^{3/2} - 2C^{-1}. \tag{114}$$

Using $w_{r,1}^{(\tau)} > 0.5W_{\text{th}}$, we further obtain

$$y_i f_{\mathbf{W}^{(\tau)}}(\tilde{\mathbf{X}}_i^{(\tau)}) > \frac{27}{8}\left(1 - \frac{\epsilon}{\alpha}\right)^3\log T - (\alpha - \epsilon)^3 m\sigma_0^3\left(2\log\left(\frac{16m}{\delta}\right)\right)^{3/2} - 2C^{-1}. \tag{115}$$

Under condition (C6), the factor $\left(1 - \frac{\epsilon}{\alpha}\right)^3$ can be strictly lower-bounded by 0.9. Substituting this yields:

$$y_i f_{\mathbf{W}^{(\tau)}}(\tilde{\mathbf{X}}_i^{(\tau)}) > 3.0375 \log T - \alpha^3 m \sigma_0^3 \left(2 \log \left(\frac{16m}{\delta}\right)\right)^{3/2} - 2C^{-1}. \tag{116}$$

By condition (C4), the negative initialization term is negligibly small, and the noise term $2C^{-1}$ is small. Since the excess margin of $0.0375 \log T$ strictly dominates these terms, we can absorb them to establish a lower bound:

$$y_i f_{\mathbf{W}^{(\tau)}}(\tilde{\mathbf{X}}_i^{(\tau)}) \geq 3 \log T. \tag{117}$$

Using $\psi(z) \leq \exp(-z)$, it follows that

$$\psi\left(y_i f_{\mathbf{W}^{(\tau)}}(\tilde{\mathbf{X}}_i^{(\tau)})\right) \leq \exp(-3 \log T) = T^{-3}. \tag{118}$$

Moreover, because $\tau < t$, the choice of $t$ implies

$$w_{r',1}^{(\tau)} \leq W_{\text{th}} \tag{119}$$

for all $r' \in [m]$. Summing the update rule over $\tau \in [t_0 + 1, t - 1]$, Lemma F.1 gives

$$\sum_{\tau=t_0+1}^{t-1} \left(w_{r,1}^{(\tau+1)} - w_{r,1}^{(\tau)}\right) \leq T \cdot 3\eta\alpha^3 (W_{\text{th}})^2 \cdot T^{-3}. \tag{120}$$

Substituting the definition of $W_{\text{th}}$,

$$\sum_{\tau=t_0+1}^{t-1} \left(w_{r,1}^{(\tau+1)} - w_{r,1}^{(\tau)}\right) \leq \frac{27\eta\alpha \log^{2/3} T}{T^2}. \tag{121}$$

Under condition (C5), the right-hand side is strictly bounded by $0.1 W_{\text{th}}$. Hence,

$$w_{r,1}^{(t)} = w_{r,1}^{(t_0+1)} + \sum_{\tau=t_0+1}^{t-1} \left(w_{r,1}^{(\tau+1)} - w_{r,1}^{(\tau)}\right) \leq 0.6 W_{\text{th}} + 0.1 W_{\text{th}} = 0.7 W_{\text{th}}. \tag{122}$$

This contradicts the choice of $t$. Therefore, for all $r \in [m]$ and all $t \leq T$,

$$w_{r,1}^{(t)} \leq \frac{3 \log^{1/3} T}{\alpha}. \tag{123}$$

$\square$

**Lemma F.8** (Gradient Decay on Learnable Samples)**.** On the event $\mathcal{E}$ defined in Lemma E.1, suppose that Hypothesis F.3 holds for all iterations $\tau \leq t$. Then, for any iteration $t \in [T_0 + 1, T]$, where $T_0$ is defined in Lemma F.5, and for any learnable sample $i \in \mathcal{S}_L$, we have

$$\sum_{\tau=T_0}^{t-1} \psi\left(y_i f_{\mathbf{W}^{(\tau)}}(\tilde{\mathbf{X}}_i^{(\tau)})\right) \leq \frac{2m^{2/3} \log^{1/3} T}{C_0^2 \left(1 - \frac{\epsilon}{\alpha}\right)^3 \eta\alpha^2}. \tag{124}$$

Furthermore, this implies a sublinear decay at the current iteration $t$:

$$\psi\left(y_i f_{\mathbf{W}^{(t)}}(\tilde{\mathbf{X}}_i^{(t)})\right) \leq \frac{4m^{2/3} \log^{1/3} T}{C_0^2 \left(1 - \frac{\epsilon}{\alpha}\right)^3 \eta\alpha^2 (t - T_0)}. \tag{125}$$

*Proof of Lemma F.8.* By Lemma F.5, for every $\tau \geq T_0$ we have

$$\max_{r \in [m]} w_{r,1}^{(\tau)} \geq \frac{C_0}{\alpha m^{1/3}}. \tag{126}$$

Applying Lemma F.1, we obtain

$$
\max_{r\in[m]} w_{r,1}^{(\tau+1)} - \max_{r\in[m]} w_{r,1}^{(\tau)} \geq \frac{3\eta(\alpha-\epsilon)^3}{N} \left(\max_{r\in[m]} w_{r,1}^{(\tau)}\right)^2 \sum_{i\in\mathcal{S}_L} \psi\left(y_i f_{\mathbf{W}^{(\tau)}}(\tilde{\mathbf{X}}_i^{(\tau)})\right)
$$

$$
\geq \frac{3\eta\alpha C_0^2}{m^{2/3}} \left(1-\frac{\epsilon}{\alpha}\right)^3 \frac{1}{N} \sum_{i\in\mathcal{S}_L} \psi\left(y_i f_{\mathbf{W}^{(\tau)}}(\tilde{\mathbf{X}}_i^{(\tau)})\right). \tag{127}
$$

Rearranging gives

$$
\frac{1}{N} \sum_{i\in\mathcal{S}_L} \psi\left(y_i f_{\mathbf{W}^{(\tau)}}(\tilde{\mathbf{X}}_i^{(\tau)})\right) \leq \frac{m^{2/3}}{3C_0^2\left(1-\frac{\epsilon}{\alpha}\right)^3\eta\alpha} \left(\max_{r\in[m]} w_{r,1}^{(\tau+1)} - \max_{r\in[m]} w_{r,1}^{(\tau)}\right). \tag{128}
$$

Summing from $\tau = T_0$ to $t-1$ yields

$$
\sum_{\tau=T_0}^{t-1} \frac{1}{N} \sum_{i\in\mathcal{S}_L} \psi\left(y_i f_{\mathbf{W}^{(\tau)}}(\tilde{\mathbf{X}}_i^{(\tau)})\right) \leq \frac{m^{2/3}}{3C_0^2\left(1-\frac{\epsilon}{\alpha}\right)^3\eta\alpha} \sum_{\tau=T_0}^{t-1} \left(\max_{r\in[m]} w_{r,1}^{(\tau+1)} - \max_{r\in[m]} w_{r,1}^{(\tau)}\right)
$$

$$
= \frac{m^{2/3}}{3C_0^2\left(1-\frac{\epsilon}{\alpha}\right)^3\eta\alpha} \left(\max_{r\in[m]} w_{r,1}^{(t)} - \max_{r\in[m]} w_{r,1}^{(T_0)}\right). \tag{129}
$$

Now invoke Lemma F.7 to obtain

$$
\max_{r\in[m]} w_{r,1}^{(t)} \leq \frac{3\log^{1/3} T}{\alpha}. \tag{130}
$$

Since $\max_{r\in[m]} w_{r,1}^{(T_0)} \geq 0$, it follows that

$$
\sum_{\tau=T_0}^{t-1} \frac{1}{N} \sum_{i\in\mathcal{S}_L} \psi\left(y_i f_{\mathbf{W}^{(\tau)}}(\tilde{\mathbf{X}}_i^{(\tau)})\right) \leq \frac{m^{2/3}\log^{1/3} T}{C_0^2\left(1-\frac{\epsilon}{\alpha}\right)^3\eta\alpha^2}. \tag{131}
$$

Next, we convert the cumulative average bound into an individual-sample bound. By Hypothesis F.3, for every $r\in[m]$, every $j\neq s(\mathbf{X}_i)$, and every $\tau\in[T_0, t]$, we have

$$
\left|\langle \mathbf{w}_r^{(\tau)}, \mathbf{x}_{i,j}\rangle\right| \leq 3\sigma_0\sigma_n\sqrt{d\log\left(\frac{16NmP}{\delta}\right)} + \frac{21p_{\mathrm{un}}NP\log^{1/3} T}{\sqrt{d}}\sqrt{\log\left(\frac{16N^2P^2}{\delta}\right)}. \tag{132}
$$

Therefore,

$$
\left|\sum_{r\in[m]}\sum_{j\neq s(\mathbf{X}_i)} y_i\langle \mathbf{w}_r^{(\tau)}, \mathbf{x}_{i,j}\rangle^3\right| \leq mP\left(3\sigma_0\sigma_n\sqrt{d\log\left(\frac{16NmP}{\delta}\right)} + \frac{21p_{\mathrm{un}}NP\log^{1/3} T}{\sqrt{d}}\sqrt{\log\left(\frac{16N^2P^2}{\delta}\right)}\right)^3. \tag{133}
$$

Using $(a+b)^3 \leq 4(a^3 + b^3)$, we obtain

$$
\left|\sum_{r\in[m]}\sum_{j\neq s(\mathbf{X}_i)} y_i\langle \mathbf{w}_r^{(\tau)}, \mathbf{x}_{i,j}\rangle^3\right| \leq 108\, mP\, \sigma_0^3\sigma_n^3 d^{3/2}\log^{3/2}\left(\frac{16NmP}{\delta}\right)
$$

$$
+ 4\cdot 21^3\, mP\, \frac{p_{\mathrm{un}}^3 N^3 P^3\log T}{d^{3/2}}\log^{3/2}\left(\frac{16N^2P^2}{\delta}\right) \tag{134}
$$

$$
\leq C^{-1} + C^{-1}mP\, p_{\mathrm{un}}^3\log T
$$

$$
\leq 2C^{-1},
$$

where the second inequality follows from conditions (C2) and (C4), and the third inequality follows from condition (C7).

Hence, for every $\tau \in [T_0, t]$ and every learnable sample $i \in \mathcal{S}_L$,

$$(\alpha - \epsilon)^3 \sum_{r \in [m]} (w_{r,1}^{(\tau)})^3 - 2C^{-1} \leq y_i f_{\mathbf{W}^{(\tau)}}(\tilde{\mathbf{X}}_i^{(\tau)}) \leq (\alpha - \epsilon)^3 \sum_{r \in [m]} (w_{r,1}^{(\tau)})^3 + 2C^{-1}. \tag{135}$$

Therefore, for any two learnable samples $i, k \in \mathcal{S}_L$,

$$\frac{\psi\left(y_i f_{\mathbf{W}^{(\tau)}}(\tilde{\mathbf{X}}_i^{(\tau)})\right)}{\psi\left(y_k f_{\mathbf{W}^{(\tau)}}(\tilde{\mathbf{X}}_k^{(\tau)})\right)} \leq \exp(4C^{-1}). \tag{136}$$

Averaging over $k \in \mathcal{S}_L$ gives

$$\psi\left(y_i f_{\mathbf{W}^{(\tau)}}(\tilde{\mathbf{X}}_i^{(\tau)})\right) \leq \frac{\exp(4C^{-1})}{1 - p_{\text{un}}} \left( \frac{1}{N} \sum_{k \in \mathcal{S}_L} \psi\left(y_k f_{\mathbf{W}^{(\tau)}}(\tilde{\mathbf{X}}_k^{(\tau)})\right) \right)$$
$$\leq 2 \left( \frac{1}{N} \sum_{k \in \mathcal{S}_L} \psi\left(y_k f_{\mathbf{W}^{(\tau)}}(\tilde{\mathbf{X}}_k^{(\tau)})\right) \right), \tag{137}$$

where the second inequality follows from condition (C7) by taking $C$ sufficiently large.

Summing this inequality over $\tau = T_0, \ldots, t - 1$ and using the cumulative average bound established above, we conclude that

$$\sum_{\tau = T_0}^{t-1} \psi\left(y_i f_{\mathbf{W}^{(\tau)}}(\tilde{\mathbf{X}}_i^{(\tau)})\right) \leq 2 \left( \sum_{\tau = T_0}^{t-1} \frac{1}{N} \sum_{i \in \mathcal{S}_L} \psi\left(y_i f_{\mathbf{W}^{(\tau)}}(\tilde{\mathbf{X}}_i^{(\tau)})\right) \right) \leq \frac{2m^{2/3} \log^{1/3} T}{C_0^2 \left(1 - \frac{\epsilon}{\alpha}\right)^3 \eta \alpha^2}. \tag{138}$$

To extract the bound at iteration $t$, we use the monotonicity of the signal weights. By Lemma F.1, $w_{r,1}^{(\tau)} \leq w_{r,1}^{(t)}$ for all $\tau \leq t$. Thus, the upper bound of the margin on (135) for any $\tau \in [T_0, t]$ satisfies

$$y_i f_{\mathbf{W}^{(\tau)}}(\tilde{\mathbf{X}}_i^{(\tau)}) \leq (\alpha - \epsilon)^3 \sum_{r \in [m]} (w_{r,1}^{(t)})^3 + 2C^{-1}. \tag{139}$$

Since $\psi$ is monotonically decreasing, this provides a uniform lower bound for the past gradients:

$$\psi\left(y_i f_{\mathbf{W}^{(\tau)}}(\tilde{\mathbf{X}}_i^{(\tau)})\right) \geq \psi\left( (\alpha - \epsilon)^3 \sum_{r \in [m]} (w_{r,1}^{(t)})^3 + 2C^{-1} \right). \tag{140}$$

Summing this uniform lower bound over $\tau \in [T_0, t-1]$ and applying the cumulative bound derived above, we obtain

$$(t - T_0) \psi\left( (\alpha - \epsilon)^3 \sum_{r \in [m]} (w_{r,1}^{(t)})^3 + 2C^{-1} \right) \leq \frac{2m^{2/3} \log^{1/3} T}{C_0^2 \left(1 - \frac{\epsilon}{\alpha}\right)^3 \eta \alpha^2}. \tag{141}$$

At the current iteration $t$, the margin on (135) is lower-bounded by

$$y_i f_{\mathbf{W}^{(t)}}(\tilde{\mathbf{X}}_i^{(t)}) \geq (\alpha - \epsilon)^3 \sum_{r \in [m]} (w_{r,1}^{(t)})^3 - 2C^{-1}. \tag{142}$$

Using the property $\psi(z_1) \leq \exp(z_2 - z_1) \psi(z_2)$, we connect the gradient at $t$ to the uniform lower bound:

$$\psi\left(y_i f_{\mathbf{W}^{(t)}}(\tilde{\mathbf{X}}_i^{(t)})\right) \leq \exp(4C^{-1}) \psi\left( (\alpha - \epsilon)^3 \sum_{r \in [m]} (w_{r,1}^{(t)})^3 + 2C^{-1} \right). \tag{143}$$

For a sufficiently large constant $C$, we have $\exp(4C^{-1}) \leq 2$. Combining these inequalities yields the bound:

$$\psi\left(y_i f_{\mathbf{W}^{(t)}}(\tilde{\mathbf{X}}_i^{(t)})\right) \leq \frac{4m^{2/3} \log^{1/3} T}{C_0^2 \left(1 - \frac{\epsilon}{\alpha}\right)^3 \eta \alpha^2 (t - T_0)}. \tag{144}$$

$$\square$$

### F.5. Global Noise Control

We prove a uniform bound on the noise coefficients and then use it to verify the noise stability hypothesis on learnable samples.

**Lemma F.9** (Uniform Noise Coefficient Bound). On the event $\mathcal{E}$ defined in Lemma E.1, for any iteration $t \leq T$, all filters $r \in [m]$, and all noise patches $(i,j)$, the noise coefficient is uniformly bounded by

$$\rho_{i,j,r}^{(t)} \leq \frac{10 \log^{1/3} T}{\sigma_n^2 d}. \tag{145}$$

*Proof of Lemma F.9.* Define the threshold

$$\rho_{\text{th}} := \frac{10 \log^{1/3} T}{\sigma_n^2 d}. \tag{146}$$

Assume for contradiction that $t \leq T$ is the first iteration where the bound is violated. Then there exist a specific noise patch $(i,j)$ and a specific filter $r$ such that

$$\rho_{i,j,r}^{(t)} > \rho_{\text{th}}. \tag{147}$$

By definition of $t$ being the first such iteration, for all preceding steps $\tau < t$,

$$\max_{k,q,r'} \rho_{k,q,r'}^{(\tau)} \leq \rho_{\text{th}}. \tag{148}$$

At initialization, $\rho_{i,j,r}^{(0)} = 0 < 0.5\rho_{\text{th}}$. Thus, there exists a last iteration $t_0 < t$ such that $\rho_{i,j,r}^{(t_0)} \leq 0.5\rho_{\text{th}}$. Consequently, for all $\tau \in [t_0 + 1, t - 1]$,

$$\rho_{i,j,r}^{(\tau)} > 0.5\rho_{\text{th}}. \tag{149}$$

At iteration $t_0$, by the decomposition in Lemma F.2, the triangle inequality and the nonnegativity of the noise coefficients give

$$|\langle \mathbf{w}_r^{(t_0)}, \mathbf{x}_{i,j} \rangle| \leq |\langle \mathbf{w}_r^{(0)}, \mathbf{x}_{i,j} \rangle| + \rho_{i,j,r}^{(t_0)} \|\mathbf{x}_{i,j}\|_2^2 + \sum_{(k,q) \neq (i,j)} \rho_{k,q,r}^{(t_0)} |\langle \mathbf{x}_{k,q}, \mathbf{x}_{i,j} \rangle|. \tag{150}$$

On the event $\mathcal{E}$, the bounds (P1), (P2), and (P6) imply

$$\|\mathbf{x}_{i,j}\|_2^2 \leq \frac{3}{2}\sigma_n^2 d, \quad |\langle \mathbf{x}_{k,q}, \mathbf{x}_{i,j} \rangle| \leq 2\sigma_n^2 \sqrt{d \log\left(\frac{16N^2P^2}{\delta}\right)}, \quad \text{and} \quad \left|\langle \mathbf{w}_r^{(0)}, \mathbf{x}_{i,j} \rangle\right| \leq 2\sigma_0 \sigma_n \sqrt{d \log\left(\frac{16NmP}{\delta}\right)}. \tag{151}$$

Substituting these bounds, together with $\max_{k,q,r'} \rho_{k,q,r'}^{(t_0)} \leq \rho_{\text{th}}$, into the preceding decomposition, we obtain

$$|\langle \mathbf{w}_r^{(t_0)}, \mathbf{x}_{i,j} \rangle| \leq 2\sigma_0 \sigma_n \sqrt{d \log\left(\frac{16NmP}{\delta}\right)} + \rho_{\text{th}} \sigma_n^2 d \left(\frac{3}{2} + \frac{2NP}{\sqrt{d}} \sqrt{\log\left(\frac{16N^2P^2}{\delta}\right)}\right). \tag{152}$$

By condition (C2), the cross-patch interference term is sufficiently small, and by condition (C4), the initialization term is strictly dominated by the threshold term. Since $\rho_{\text{th}} \sigma_n^2 d = 10 \log^{1/3} T$, the total inner product is bounded by

$$|\langle \mathbf{w}_r^{(t_0)}, \mathbf{x}_{i,j} \rangle| \leq 16 \log^{1/3} T. \tag{153}$$

Thus, using Lemma F.2,

$$\rho_{i,j,r}^{(t_0+1)} - \rho_{i,j,r}^{(t_0)} \leq \frac{3\eta}{N} \langle \mathbf{w}_r^{(t_0)}, \mathbf{x}_{i,j} \rangle^2 \leq \frac{3\eta}{N} \left(16 \log^{1/3} T\right)^2 = \frac{768\eta \log^{2/3} T}{N}. \tag{154}$$

By condition (C5), this single-step increment is strictly bounded by $0.1\rho_{\text{th}}$. Hence,

$$\rho_{i,j,r}^{(t_0+1)} \leq 0.5\rho_{\text{th}} + 0.1\rho_{\text{th}} = 0.6\rho_{\text{th}}. \tag{155}$$

Now consider any subsequent iteration $\tau \in [t_0 + 1, t - 1]$. Since

$$\rho_{i,j,r}^{(\tau)} > 0.5\rho_{\text{th}} = \frac{5\log^{1/3} T}{\sigma_n^2 d}, \tag{156}$$

we show that the margin $y_i f_{\mathbf{W}^{(\tau)}}(\tilde{\mathbf{X}}_i^{(\tau)})$ is already large. We begin by separating the target noise term $(r, j)$ from all remaining terms. Using the signal contribution bound from monotonicity of the signal coordinates and (P5), we have

$$y_i f_{\mathbf{W}^{(\tau)}}(\tilde{\mathbf{X}}_i^{(\tau)}) \geq -(\alpha - \epsilon)^3 m \left(\sigma_0 \sqrt{2\log\left(\frac{16m}{\delta}\right)}\right)^3 + \sum_{(r',q)\neq(r,j)} y_i \langle \mathbf{w}_{r'}^{(\tau)}, \mathbf{x}_{i,q}\rangle^3 + y_i \langle \mathbf{w}_r^{(\tau)}, \mathbf{x}_{i,j}\rangle^3. \tag{157}$$

We first control the non-target terms. For each $(r', q) \neq (r, j)$, Lemma F.2 gives

$$y_i \langle \mathbf{w}_{r'}^{(\tau)}, \mathbf{x}_{i,q}\rangle \geq -\left|\langle \mathbf{w}_{r'}^{(0)}, \mathbf{x}_{i,q}\rangle\right| - \sum_{(k,p)\neq(i,q)} \rho_{k,p,r'}^{(\tau)} \left|\langle \mathbf{x}_{k,p}, \mathbf{x}_{i,q}\rangle\right|. \tag{158}$$

Since $\max_{k,p,r'} \rho_{k,p,r'}^{(\tau)} \leq \rho_{\text{th}}$, using (P2) and (P6), we obtain

$$\begin{aligned}
y_i \langle \mathbf{w}_{r'}^{(\tau)}, \mathbf{x}_{i,q}\rangle &\geq -2\sigma_0 \sigma_n \sqrt{d\log\left(\frac{16NmP}{\delta}\right)} - 2NP\rho_{\text{th}}\sigma_n^2 \sqrt{d\log\left(\frac{16N^2P^2}{\delta}\right)} \\
&\geq -\frac{C^{-1}\log^{1/3} T}{m^{1/3} P^{1/3}},
\end{aligned} \tag{159}$$

where the last inequality follows from conditions (C2) and (C4). This implies

$$y_i \langle \mathbf{w}_{r'}^{(\tau)}, \mathbf{x}_{i,q}\rangle^3 \geq -\frac{C^{-1}\log T}{mP} \tag{160}$$

for every $(r', q) \neq (r, j)$. Summing over all non-target pairs, we get

$$\sum_{(r',q)\neq(r,j)} y_i \langle \mathbf{w}_{r'}^{(\tau)}, \mathbf{x}_{i,q}\rangle^3 \geq -C^{-1}\log T. \tag{161}$$

We next lower-bound the target term. By Lemma F.2,

$$y_i \langle \mathbf{w}_r^{(\tau)}, \mathbf{x}_{i,j}\rangle \geq \rho_{i,j,r}^{(\tau)} \|\mathbf{x}_{i,j}\|_2^2 - \sum_{(k,q)\neq(i,j)} \rho_{k,q,r}^{(\tau)} \left|\langle \mathbf{x}_{k,q}, \mathbf{x}_{i,j}\rangle\right| - \left|\langle \mathbf{w}_r^{(0)}, \mathbf{x}_{i,j}\rangle\right|. \tag{162}$$

Using $\max_{k,q,r'} \rho_{k,q,r'}^{(\tau)} \leq \rho_{\text{th}}$, together with (P1), (P2), and (P6), we obtain

$$\begin{aligned}
y_i \langle \mathbf{w}_r^{(\tau)}, \mathbf{x}_{i,j}\rangle &\geq \rho_{i,j,r}^{(\tau)}\left(\frac{1}{2}\sigma_n^2 d\right) - 2NP\rho_{\text{th}}\sigma_n^2 \sqrt{d\log\left(\frac{16N^2P^2}{\delta}\right)} - 2\sigma_0\sigma_n \sqrt{d\log\left(\frac{16NmP}{\delta}\right)} \\
&\geq \rho_{i,j,r}^{(\tau)}\left(\frac{1}{2}\sigma_n^2 d\right) - \rho_{i,j,r}^{(\tau)}\left(\frac{1}{2}\sigma_n^2 d\right)\left(\frac{8NP}{\sqrt{d}}\sqrt{\log\left(\frac{16N^2P^2}{\delta}\right)} + \frac{8\sigma_0\sigma_n\sqrt{d\log\left(\frac{16NmP}{\delta}\right)}}{\rho_{\text{th}}\sigma_n^2 d}\right) \\
&= \rho_{i,j,r}^{(\tau)}\left(\frac{1}{2}\sigma_n^2 d\right)\left(1 - \frac{8NP}{\sqrt{d}}\sqrt{\log\left(\frac{16N^2P^2}{\delta}\right)} - \frac{8\sigma_0\sigma_n\sqrt{d\log\left(\frac{16NmP}{\delta}\right)}}{10\log^{1/3} T}\right).
\end{aligned} \tag{163}$$

By conditions (C2) and (C4), the quantity in parentheses is at least $3/5$. Therefore,

$$y_i \langle \mathbf{w}_r^{(\tau)}, \mathbf{x}_{i,j}\rangle \geq \frac{3}{10}\rho_{i,j,r}^{(\tau)}\sigma_n^2 d. \tag{164}$$

Combining the signal bound, the non-target bound, and the target lower bound, we obtain

$$y_i f_{\mathbf{W}^{(\tau)}}(\tilde{\mathbf{X}}_i^{(\tau)}) \geq -(\alpha - \epsilon)^3 m \left( \sigma_0 \sqrt{2 \log\left(\frac{16m}{\delta}\right)} \right)^3 - C^{-1} \log T + \left( \frac{3}{10} \rho_{i,j,r}^{(\tau)} \sigma_n^2 d \right)^3$$

$$\geq \frac{27}{8} \log T - (\alpha - \epsilon)^3 m \left( \sigma_0 \sqrt{2 \log\left(\frac{16m}{\delta}\right)} \right)^3 - C^{-1} \log T, \tag{165}$$

where the last inequality uses $\rho_{i,j,r}^{(\tau)} > 0.5 \rho_{\text{th}} = \frac{5 \log^{1/3} T}{\sigma_n^2 d}$.

By condition (C4) and the preceding non-target bound, the last two terms are at most $C^{-1} \log T$. Choosing $C$ sufficiently large, we obtain

$$y_i f_{\mathbf{W}^{(\tau)}}(\tilde{\mathbf{X}}_i^{(\tau)}) \geq 3 \log T. \tag{166}$$

Since the negative sigmoid function satisfies $\psi(z) \leq \exp(-z)$, it follows that

$$\psi\left( y_i f_{\mathbf{W}^{(\tau)}}(\tilde{\mathbf{X}}_i^{(\tau)}) \right) \leq \exp(-3 \log T) = T^{-3}. \tag{167}$$

Since $\tau < t$, we still have $\max_{k,q,r'} \rho_{k,q,r'}^{(\tau)} \leq \rho_{\text{th}}$. Therefore, summing the update rule over $\tau \in [t_0 + 1, t - 1]$ gives

$$\sum_{\tau=t_0+1}^{t-1} \left( \rho_{i,j,r}^{(\tau+1)} - \rho_{i,j,r}^{(\tau)} \right) \leq T \cdot \frac{3\eta}{N} \left( 16 \log^{1/3} T \right)^2 \cdot T^{-3} = \frac{768 \eta \log^{2/3} T}{NT^2}. \tag{168}$$

By condition (C5), the right-hand side is bounded by $0.1 \rho_{\text{th}}$. Thus,

$$\rho_{i,j,r}^{(t)} = \rho_{i,j,r}^{(t_0+1)} + \sum_{\tau=t_0+1}^{t-1} \left( \rho_{i,j,r}^{(\tau+1)} - \rho_{i,j,r}^{(\tau)} \right) \leq 0.6 \rho_{\text{th}} + 0.1 \rho_{\text{th}} = 0.7 \rho_{\text{th}} < \rho_{\text{th}}. \tag{169}$$

This contradicts the choice of $t$. Therefore the uniform noise coefficient bound holds for all $i, j, r$ and all $t \leq T$. $\qquad \square$

**Lemma F.10** (Verification of Noise Stability on Learnable Samples). This lemma verifies Hypothesis F.3. Fix any iteration $t \leq T + 1$, and define the unlearnable noise coefficient maximum at this iteration by

$$\rho_{\max}^{\mathcal{S}_U} := \max_{k \in \mathcal{S}_U, \, q \neq s(\mathbf{X}_k), \, r \in [m]} \rho_{k,q,r}^{(t)}. \tag{170}$$

On the event $\mathcal{E}$ defined in Lemma E.1, for every learnable sample $i \in \mathcal{S}_L$, every noise patch $j \neq s(\mathbf{X}_i)$, and every filter $r \in [m]$, the learnable noise coefficient satisfies

$$\rho_{i,j,r}^{(t)} \leq \frac{\sigma_0}{\sigma_n \sqrt{d}} + \frac{p_{\text{un}} N \sigma_n^2 \sqrt{d}}{100 \log^{2/3} T} \left( \rho_{\max}^{\mathcal{S}_U} \right)^2. \tag{171}$$

Consequently, by Lemma F.9, for every learnable sample $i \in \mathcal{S}_L$, every noise patch $j \neq s(\mathbf{X}_i)$, every filter $r \in [m]$, and every iteration $t \leq T + 1$, the noise inner product satisfies

$$\left| \langle \mathbf{w}_r^{(t)}, \mathbf{x}_{i,j} \rangle \right| \leq 3 \sigma_0 \sigma_n \sqrt{d \log\left(\frac{16 N m P}{\delta}\right)} + \frac{21 p_{\text{un}} N P \log^{1/3} T}{\sqrt{d}} \sqrt{\log\left(\frac{16 N^2 P^2}{\delta}\right)}. \tag{172}$$

*Proof of Lemma F.10.* Since the noise coefficients are non-decreasing from Lemma F.2, for every $\tau \leq t, k \in \mathcal{S}_U, q \neq s(\mathbf{X}_k)$, and $r \in [m]$,

$$\rho_{k,q,r}^{(\tau)} \leq \rho_{\max}^{\mathcal{S}_U}. \tag{173}$$

Moreover, by Lemma F.9,

$$\rho_{\max}^{\mathcal{S}_U} \leq \frac{10 \log^{1/3} T}{\sigma_n^2 d}. \tag{174}$$

Define the threshold

$$
\begin{aligned}
\rho_{\text{th}} := {} & 100 \exp(2C_0^3) \log\left(\frac{16NmP}{\delta}\right) \cdot \frac{\sigma_0 \sigma_n^2 d}{N(\alpha - \epsilon)^3(1 - p_{\text{un}})} \\
& + \frac{18m^{2/3}\log^{1/3}T}{C_0^2\left(1 - \frac{\epsilon}{\alpha}\right)^3 N\alpha^2}\left(18\sigma_0^2\sigma_n^2 d\log\left(\frac{16NmP}{\delta}\right) + 18p_{\text{un}}^2 N^2 P^2 \left(\rho_{\max}^{\mathcal{S}_U}\right)^2 \sigma_n^4 d\log\left(\frac{16N^2P^2}{\delta}\right)\right).
\end{aligned}
\tag{175}
$$

Using conditions (C6) and (C7), for a sufficiently large constant $C$, the threshold expansion satisfies

$$
\begin{aligned}
\rho_{\text{th}} \leq {} & C\log\left(\frac{16NmP}{\delta}\right)\frac{\sigma_0\sigma_n^2 d}{N\alpha^3} + \frac{Cm^{2/3}\sigma_0^2\sigma_n^2 d\log^{1/3}T\log\left(\frac{16NmP}{\delta}\right)}{N\alpha^2} \\
& + \frac{Cp_{\text{un}}^2 NP^2 m^{2/3}\left(\rho_{\max}^{\mathcal{S}_U}\right)^2 \sigma_n^4 d\log^{1/3}T\log\left(\frac{16N^2P^2}{\delta}\right)}{\alpha^2}.
\end{aligned}
\tag{176}
$$

Applying condition (C4), the second term is absorbed into the first term. Hence

$$
\rho_{\text{th}} \leq 2C\log\left(\frac{16NmP}{\delta}\right)\frac{\sigma_0\sigma_n^2 d}{N\alpha^3} + C\frac{p_{\text{un}}^2 NP^2 m^{2/3}\left(\rho_{\max}^{\mathcal{S}_U}\right)^2 \sigma_n^4 d\log^{1/3}T\log\left(\frac{16N^2P^2}{\delta}\right)}{\alpha^2}.
\tag{177}
$$

Next, we bound both terms. For the first term, it is enough that $\alpha^3 \geq C\frac{\sigma_n^3 d^{3/2}\log\left(\frac{16NmP}{\delta}\right)}{N}$, which is implied by condition (C3). Hence

$$
2C\log\left(\frac{16NmP}{\delta}\right)\frac{\sigma_0\sigma_n^2 d}{N\alpha^3} \leq \frac{\sigma_0}{\sigma_n d^{1/2}}.
\tag{178}
$$

For the second term, using condition (C3) first gives

$$
C\frac{p_{\text{un}}^2 NP^2 m^{2/3}\left(\rho_{\max}^{\mathcal{S}_U}\right)^2 \sigma_n^4 d\log^{1/3}T\log\left(\frac{16N^2P^2}{\delta}\right)}{\alpha^2} \leq C\frac{p_{\text{un}}^2 N^{5/3}P^2 m^{2/3}\left(\rho_{\max}^{\mathcal{S}_U}\right)^2 \sigma_n^2 \log^{1/3}T\log\left(\frac{16N^2P^2}{\delta}\right)}{d}.
\tag{179}
$$

By condition (C2), the last term is bounded by

$$
C\frac{p_{\text{un}}^2 N^{5/3}P^2 m^{2/3}\left(\rho_{\max}^{\mathcal{S}_U}\right)^2 \sigma_n^2 \log^{1/3}T\log\left(\frac{16N^2P^2}{\delta}\right)}{d} \leq \frac{p_{\text{un}} N\sigma_n^2\sqrt{d}}{100\log^{2/3}T}\left(\rho_{\max}^{\mathcal{S}_U}\right)^2.
\tag{180}
$$

Combining the two bounds gives

$$
\rho_{\text{th}} \leq \frac{\sigma_0}{\sigma_n\sqrt{d}} + \frac{p_{\text{un}} N\sigma_n^2\sqrt{d}}{100\log^{2/3}T}\left(\rho_{\max}^{\mathcal{S}_U}\right)^2.
\tag{181}
$$

We now prove by induction on $t = 0, 1, \ldots, T$ that

$$
\rho_{i,j,r}^{(t+1)} \leq \rho_{\text{th}} \qquad \text{for all } i \in \mathcal{S}_L, \; j \neq s(\mathbf{X}_i), \; r \in [m].
\tag{182}
$$

Fix any $t \in [0, T]$ and assume that the claim holds for all iterations up to $t$.

If $t + 1 \leq T_0$, then Lemma F.6 directly gives

$$
\rho_{i,j,r}^{(t+1)} \leq 100\exp(2C_0^3)\log\left(\frac{16NmP}{\delta}\right)\cdot\frac{\sigma_0\sigma_n^2 d}{N(\alpha - \epsilon)^3(1 - p_{\text{un}})} \leq \rho_{\text{th}}
\tag{183}
$$

for every $i \in \mathcal{S}_L$, every $j \neq s(\mathbf{X}_i)$, and every $r \in [m]$.

It therefore remains to consider the regime $t + 1 > T_0$. In this case, for every $\tau \in [T_0, t]$, the induction hypothesis yields

$$
\rho_{i,j,r}^{(\tau)} \leq \rho_{\text{th}} \qquad \text{for all } i \in \mathcal{S}_L, \; j \neq s(\mathbf{X}_i), \; r \in [m].
\tag{184}
$$

Fix any $\tau \in [T_0, t]$, any learnable sample $i \in \mathcal{S}_L$, any noise patch $j \neq s(\mathbf{X}_i)$, and any filter $r \in [m]$. By the decomposition in Lemma F.2, the triangle inequality and the nonnegativity of the noise coefficients give

$$\left|\langle \mathbf{w}_r^{(\tau)}, \mathbf{x}_{i,j}\rangle\right| \leq \left|\langle \mathbf{w}_r^{(0)}, \mathbf{x}_{i,j}\rangle\right| + \rho_{i,j,r}^{(\tau)}\|\mathbf{x}_{i,j}\|_2^2 + \sum_{\substack{k \in \mathcal{S}_L, \, q \neq s(\mathbf{X}_k) \\ (k,q) \neq (i,j)}} \rho_{k,q,r}^{(\tau)} |\langle \mathbf{x}_{k,q}, \mathbf{x}_{i,j}\rangle| + \sum_{k \in \mathcal{S}_U, \, q \neq s(\mathbf{X}_k)} \rho_{k,q,r}^{(\tau)} |\langle \mathbf{x}_{k,q}, \mathbf{x}_{i,j}\rangle|.$$

(185)

Using $\rho_{k,q,r}^{(\tau)} \leq \rho_{\text{th}}$ for $k \in \mathcal{S}_L$, $\rho_{k,q,r}^{(\tau)} \leq \rho_{\max}^{\mathcal{S}_U}$ for $k \in \mathcal{S}_U$, and the event bounds (P1), (P2), and (P6), we bound the inner product:

$$\left|\langle \mathbf{w}_r^{(\tau)}, \mathbf{x}_{i,j}\rangle\right| \leq \left|\langle \mathbf{w}_r^{(0)}, \mathbf{x}_{i,j}\rangle\right| + \rho_{\text{th}}\sigma_n^2 d\left(\frac{3}{2} + \frac{2NP}{\sqrt{d}}\sqrt{\log\left(\frac{16N^2P^2}{\delta}\right)}\right) + 2p_{\text{un}}NP\rho_{\max}^{\mathcal{S}_U}\sigma_n^2\sqrt{d\log\left(\frac{16N^2P^2}{\delta}\right)}$$

$$\leq 2\sigma_0\sigma_n\sqrt{d\log\left(\frac{16NmP}{\delta}\right)} + 2\rho_{\text{th}}\sigma_n^2 d + 2p_{\text{un}}NP\rho_{\max}^{\mathcal{S}_U}\sigma_n^2\sqrt{d\log\left(\frac{16N^2P^2}{\delta}\right)},$$

(186)

where the second inequality holds under condition (C2).

Using (181) and (174), we have

$$2\rho_{\text{th}}\sigma_n^2 d \leq 2\sigma_0\sigma_n\sqrt{d} + \frac{2p_{\text{un}}N\sigma_n^4 d^{3/2}}{100\log^{2/3}T}\left(\rho_{\max}^{\mathcal{S}_U}\right)^2$$

$$\leq 2\sigma_0\sigma_n\sqrt{d} + p_{\text{un}}NP\rho_{\max}^{\mathcal{S}_U}\sigma_n^2\sqrt{d\log\left(\frac{16N^2P^2}{\delta}\right)}.$$

(187)

Substituting this bound into the preceding display gives

$$\left|\langle \mathbf{w}_r^{(\tau)}, \mathbf{x}_{i,j}\rangle\right| \leq 2\sigma_0\sigma_n\sqrt{d\log\left(\frac{16NmP}{\delta}\right)} + 2\sigma_0\sigma_n\sqrt{d} + 3p_{\text{un}}NP\rho_{\max}^{\mathcal{S}_U}\sigma_n^2\sqrt{d\log\left(\frac{16N^2P^2}{\delta}\right)}$$

$$\leq 3\sigma_0\sigma_n\sqrt{d\log\left(\frac{16NmP}{\delta}\right)} + 3p_{\text{un}}NP\rho_{\max}^{\mathcal{S}_U}\sigma_n^2\sqrt{d\log\left(\frac{16N^2P^2}{\delta}\right)}.$$

(188)

Applying $(a+b)^2 \leq 2a^2 + 2b^2$, we square this inner product to bound the update magnitude:

$$\langle \mathbf{w}_r^{(\tau)}, \mathbf{x}_{i,j}\rangle^2 \leq 18\sigma_0^2\sigma_n^2 d\log\left(\frac{16NmP}{\delta}\right) + 18p_{\text{un}}^2N^2P^2\left(\rho_{\max}^{\mathcal{S}_U}\right)^2\sigma_n^4 d\log\left(\frac{16N^2P^2}{\delta}\right) := W_{\text{th}}^2. \qquad (189)$$

Summing the update rule from $\tau = T_0 + 1$ to $t$ and first substituting the uniform bound on $\langle \mathbf{w}_r^{(\tau)}, \mathbf{x}_{i,j}\rangle^2$, we obtain

$$\rho_{i,j,r}^{(t+1)} \leq \rho_{i,j,r}^{(T_0)} + \sum_{\tau=T_0+1}^{t} \frac{3\eta}{N}\psi\left(y_i f_{\mathbf{W}^{(\tau)}}(\tilde{\mathbf{X}}_i^{(\tau)})\right)\langle \mathbf{w}_r^{(\tau)}, \mathbf{x}_{i,j}\rangle^2$$

$$\leq \rho_{i,j,r}^{(T_0)} + \frac{3\eta}{N}W_{\text{th}}^2\sum_{\tau=T_0+1}^{t}\psi\left(y_i f_{\mathbf{W}^{(\tau)}}(\tilde{\mathbf{X}}_i^{(\tau)})\right).$$

(190)

Applying the cumulative and pointwise bounds from Lemma F.8, we first control the sum of negative sigmoid factors:

$$\sum_{\tau=T_0+1}^{t}\psi\left(y_i f_{\mathbf{W}^{(\tau)}}(\tilde{\mathbf{X}}_i^{(\tau)})\right) \leq \sum_{\tau=T_0}^{t-1}\psi\left(y_i f_{\mathbf{W}^{(\tau)}}(\tilde{\mathbf{X}}_i^{(\tau)})\right) + \psi\left(y_i f_{\mathbf{W}^{(t)}}(\tilde{\mathbf{X}}_i^{(t)})\right)$$

$$\leq \frac{2m^{2/3}\log^{1/3}T}{C_0^2\left(1 - \frac{\epsilon}{\alpha}\right)^3\eta\alpha^2} + \frac{4m^{2/3}\log^{1/3}T}{C_0^2\left(1 - \frac{\epsilon}{\alpha}\right)^3\eta\alpha^2(t - T_0)}$$

$$\leq \frac{6m^{2/3}\log^{1/3}T}{C_0^2\left(1 - \frac{\epsilon}{\alpha}\right)^3\eta\alpha^2}.$$

(191)

Substituting this bound into the accumulated update gives

$$
\begin{aligned}
\rho_{i,j,r}^{(t+1)} &\leq \rho_{i,j,r}^{(T_0)} + \frac{3\eta}{N} W_{\text{th}}^2 \sum_{\tau=T_0+1}^{t} \psi\left(y_i f_{\mathbf{W}^{(\tau)}}(\tilde{\mathbf{X}}_i^{(\tau)})\right) \\
&\leq \rho_{i,j,r}^{(T_0)} + \frac{18m^{2/3}\log^{1/3} T}{C_0^2\left(1-\frac{\epsilon}{\alpha}\right)^3 N\alpha^2} W_{\text{th}}^2.
\end{aligned}
\tag{192}
$$

Finally, using the phase-one bound from Lemma F.6 and the definition of $W_{\text{th}}^2$ in (189), we obtain

$$
\begin{aligned}
\rho_{i,j,r}^{(t+1)} &\leq 100\exp(2C_0^3)\log\left(\frac{16NmP}{\delta}\right)\cdot \frac{\sigma_0\sigma_n^2 d}{N(\alpha-\epsilon)^3(1-p_{\text{un}})} \\
&\quad + \frac{18m^{2/3}\log^{1/3} T}{C_0^2\left(1-\frac{\epsilon}{\alpha}\right)^3 N\alpha^2}\left(18\sigma_0^2\sigma_n^2 d\log\left(\frac{16NmP}{\delta}\right) + 18p_{\text{un}}^2 N^2 P^2 \left(\rho_{\max}^{\mathcal{S}_U}\right)^2 \sigma_n^4 d\log\left(\frac{16N^2P^2}{\delta}\right)\right) \\
&= \rho_{\text{th}}.
\end{aligned}
\tag{193}
$$

This proves $\rho_{i,j,r}^{(t+1)} \leq \rho_{\text{th}}$. Therefore, by (181),

$$
\rho_{i,j,r}^{(t)} \leq \frac{\sigma_0}{\sigma_n\sqrt{d}} + \frac{p_{\text{un}} N\sigma_n^2\sqrt{d}}{100\log^{2/3} T}\left(\rho_{\max}^{\mathcal{S}_U}\right)^2.
\tag{194}
$$

This proves (171).

By (174), (171) gives

$$
\rho_{i,j,r}^{(t)} \leq \frac{\sigma_0}{\sigma_n\sqrt{d}} + \frac{p_{\text{un}} N}{\sigma_n^2 d^{3/2}}.
\tag{195}
$$

Moreover, substituting (174) into the unlearnable contribution in (186) gives

$$
2p_{\text{un}} NP\rho_{\max}^{\mathcal{S}_U}\sigma_n^2\sqrt{d\log\left(\frac{16N^2P^2}{\delta}\right)} \leq \frac{20p_{\text{un}} NP\log^{1/3} T}{\sqrt{d}}\sqrt{\log\left(\frac{16N^2P^2}{\delta}\right)}.
\tag{196}
$$

Applying these bounds to the learnable and unlearnable coefficient terms in (186), respectively, we obtain

$$
\begin{aligned}
\left|\langle\mathbf{w}_r^{(t)}, \mathbf{x}_{i,j}\rangle\right| &\leq 2\sigma_0\sigma_n\sqrt{d\log\left(\frac{16NmP}{\delta}\right)} + 2\sigma_0\sigma_n\sqrt{d} + \frac{2p_{\text{un}} N}{\sqrt{d}} + \frac{20p_{\text{un}} NP\log^{1/3} T}{\sqrt{d}}\sqrt{\log\left(\frac{16N^2P^2}{\delta}\right)} \\
&\leq 3\sigma_0\sigma_n\sqrt{d\log\left(\frac{16NmP}{\delta}\right)} + \frac{21p_{\text{un}} NP\log^{1/3} T}{\sqrt{d}}\sqrt{\log\left(\frac{16N^2P^2}{\delta}\right)}.
\end{aligned}
\tag{197}
$$

This proves Hypothesis F.3. $\qquad\square$

# G. The Dichotomy of Adversarial Training

In this section, we derive the dichotomy of adversarial training by combining the results from the previous section. In the learnable-only regime, the noise responses remain stable after the signal is learned, so adversarial training generalizes through the learnable feature. In contrast, in the sparse-unlearnable regime, the residual gradients on unlearnable samples drive the model to memorize sample-specific noise, which eventually undermines robust generalization. Together, these two cases yield the main dichotomy theorem for adversarial training.

## G.1. Adversarial Training without Unlearnable Samples

We first consider the learnable-only regime. In this case, the learned signal remains stable and the noise responses stay at their initialization scale throughout training. As a result, adversarial training continues to classify through the learnable feature even under adversarial perturbations, yielding robust generalization on the learnable-sample distribution.

**Lemma G.1** (Noise Stability without Unlearnable Samples). We formalize the noise-stability statement in Lemma 4.10. On the event $\mathcal{E}$ defined in Lemma E.1, assume the learnable-only regime $p_{\mathrm{un}} = 0$ (equivalently, $\mathcal{S}_U = \emptyset$). Then, for every iteration $t \leq T$, the noise coefficients satisfy

$$\max_{i \in [N],\, j \neq s(\mathbf{X}_i),\, r \in [m]} \rho_{i,j,r}^{(t)} \leq 200 \exp(2C_0^3) \log\left(\frac{16NmP}{\delta}\right) \cdot \frac{\sigma_0 \sigma_n^2 d}{N(\alpha - \epsilon)^3}. \tag{198}$$

Consequently, for every $i \in [N]$, $j \neq s(\mathbf{X}_i)$, $r \in [m]$, and $t \leq T$,

$$\left| \langle \mathbf{w}_r^{(t)}, \mathbf{x}_{i,j} \rangle \right| \leq 3\sigma_0 \sigma_n \sqrt{d \log\left(\frac{16NmP}{\delta}\right)}. \tag{199}$$

*Proof of Lemma G.1.* If $t \leq T_0$, the claim follows directly from Lemma F.6. Thus it remains to consider $t \in (T_0, T]$.

Define the threshold

$$\rho_{\mathrm{th}} := \max_{i,j,r} \rho_{i,j,r}^{(T_0)} + \frac{30 m^{2/3} \log\left(\frac{16NmP}{\delta}\right) \log^{1/3} T \sigma_0^2 \sigma_n^2 d}{C_0^2 \left(1 - \frac{\epsilon}{\alpha}\right)^3 N\alpha^2}. \tag{200}$$

By Lemma F.6, we have

$$\max_{i,j,r} \rho_{i,j,r}^{(T_0)} \leq 100 \exp(2C_0^3) \log\left(\frac{16NmP}{\delta}\right) \cdot \frac{\sigma_0 \sigma_n^2 d}{N(\alpha - \epsilon)^3}. \tag{201}$$

Therefore,

$$\begin{aligned}
\rho_{\mathrm{th}} &\leq 100 \exp(2C_0^3) \log\left(\frac{16NmP}{\delta}\right) \cdot \frac{\sigma_0 \sigma_n^2 d}{N(\alpha - \epsilon)^3} + \frac{30 m^{2/3} \log\left(\frac{16NmP}{\delta}\right) \log^{1/3} T \sigma_0^2 \sigma_n^2 d}{C_0^2 \left(1 - \frac{\epsilon}{\alpha}\right)^3 N\alpha^2} \\
&= \log\left(\frac{16NmP}{\delta}\right) \cdot \frac{\sigma_0 \sigma_n^2 d}{N(\alpha - \epsilon)^3} \left( 100 \exp(2C_0^3) + \frac{30 m^{2/3} \alpha \sigma_0 \log^{1/3} T}{C_0^2} \right) \\
&\leq 200 \exp(2C_0^3) \log\left(\frac{16NmP}{\delta}\right) \cdot \frac{\sigma_0 \sigma_n^2 d}{N(\alpha - \epsilon)^3},
\end{aligned} \tag{202}$$

where the last inequality follows from condition (C4).

Assume for contradiction that there exists an iteration $t \in (T_0, T]$ such that $t$ is the first iteration where

$$\max_{i,j,r} \rho_{i,j,r}^{(t)} > \rho_{\mathrm{th}}. \tag{203}$$

Then, by the minimality of $t$, we have $\rho_{i,j,r}^{(\tau)} \leq \rho_{\mathrm{th}}$ for all $\tau \in [T_0, t - 1]$ and all $i, j, r$.

Fix any iteration $\tau < t$. By the decomposition in Lemma F.2, the triangle inequality and the nonnegativity of the noise coefficients give

$$\left| \langle \mathbf{w}_r^{(\tau)}, \mathbf{x}_{i,j} \rangle \right| \leq \left| \langle \mathbf{w}_r^{(0)}, \mathbf{x}_{i,j} \rangle \right| + \rho_{i,j,r}^{(\tau)} \|\mathbf{x}_{i,j}\|_2^2 + \sum_{\substack{k,q \\ (k,q) \neq (i,j)}} \rho_{k,q,r}^{(\tau)} \left| \langle \mathbf{x}_{k,q}, \mathbf{x}_{i,j} \rangle \right|. \tag{204}$$

On the event $\mathcal{E}$, the bounds (P1), (P2), and (P6) imply

$$\|\mathbf{x}_{i,j}\|_2^2 \leq \frac{3}{2} \sigma_n^2 d, \quad \left| \langle \mathbf{x}_{k,q}, \mathbf{x}_{i,j} \rangle \right| \leq 2\sigma_n^2 \sqrt{d \log\left(\frac{16N^2P^2}{\delta}\right)}, \quad \text{and} \quad \left| \langle \mathbf{w}_r^{(0)}, \mathbf{x}_{i,j} \rangle \right| \leq 2\sigma_0 \sigma_n \sqrt{d \log\left(\frac{16NmP}{\delta}\right)}. \tag{205}$$

Substituting these bounds together with $\rho_{k,q,r}^{(\tau)} \leq \rho_{\text{th}}$, we obtain

$$
\begin{aligned}
\left|\langle \mathbf{w}_r^{(\tau)}, \mathbf{x}_{i,j}\rangle\right| &\leq 2\sigma_0\sigma_n\sqrt{d\log\left(\frac{16NmP}{\delta}\right)} + \rho_{\text{th}}\cdot\frac{3}{2}\sigma_n^2 d + 2NP\rho_{\text{th}}\sigma_n^2\sqrt{d\log\left(\frac{16N^2P^2}{\delta}\right)} \\
&= 2\sigma_0\sigma_n\sqrt{d\log\left(\frac{16NmP}{\delta}\right)} + \rho_{\text{th}}\sigma_n^2 d\left(\frac{3}{2} + \frac{2NP}{\sqrt{d}}\sqrt{\log\left(\frac{16N^2P^2}{\delta}\right)}\right) \\
&\leq 2\sigma_0\sigma_n\sqrt{d\log\left(\frac{16NmP}{\delta}\right)} + 2\rho_{\text{th}}\sigma_n^2 d \\
&\leq \sqrt{5}\sigma_0\sigma_n\sqrt{d\log\left(\frac{16NmP}{\delta}\right)},
\end{aligned}
\tag{206}
$$

where the second inequality follows from condition (C2), and the final inequality follows from the bound on $\rho_{\text{th}}$ together with conditions (C3) and (C6).

With the inner product bounded, we next control the cumulative update after $T_0$. Since $p_{\text{un}} = 0$, every training sample is learnable, and hence Lemma F.8 applies to every $i \in [N]$. Therefore,

$$
\sum_{\tau=T_0}^{t-1} \psi\left(y_i f_{\mathbf{W}^{(\tau)}}(\tilde{\mathbf{X}}_i^{(\tau)})\right) \leq \frac{2m^{2/3}\log^{1/3}T}{C_0^2\left(1-\frac{\epsilon}{\alpha}\right)^3\eta\alpha^2}.
\tag{207}
$$

Summing the noise-coefficient update in Lemma F.2 from $\tau = T_0$ to $t-1$, and using the preceding inner-product bound, gives

$$
\begin{aligned}
\rho_{i,j,r}^{(t)} &= \rho_{i,j,r}^{(T_0)} + \sum_{\tau=T_0}^{t-1}\frac{3\eta}{N}\psi\left(y_i f_{\mathbf{W}^{(\tau)}}(\tilde{\mathbf{X}}_i^{(\tau)})\right)\langle \mathbf{w}_r^{(\tau)}, \mathbf{x}_{i,j}\rangle^2 \\
&\leq \max_{i,j,r}\rho_{i,j,r}^{(T_0)} + \frac{3\eta}{N}\sum_{\tau=T_0}^{t-1}\psi\left(y_i f_{\mathbf{W}^{(\tau)}}(\tilde{\mathbf{X}}_i^{(\tau)})\right)\cdot 5\sigma_0^2\sigma_n^2 d\log\left(\frac{16NmP}{\delta}\right) \\
&\leq \max_{i,j,r}\rho_{i,j,r}^{(T_0)} + \frac{30m^{2/3}\log\left(\frac{16NmP}{\delta}\right)\log^{1/3}T\sigma_0^2\sigma_n^2 d}{C_0^2\left(1-\frac{\epsilon}{\alpha}\right)^3 N\alpha^2}.
\end{aligned}
\tag{208}
$$

This contradicts the choice of $t$. Therefore, for all $i, j, r$ and all $t \in (T_0, T]$,

$$
\rho_{i,j,r}^{(t)} \leq \rho_{\text{th}} \leq 200\exp(2C_0^3)\log\left(\frac{16NmP}{\delta}\right)\cdot\frac{\sigma_0\sigma_n^2 d}{N(\alpha-\epsilon)^3}.
\tag{209}
$$

Combining this bound for $t \in (T_0, T]$ with Lemma F.6 for $t \leq T_0$ proves the claimed coefficient bound for all $t \leq T$.

It remains to record the corresponding noise-response bound. The inner-product estimate above used only the event bounds (P1), (P2), (P6), and a uniform coefficient bound by $\rho_{\text{th}}$. Therefore, after the coefficient bound has been established for all $t \leq T$, the same estimate gives, for every $i \in [N]$, $j \neq s(\mathbf{X}_i)$, $r \in [m]$, and $t \leq T$,

$$
\left|\langle \mathbf{w}_r^{(t)}, \mathbf{x}_{i,j}\rangle\right| \leq \sqrt{5}\sigma_0\sigma_n\sqrt{d\log\left(\frac{16NmP}{\delta}\right)} \leq 3\sigma_0\sigma_n\sqrt{d\log\left(\frac{16NmP}{\delta}\right)}.
\tag{210}
$$

This completes the proof. $\qquad\square$

**Theorem G.2** (Adversarial Training without Unlearnable Samples). *On the event $\mathcal{E}$ defined in Lemma E.1, assume that the requirements in Condition D.5 hold with (C7) replaced by $p_{\text{un}} = 0$ (equivalently, $\mathcal{S}_U = \emptyset$), and run AT with only learnable samples for $T$ iterations. Then the learned weights satisfy:*

1. *There exists at least one filter $r \in [m]$ whose first coordinate is aligned with the learnable feature:*

$$
w_{r,1}^{(T)} \geq \tilde{\Omega}(\alpha^{-1}).
\tag{211}
$$

2. *The network barely memorizes the noise features. More precisely, for every filter $r \in [m]$ and every noise patch $(i,j)$ with $i \in [N]$ and $j \neq s(\mathbf{X}_i)$,*

$$\left| \langle \mathbf{w}_r^{(T)}, \mathbf{x}_{i,j} \rangle \right| \leq \tilde{O}(\sigma_0 \sigma_n \sqrt{d}) \tag{212}$$

*Consequently, both the robust training error and robust test error are $o(1)$.*

*Proof of Theorem G.2.* **Step 1: Signal Alignment.** By Lemma F.5, after $T_0 \leq \frac{5 \exp(2C_0^3)}{\eta(\alpha-\epsilon)^3 \sigma_0}$ iterations, there exists at least one filter $r \in [m]$ such that $w_{r,1}^{(T_0)} \geq \frac{C_0}{\alpha m^{1/3}}$. Since the lower bound of the signal update in (53) is non-negative, $w_{r,1}^{(t)}$ is non-decreasing in $t$. Hence,

$$w_{r,1}^{(T)} \geq \frac{C_0}{\alpha m^{1/3}} \geq \tilde{\Omega}(\alpha^{-1}), \tag{213}$$

where the last inequality follows from condition (C8).

**Step 2: Noise Stability.** By Lemma G.1, for every $i \in [N]$, $j \neq s(\mathbf{X}_i)$, and $r \in [m]$, the noise response at time $T$ satisfies

$$\left| \langle \mathbf{w}_r^{(T)}, \mathbf{x}_{i,j} \rangle \right| \leq 3\sigma_0 \sigma_n \sqrt{d \log\left(\frac{16NmP}{\delta}\right)} \leq \widetilde{O}(\sigma_0 \sigma_n \sqrt{d}). \tag{214}$$

**Step 3: Training Error.** From the loss convergence guarantee in Lemma F.8, the average logistic loss satisfies:

$$\frac{1}{N} \sum_{i \in [N]} \ell\left(y_i f_{\mathbf{W}^{(T)}}(\tilde{\mathbf{X}}_i^{(T)})\right) \leq \frac{10}{N} \sum_{i \in [N]} \psi\left(y_i f_{\mathbf{W}^{(T)}}(\tilde{\mathbf{X}}_i^{(T)})\right) \leq \frac{40 m^{2/3} \log^{1/3} T}{C_0^2 \left(1 - \frac{\epsilon}{\alpha}\right)^3 \eta \alpha^2 (T - T_0)} \leq o(1), \tag{215}$$

where the first inequality follows from Lemma I.2, and the last inequality follows from (C1) and (C3).

**Step 4: Test Error.**

**Step 4.1: Noise norm bound.** We decompose the final weight into the signal and noise components:

$$\mathbf{w}_r^{(T)} = w_{r,1}^{(T)} \mathbf{e}_1 + \mathbf{z}_r^{(T)}, \qquad r \in [m], \tag{216}$$

where $\mathbf{z}_r^{(T)} \in \text{span}(\mathbf{e}_1)^{\perp}$. By the update rule decomposition,

$$\mathbf{z}_r^{(T)} = \mathbf{z}_r^{(0)} + \sum_{i=1}^{N} \sum_{j \neq s(\mathbf{X}_i)} \rho_{i,j,r}^{(T)} y_i \mathbf{x}_{i,j}, \tag{217}$$

and hence

$$\|\mathbf{z}_r^{(T)}\|_2 \leq \|\mathbf{z}_r^{(0)}\|_2 + \sum_{i=1}^{N} \sum_{j \neq s(\mathbf{X}_i)} \rho_{i,j,r}^{(T)} \|\mathbf{x}_{i,j}\|_2. \tag{218}$$

On the event $\mathcal{E}$, using (P1), (P4), and Lemma G.1, we obtain

$$\begin{aligned} \|\mathbf{z}_r^{(T)}\|_2 &\leq 2\sigma_0 \sqrt{d} + 300 P \exp(2C_0^3) \log\left(\frac{16NmP}{\delta}\right) \cdot \frac{\sigma_0 \sigma_n^3 d^{3/2}}{(\alpha - \epsilon)^3} \\ &\leq 2\sigma_0 \sqrt{d} + 300 NP \exp(2C_0^3) \sigma_0 \\ &\leq 3\sigma_0 \sqrt{d}, \end{aligned} \tag{219}$$

where the second inequality follows from conditions (C3) and (C6), and the last inequality follows from condition (C2). Therefore,

$$\|\mathbf{z}_r^{(T)}\|_1 \leq \sqrt{d} \|\mathbf{z}_r^{(T)}\|_2 \leq 3\sigma_0 d \qquad \text{for all } r \in [m]. \tag{220}$$

**Step 4.2: Tail reduction.** For a test sample $(\mathbf{X}, y) \sim \mathcal{D}_L$, let $\tilde{\mathbf{x}}_j = \mathbf{x}_j + \boldsymbol{\delta}_j$ with $\|\boldsymbol{\delta}_j\|_\infty \leq \epsilon$ for each non-signal patch $j \neq s(\mathbf{X})$. By Lemma F.1, the signal weights are monotonically non-decreasing, so $w_{r,1}^{(T)} \geq w_{r,1}^{(0)}$. On the event $\mathcal{E}$, (P5) gives

$$w_{r,1}^{(0)} \geq -\sigma_0 \sqrt{2 \log \left( \frac{16m}{\delta} \right)} \qquad \text{for all } r \in [m]. \tag{221}$$

Moreover, by Lemma F.5, there exists at least one filter $r \in [m]$ such that $w_{r,1}^{(T)} \geq \frac{C_0}{\alpha m^{1/3}}$. Therefore,

$$\alpha^3 \left( 1 - \frac{\epsilon}{\alpha} \right)^3 \sum_{r \in [m]} (w_{r,1}^{(T)})^3 \geq \alpha^3 \left( 1 - \frac{\epsilon}{\alpha} \right)^3 \left[ \left( \frac{C_0}{\alpha m^{1/3}} \right)^3 - m \left( \sigma_0 \sqrt{2 \log \left( \frac{16m}{\delta} \right)} \right)^3 \right]$$
$$\geq \frac{C_0^3}{2m} \left( 1 - \frac{\epsilon}{\alpha} \right)^3, \tag{222}$$

where the last inequality follows from condition (C4).

Using this signal margin, the robust loss is bounded by

$$\mathcal{L}_{\mathcal{D}_L}^{\mathrm{rob}}(f_{\mathbf{W}^{(T)}}) \leq \Pr_{(\mathbf{X},y) \sim \mathcal{D}_L} \left[ \frac{C_0^3}{2m} \left( 1 - \frac{\epsilon}{\alpha} \right)^3 + y \sum_{r \in [m]} \sum_{j \neq s(\mathbf{X})} \langle \mathbf{z}_r^{(T)}, \tilde{\mathbf{x}}_j \rangle^3 < 0 \right]. \tag{223}$$

Expanding the cubic term gives

$$\langle \mathbf{z}_r^{(T)}, \tilde{\mathbf{x}}_j \rangle^3 = \langle \mathbf{z}_r^{(T)}, \mathbf{x}_j \rangle^3 + 3\langle \mathbf{z}_r^{(T)}, \mathbf{x}_j \rangle^2 \langle \mathbf{z}_r^{(T)}, \boldsymbol{\delta}_j \rangle + 3\langle \mathbf{z}_r^{(T)}, \mathbf{x}_j \rangle \langle \mathbf{z}_r^{(T)}, \boldsymbol{\delta}_j \rangle^2 + \langle \mathbf{z}_r^{(T)}, \boldsymbol{\delta}_j \rangle^3. \tag{224}$$

Moreover, since $\|\boldsymbol{\delta}_j\|_\infty \leq \epsilon$, we have

$$\left| \langle \mathbf{z}_r^{(T)}, \boldsymbol{\delta}_j \rangle \right| \leq \epsilon \|\mathbf{z}_r^{(T)}\|_1 \leq 3\epsilon\sigma_0 d. \tag{225}$$

Therefore,

$$\left| y \sum_{r \in [m]} \sum_{j \neq s(\mathbf{X})} \langle \mathbf{z}_r^{(T)}, \boldsymbol{\delta}_j \rangle^3 \right| \leq mP (3\epsilon\sigma_0 d)^3. \tag{226}$$

By condition (C6), this deterministic term is absorbed by the signal margin:

$$mP (3\epsilon\sigma_0 d)^3 \leq \frac{C_0^3}{4m} \left( 1 - \frac{\epsilon}{\alpha} \right)^3. \tag{227}$$

Hence,

$$\mathcal{L}_{\mathcal{D}_L}^{\mathrm{rob}}(f_{\mathbf{W}^{(T)}}) \leq \Pr_{(\mathbf{X},y) \sim \mathcal{D}_L} \left[ y \sum_{r \in [m]} \sum_{j \neq s(\mathbf{X})} \left( \langle \mathbf{z}_r^{(T)}, \mathbf{x}_j \rangle^3 + 3\langle \mathbf{z}_r^{(T)}, \mathbf{x}_j \rangle^2 \langle \mathbf{z}_r^{(T)}, \boldsymbol{\delta}_j \rangle \right. \right.$$
$$\left. \left. + 3\langle \mathbf{z}_r^{(T)}, \mathbf{x}_j \rangle \langle \mathbf{z}_r^{(T)}, \boldsymbol{\delta}_j \rangle^2 \right) < -\frac{C_0^3}{4m} \left( 1 - \frac{\epsilon}{\alpha} \right)^3 \right]. \tag{228}$$

**Step 4.3: Tail bound.** Define

$$S_3(\mathbf{X}, y) := y \sum_{r \in [m]} \sum_{j \neq s(\mathbf{X})} \langle \mathbf{z}_r^{(T)}, \mathbf{x}_j \rangle^3, \tag{229}$$

$$S_2(\mathbf{X}, y) := 3y \sum_{r \in [m]} \sum_{j \neq s(\mathbf{X})} \langle \mathbf{z}_r^{(T)}, \mathbf{x}_j \rangle^2 \langle \mathbf{z}_r^{(T)}, \boldsymbol{\delta}_j \rangle, \tag{230}$$

$$S_1(\mathbf{X}, y) := 3y \sum_{r \in [m]} \sum_{j \neq s(\mathbf{X})} \langle \mathbf{z}_r^{(T)}, \mathbf{x}_j \rangle \langle \mathbf{z}_r^{(T)}, \boldsymbol{\delta}_j \rangle^2, \tag{231}$$

and

$$\gamma := \frac{C_0^3}{12m}\left(1 - \frac{\epsilon}{\alpha}\right)^3. \tag{232}$$

Then

$$\mathcal{L}_{\mathcal{D}_L}^{\text{rob}}(f_{\mathbf{W}^{(T)}}) \le \Pr_{(\mathbf{X},y)\sim\mathcal{D}_L}\left[S_3(\mathbf{X},y) + S_2(\mathbf{X},y) + S_1(\mathbf{X},y) < -3\gamma\right]. \tag{233}$$

Applying the union bound, we obtain

$$\mathcal{L}_{\mathcal{D}_L}^{\text{rob}}(f_{\mathbf{W}^{(T)}}) \le \Pr_{(\mathbf{X},y)\sim\mathcal{D}_L}\left[|S_3(\mathbf{X},y)| > \gamma\right] + \Pr_{(\mathbf{X},y)\sim\mathcal{D}_L}\left[|S_2(\mathbf{X},y)| > \gamma\right] + \Pr_{(\mathbf{X},y)\sim\mathcal{D}_L}\left[|S_1(\mathbf{X},y)| > \gamma\right]. \tag{234}$$

We first control $S_3$. For each $(r,j)$, the inner product $\langle\mathbf{z}_r^{(T)},\mathbf{x}_j\rangle$ is Gaussian with variance $\sigma_n^2\|\mathbf{z}_r^{(T)}\|_2^2$, and therefore Lemma I.11 with $\alpha = 2/3$ and $L = 1$ gives

$$\left\|\langle\mathbf{z}_r^{(T)},\mathbf{x}_j\rangle^3\right\|_{\Psi_{2/3,1}} \le c\,\sigma_n^3\|\mathbf{z}_r^{(T)}\|_2^3 \le 27c\,\sigma_0^3\sigma_n^3 d^{3/2}, \tag{235}$$

where we used $\|\mathbf{z}_r^{(T)}\|_2 \le 3\sigma_0\sqrt{d}$. Applying Lemma I.10 with $Q_{2/3} = (2e^{8/3})^{3/2}$, we obtain

$$\|S_3(\mathbf{X},y)\|_{\Psi_{2/3,1}} \le 27cQ_{2/3}\,mP\,\sigma_0^3\sigma_n^3 d^{3/2} := \kappa_3. \tag{236}$$

Now apply Lemma I.8 with $\alpha = 2/3$, $L = 1$, and the parameter $\kappa_3$:

$$\Pr_{(\mathbf{X},y)\sim\mathcal{D}_L}\left[|S_3(\mathbf{X},y)| > \gamma\right] \le 2\exp\left(-\min\left\{\left(\frac{\gamma}{2\kappa_3}\right)^2, \left(\frac{\gamma}{2\kappa_3}\right)^{2/3}\right\}\right). \tag{237}$$

Next, we control $S_2$. Since $\left|\langle\mathbf{z}_r^{(T)},\boldsymbol{\delta}_j\rangle\right| \le 3\epsilon\sigma_0 d$, Lemma I.11 with $\alpha = 1$ and $L = 1$ yields

$$\left\|\langle\mathbf{z}_r^{(T)},\mathbf{x}_j\rangle^2\right\|_{\Psi_{1,1}} \le c\,\sigma_n^2\|\mathbf{z}_r^{(T)}\|_2^2 \le 9c\,\sigma_0^2\sigma_n^2 d. \tag{238}$$

Therefore, for each $(r,j)$,

$$\left\|3\langle\mathbf{z}_r^{(T)},\mathbf{x}_j\rangle^2\langle\mathbf{z}_r^{(T)},\boldsymbol{\delta}_j\rangle\right\|_{\Psi_{1,1}} \le 3\left|\langle\mathbf{z}_r^{(T)},\boldsymbol{\delta}_j\rangle\right|\left\|\langle\mathbf{z}_r^{(T)},\mathbf{x}_j\rangle^2\right\|_{\Psi_{1,1}}$$
$$\le 81c\,\epsilon\,\sigma_0^3\sigma_n^2 d^2. \tag{239}$$

Applying Lemma I.10 with $Q_1 = 1$, we obtain

$$\|S_2(\mathbf{X},y)\|_{\Psi_{1,1}} \le 81c\,mP\,\epsilon\,\sigma_0^3\sigma_n^2 d^2 := \kappa_2. \tag{240}$$

Now apply Lemma I.8 with $\alpha = 1$, $L = 1$, and the parameter $\kappa_2$:

$$\Pr_{(\mathbf{X},y)\sim\mathcal{D}_L}\left[|S_2(\mathbf{X},y)| > \gamma\right] \le 2\exp\left(-\min\left\{\left(\frac{\gamma}{2\kappa_2}\right)^2, \frac{\gamma}{2\kappa_2}\right\}\right). \tag{241}$$

Finally, we control $S_1$. By Lemma I.11 with $\alpha = 2$ and $L = 1$,

$$\left\|\langle\mathbf{z}_r^{(T)},\mathbf{x}_j\rangle\right\|_{\Psi_{2,1}} \le c\,\sigma_n\|\mathbf{z}_r^{(T)}\|_2 \le 3c\,\sigma_0\sigma_n\sqrt{d}. \tag{242}$$

Since $\left|\langle\mathbf{z}_r^{(T)},\boldsymbol{\delta}_j\rangle\right| \le 3\epsilon\sigma_0 d$, for each $(r,j)$,

$$\left\|3\langle\mathbf{z}_r^{(T)},\mathbf{x}_j\rangle\langle\mathbf{z}_r^{(T)},\boldsymbol{\delta}_j\rangle^2\right\|_{\Psi_{2,1}} \le 3\left|\langle\mathbf{z}_r^{(T)},\boldsymbol{\delta}_j\rangle\right|^2\left\|\langle\mathbf{z}_r^{(T)},\mathbf{x}_j\rangle\right\|_{\Psi_{2,1}}$$
$$\le 81c\,\epsilon^2\,\sigma_0^3\sigma_n d^{5/2}. \tag{243}$$

Applying Lemma I.10 with $Q_2 = 1$, we obtain

$$\|S_1(\mathbf{X}, y)\|_{\Psi_{2,1}} \leq 81 c\, mP\, \epsilon^2\, \sigma_0^3 \sigma_n d^{5/2} \coloneqq \kappa_1. \tag{244}$$

Now apply Lemma I.8 with $\alpha = 2$, $L = 1$, and the parameter $\kappa_1$:

$$\Pr_{(\mathbf{X}, y) \sim \mathcal{D}_L} [|S_1(\mathbf{X}, y)| > \gamma] \leq 2 \exp\left(-\left(\frac{\gamma}{2\kappa_1}\right)^2\right). \tag{245}$$

Applying the three tail bounds above and the union bound, we obtain

$$\mathcal{L}_{\mathcal{D}_L}^{\text{rob}}(f_{\mathbf{W}^{(T)}}) \leq \Pr_{(\mathbf{X}, y) \sim \mathcal{D}_L} [|S_3(\mathbf{X}, y)| > \gamma] + \Pr_{(\mathbf{X}, y) \sim \mathcal{D}_L} [|S_2(\mathbf{X}, y)| > \gamma] + \Pr_{(\mathbf{X}, y) \sim \mathcal{D}_L} [|S_1(\mathbf{X}, y)| > \gamma]$$

$$\leq 2 \exp\left(-\min\left\{\left(\frac{\gamma}{2\kappa_3}\right)^2, \left(\frac{\gamma}{2\kappa_3}\right)^{2/3}\right\}\right) + 2 \exp\left(-\min\left\{\left(\frac{\gamma}{2\kappa_2}\right)^2, \frac{\gamma}{2\kappa_2}\right\}\right) \tag{246}$$

$$+ 2 \exp\left(-\left(\frac{\gamma}{2\kappa_1}\right)^2\right).$$

Under conditions (C4) and (C6), the threshold $\gamma$ satisfies $\gamma \gg \kappa_3$, $\gamma \gg \kappa_2$, and $\gamma \gg \kappa_1$. Therefore,

$$\mathcal{L}_{\mathcal{D}_L}^{\text{rob}}(f_{\mathbf{W}^{(T)}}) \leq o(1). \tag{247}$$

Combining this robust test error bound with the signal alignment, the noise-response control, and the robust training error estimate established above proves all claims of the theorem. $\qquad\square$

## G.2. Noise Growth on Unlearnable Samples

We now analyze the sparse-unlearnable regime. Unlike the learnable-only case, unlearnable samples continue to induce residual gradients, which drive the growth of a shifted noise coefficient and eventually lead to noise memorization.

**Lemma G.3** (Gradient Bound on Unlearnable Samples). Define the shifted noise coefficient and its maximum over unlearnable noise patches as:

$$\hat{\rho}_{i,j,r}^{(t)} \coloneqq \rho_{i,j,r}^{(t)} + y_i \frac{\langle \mathbf{w}_r^{(0)}, \mathbf{x}_{i,j} \rangle}{\|\mathbf{x}_{i,j}\|_2^2}, \qquad \hat{\rho}_{\max}^{(t)} \coloneqq \max_{r \in [m],\, k \in \mathcal{S}_U,\, q \neq s(\mathbf{X}_k)} \hat{\rho}_{k,q,r}^{(t)}. \tag{248}$$

On the event $\mathcal{E}$ defined in Lemma E.1, for any unlearnable sample $i \in \mathcal{S}_U$ at iteration $t$, provided that $\hat{\rho}_{\max}^{(t)} \leq \frac{C_1}{(mP)^{1/3}\sigma_n^2 d}$, the negative sigmoid function satisfies

$$\frac{1}{1 + \exp\left(4C_1^3\right)} \leq \psi\left(y_i f_{\mathbf{W}^{(t)}}(\tilde{\mathbf{X}}_i^{(t)})\right) \leq 1. \tag{249}$$

*Proof of Lemma G.3.* Consider an unlearnable sample $i \in \mathcal{S}_U$. Its signal patch is aligned with $\mathbf{v} = \mathbf{e}_d$, while the weights remain orthogonal to $\mathbf{v}$. Hence the signal-patch contribution vanishes, and the margin is determined entirely by the noise patches:

$$y_i f_{\mathbf{W}^{(t)}}(\tilde{\mathbf{X}}_i^{(t)}) = \sum_{r \in [m]} \sum_{j \neq s(\mathbf{X}_i)} y_i \langle \mathbf{w}_r^{(t)}, \mathbf{x}_{i,j} \rangle^3. \tag{250}$$

On the event $\mathcal{E}$, by (P1) and (P6),

$$\left|\frac{\langle \mathbf{w}_r^{(0)}, \mathbf{x}_{k,q} \rangle}{\|\mathbf{x}_{k,q}\|_2^2}\right| \leq \frac{4\sigma_0 \sqrt{\log\left(\frac{16NmP}{\delta}\right)}}{\sigma_n \sqrt{d}}. \tag{251}$$

By condition (C4), this implies

$$\left|\frac{\langle \mathbf{w}_r^{(0)}, \mathbf{x}_{k,q} \rangle}{\|\mathbf{x}_{k,q}\|_2^2}\right| \leq \frac{C_1}{(mP)^{1/3}\sigma_n^2 d} \tag{252}$$

for all $r \in [m]$, $k \in [N]$, and $q \neq s(\mathbf{X}_k)$.

Now fix $k \in \mathcal{S}_U$, $q \neq s(\mathbf{X}_k)$, and $r \in [m]$. Since $\rho_{k,q,r}^{(t)} \geq 0$, we have

$$\hat{\rho}_{k,q,r}^{(t)} = \rho_{k,q,r}^{(t)} + y_k \frac{\langle \mathbf{w}_r^{(0)}, \mathbf{x}_{k,q} \rangle}{\|\mathbf{x}_{k,q}\|_2^2} \geq - \left| \frac{\langle \mathbf{w}_r^{(0)}, \mathbf{x}_{k,q} \rangle}{\|\mathbf{x}_{k,q}\|_2^2} \right| \geq - \frac{C_1}{(mP)^{1/3}\sigma_n^2 d}. \tag{253}$$

Together with the assumed upper bound $\hat{\rho}_{k,q,r}^{(t)} \leq \hat{\rho}_{\max}^{(t)} \leq \frac{C_1}{(mP)^{1/3}\sigma_n^2 d}$, this yields

$$\left| \hat{\rho}_{k,q,r}^{(t)} \right| \leq \frac{C_1}{(mP)^{1/3}\sigma_n^2 d} \tag{254}$$

and

$$\rho_{k,q,r}^{(t)} \leq \frac{2C_1}{(mP)^{1/3}\sigma_n^2 d} \qquad \text{for all } k \in \mathcal{S}_U, \ q \neq s(\mathbf{X}_k), \ r \in [m]. \tag{255}$$

For each $r \in [m]$ and $j \neq s(\mathbf{X}_i)$, by the decomposition in Lemma F.2, the triangle inequality and the nonnegativity of the noise coefficients give

$$\left| \langle \mathbf{w}_r^{(t)}, \mathbf{x}_{i,j} \rangle \right| \leq \left| \hat{\rho}_{i,j,r}^{(t)} \right| \|\mathbf{x}_{i,j}\|_2^2 + \sum_{\substack{k \in \mathcal{S}_U, \ q \neq s(\mathbf{X}_k) \\ (k,q) \neq (i,j)}} \rho_{k,q,r}^{(t)} |\langle \mathbf{x}_{k,q}, \mathbf{x}_{i,j} \rangle| + \sum_{k \in \mathcal{S}_L, \ q \neq s(\mathbf{X}_k)} \rho_{k,q,r}^{(t)} |\langle \mathbf{x}_{k,q}, \mathbf{x}_{i,j} \rangle|. \tag{256}$$

On the event $\mathcal{E}$, the bounds (P1) and (P2) imply

$$\|\mathbf{x}_{i,j}\|_2^2 \leq \frac{3}{2}\sigma_n^2 d, \qquad |\langle \mathbf{x}_{k,q}, \mathbf{x}_{i,j} \rangle| \leq 2\sigma_n^2 \sqrt{d \log \left( \frac{16N^2 P^2}{\delta} \right)}. \tag{257}$$

Moreover, by condition (C1), $\frac{2C_1}{(mP)^{1/3}\sigma_n^2 d} \leq \frac{10 \log^{1/3} T}{\sigma_n^2 d}$. Hence Lemma F.10, applied with $\rho_{\max}^{\mathcal{S}_U} \leq \frac{2C_1}{(mP)^{1/3}\sigma_n^2 d}$, gives

$$\rho_{k,q,r}^{(t)} \leq \frac{\sigma_0}{\sigma_n \sqrt{d}} + \frac{C_1^2 p_{\text{un}} N}{25(mP)^{2/3}\sigma_n^2 d^{3/2} \log^{2/3} T} \qquad \text{for } k \in \mathcal{S}_L. \tag{258}$$

Substituting these estimates into the previous decomposition gives

$$\begin{aligned}
\left| \langle \mathbf{w}_r^{(t)}, \mathbf{x}_{i,j} \rangle \right| &\leq \left( \frac{C_1}{(mP)^{1/3}\sigma_n^2 d} \right) \cdot \frac{3}{2}\sigma_n^2 d + p_{\text{un}} NP \left( \frac{2C_1}{(mP)^{1/3}\sigma_n^2 d} \right) \cdot 2\sigma_n^2 \sqrt{d \log \left( \frac{16N^2 P^2}{\delta} \right)} \\
&\quad + NP \left( \frac{\sigma_0}{\sigma_n \sqrt{d}} + \frac{C_1^2 p_{\text{un}} N}{25(mP)^{2/3}\sigma_n^2 d^{3/2} \log^{2/3} T} \right) \cdot 2\sigma_n^2 \sqrt{d \log \left( \frac{16N^2 P^2}{\delta} \right)} \\
&= \frac{3C_1}{2(mP)^{1/3}} + \frac{4C_1 p_{\text{un}} NP}{(mP)^{1/3}\sqrt{d}} \sqrt{\log \left( \frac{16N^2 P^2}{\delta} \right)} + 2NP\sigma_0 \sigma_n \sqrt{\log \left( \frac{16N^2 P^2}{\delta} \right)} \\
&\quad + \frac{2C_1^2 p_{\text{un}} N^2 P}{25(mP)^{2/3} d \log^{2/3} T} \sqrt{\log \left( \frac{16N^2 P^2}{\delta} \right)}.
\end{aligned} \tag{259}$$

We now bound the last three terms separately. First, by condition (C2) and condition (C7),

$$\frac{4C_1 p_{\text{un}} NP}{(mP)^{1/3}\sqrt{d}} \sqrt{\log \left( \frac{16N^2 P^2}{\delta} \right)} \leq \frac{C_1}{100(mP)^{1/3}}. \tag{260}$$

Second, by condition (C4) and condition (C2),

$$2NP\sigma_0 \sigma_n \sqrt{\log \left( \frac{16N^2 P^2}{\delta} \right)} \leq \frac{C^{-1} NP \sqrt{\log \left( \frac{16N^2 P^2}{\delta} \right)}}{(mP)^{1/3}\sqrt{d}\sqrt{\log \left( \frac{16NmP}{\delta} \right)}} \leq \frac{C_1}{100(mP)^{1/3}}. \tag{261}$$

Finally, for the last term, using condition (C7) first gives

$$\frac{2C_1^2 p_{\mathrm{un}} N^2 P}{25(mP)^{2/3} d \log^{2/3} T} \sqrt{\log\left(\frac{16 N^2 P^2}{\delta}\right)} \leq C \frac{C_1^2 NP}{(mP)^{2/3} d \log^{2/3} T} \sqrt{\log\left(\frac{16 N^2 P^2}{\delta}\right)} \leq \frac{C_1}{100(mP)^{1/3}}, \tag{262}$$

where the last inequality follows from condition (C2). Therefore,

$$\left|\langle \mathbf{w}_r^{(t)}, \mathbf{x}_{i,j}\rangle\right| \leq \frac{3C_1}{2(mP)^{1/3}} + \frac{3C_1}{100(mP)^{1/3}} \leq \frac{4^{1/3} C_1}{(mP)^{1/3}}. \tag{263}$$

Consequently,

$$\sum_{r\in[m]} \sum_{j\neq s(\mathbf{X}_i)} \left|\langle \mathbf{w}_r^{(t)}, \mathbf{x}_{i,j}\rangle\right|^3 \leq mP \left(\frac{4^{1/3} C_1}{(mP)^{1/3}}\right)^3 = 4C_1^3. \tag{264}$$

Since the signal-patch contribution vanishes for unlearnable samples, this implies

$$y_i f_{\mathbf{W}^{(t)}}(\tilde{\mathbf{X}}_i^{(t)}) \leq 4C_1^3. \tag{265}$$

Finally, since $\psi(\cdot)$ is monotonically decreasing,

$$\psi\left(y_i f_{\mathbf{W}^{(t)}}(\tilde{\mathbf{X}}_i^{(t)})\right) \geq \frac{1}{1 + \exp(4C_1^3)}. \tag{266}$$

Together with the trivial upper bound $\psi(\cdot) \leq 1$, this proves the claimed gradient bound on unlearnable samples. $\square$

**Lemma G.4** (Dynamics of the Maximum Shifted Noise Coefficient). On the event $\mathcal{E}$ defined in Lemma E.1, suppose that the maximum shifted noise coefficient $\hat{\rho}_{\max}^{(t)}$ defined in Lemma G.3 satisfies $\hat{\rho}_{\max}^{(t)} \leq \frac{C_1}{(mP)^{1/3} \sigma_n^2 d}$. Then the discrete dynamics satisfy

$$\hat{\rho}_{\max}^{(t+1)} \geq \hat{\rho}_{\max}^{(t)} + \frac{\eta \sigma_n^4 d^2}{3N\left(1 + \exp(4C_1^3)\right)} \left(\hat{\rho}_{\max}^{(t)}\right)^2, \tag{267}$$

and

$$\hat{\rho}_{\max}^{(t+1)} \leq \hat{\rho}_{\max}^{(t)} + \frac{4800 \eta \sigma_n^4 d^2 \log\left(\frac{16 N m P}{\delta}\right)}{N} \left(\hat{\rho}_{\max}^{(t)}\right)^2. \tag{268}$$

*Proof of Lemma G.4.* Let $(r_t, i_t, j_t) = \arg\max_{r\in[m],\, i\in\mathcal{S}_U,\, j\neq s(\mathbf{X}_k)} \hat{\rho}_{r,i,j}^{(t)}$. Since the initialization shift is constant over iterations, for any $(r, i, j)$ we have

$$\hat{\rho}_{i,j,r}^{(t+1)} - \hat{\rho}_{i,j,r}^{(t)} = \rho_{i,j,r}^{(t+1)} - \rho_{i,j,r}^{(t)}. \tag{269}$$

Therefore,

$$\rho_{i_t,j_t,r_t}^{(t+1)} - \rho_{i_t,j_t,r_t}^{(t)} \leq \hat{\rho}_{\max}^{(t+1)} - \hat{\rho}_{\max}^{(t)} \leq \max_{r\in[m],\, i\in\mathcal{S}_U,\, j\neq s(\mathbf{X}_i)} \left(\rho_{i,j,r}^{(t+1)} - \rho_{i,j,r}^{(t)}\right). \tag{270}$$

From Lemma F.2, the update rule is

$$\rho_{i,j,r}^{(t+1)} - \rho_{i,j,r}^{(t)} = \frac{3\eta}{N} \psi\left(y_i f_{\mathbf{W}^{(t)}}(\tilde{\mathbf{X}}_i^{(t)})\right) \langle \mathbf{w}_r^{(t)}, \mathbf{x}_{i,j}\rangle^2, \tag{271}$$

and $\rho_{i,j,r}^{(t)}$ is monotonically non-decreasing from zero. Therefore, the maximum shifted coefficient satisfies

$$\hat{\rho}_{\max}^{(t)} \geq \hat{\rho}_{\max}^{(0)} = \max_{r\in[m],\, k\in\mathcal{S}_U,\, q\neq s(\mathbf{X}_k)} y_k \frac{\langle \mathbf{w}_r^{(0)}, \mathbf{x}_{k,q}\rangle}{\|\mathbf{x}_{k,q}\|_2^2}. \tag{272}$$

On the event $\mathcal{E}$, using (P1) and (P8), we obtain

$$\hat{\rho}_{\max}^{(t)} \geq \hat{\rho}_{\max}^{(0)} \geq \frac{\frac{1}{4} \sigma_0 \sigma_n \sqrt{d}}{\frac{3}{2} \sigma_n^2 d} = \frac{1}{6} \frac{\sigma_0}{\sigma_n \sqrt{d}}. \tag{273}$$

Moreover, using (P1) and (P6), for every $r \in [m]$, $k \in \mathcal{S}_U$, and $q \neq s(\mathbf{X}_k)$,

$$\left| \frac{\langle \mathbf{w}_r^{(0)}, \mathbf{x}_{k,q} \rangle}{\|\mathbf{x}_{k,q}\|_2^2} \right| \leq \frac{2\sigma_0 \sigma_n \sqrt{d \log\left(\frac{16NmP}{\delta}\right)}}{\frac{1}{2}\sigma_n^2 d} = \frac{4\sigma_0}{\sigma_n \sqrt{d}} \sqrt{\log\left(\frac{16NmP}{\delta}\right)}. \tag{274}$$

Therefore,

$$\left| \frac{\langle \mathbf{w}_r^{(0)}, \mathbf{x}_{k,q} \rangle}{\|\mathbf{x}_{k,q}\|_2^2} \right| \leq 24 \sqrt{\log\left(\frac{16NmP}{\delta}\right)} \, \hat{\rho}_{\max}^{(t)}. \tag{275}$$

Hence,

$$\hat{\rho}_{k,q,r}^{(t)} = \rho_{k,q,r}^{(t)} + y_k \frac{\langle \mathbf{w}_r^{(0)}, \mathbf{x}_{k,q} \rangle}{\|\mathbf{x}_{k,q}\|_2^2} \geq -\left| \frac{\langle \mathbf{w}_r^{(0)}, \mathbf{x}_{k,q} \rangle}{\|\mathbf{x}_{k,q}\|_2^2} \right| \geq -24 \sqrt{\log\left(\frac{16NmP}{\delta}\right)} \, \hat{\rho}_{\max}^{(t)} \tag{276}$$

On the other hand, by definition we have $\hat{\rho}_{k,q,r}^{(t)} \leq \hat{\rho}_{\max}^{(t)}$, therefore

$$\left| \hat{\rho}_{k,q,r}^{(t)} \right| \leq 24 \sqrt{\log\left(\frac{16NmP}{\delta}\right)} \, \hat{\rho}_{\max}^{(t)} \tag{277}$$

Furthermore,

$$\rho_{k,q,r}^{(t)} = \hat{\rho}_{k,q,r}^{(t)} - y_k \frac{\langle \mathbf{w}_r^{(0)}, \mathbf{x}_{k,q} \rangle}{\|\mathbf{x}_{k,q}\|_2^2} \leq \hat{\rho}_{\max}^{(t)} + \left| \frac{\langle \mathbf{w}_r^{(0)}, \mathbf{x}_{k,q} \rangle}{\|\mathbf{x}_{k,q}\|_2^2} \right| \leq 25 \sqrt{\log\left(\frac{16NmP}{\delta}\right)} \, \hat{\rho}_{\max}^{(t)} \tag{278}$$

for all $r \in [m]$, $k \in \mathcal{S}_U$, and $q \neq s(\mathbf{X}_k)$.

For the upper dynamics bound, fix any $i \in \mathcal{S}_U$, $j \neq s(\mathbf{X}_i)$, and $r \in [m]$. Using the decomposition in Lemma F.2,

$$\left| \langle \mathbf{w}_r^{(t)}, \mathbf{x}_{i,j} \rangle \right| \leq \left| \hat{\rho}_{i,j,r}^{(t)} \right| \|\mathbf{x}_{i,j}\|_2^2 + \sum_{\substack{k \in \mathcal{S}_U, \, q \neq s(\mathbf{X}_k) \\ (k,q) \neq (i,j)}} \rho_{k,q,r}^{(t)} \left| \langle \mathbf{x}_{k,q}, \mathbf{x}_{i,j} \rangle \right| + \sum_{k \in \mathcal{S}_L, \, q \neq s(\mathbf{X}_k)} \rho_{k,q,r}^{(t)} \left| \langle \mathbf{x}_{k,q}, \mathbf{x}_{i,j} \rangle \right|. \tag{279}$$

Using the monotonicity of the noise coefficients and the preceding bound, for every $\tau \leq t$, $k \in \mathcal{S}_U$, $q \neq s(\mathbf{X}_k)$, and $r \in [m]$, we have

$$\rho_{k,q,r}^{(\tau)} \leq \rho_{k,q,r}^{(t)} \leq 25 \sqrt{\log\left(\frac{16NmP}{\delta}\right)} \, \hat{\rho}_{\max}^{(t)}. \tag{280}$$

Applying Lemma F.10 with $\rho_{\max}^{\mathcal{S}_U} \leq 25 \sqrt{\log\left(\frac{16NmP}{\delta}\right)} \hat{\rho}_{\max}^{(t)}$ gives, for every $i \in \mathcal{S}_L$, $q \neq s(\mathbf{X}_i)$, and $r \in [m]$,

$$\rho_{i,q,r}^{(t)} \leq \frac{\sigma_0}{\sigma_n \sqrt{d}} + \frac{25 p_{\mathrm{un}} N \sigma_n^2 \sqrt{d} \log\left(\frac{16NmP}{\delta}\right)}{4 \log^{2/3} T} \left( \hat{\rho}_{\max}^{(t)} \right)^2. \tag{281}$$

Combining these coefficient bounds with (P1) and (P2), we obtain

$$\left| \langle \mathbf{w}_r^{(t)}, \mathbf{x}_{i,j} \rangle \right| \leq 24 \sqrt{\log\left(\frac{16NmP}{\delta}\right)} \hat{\rho}_{\max}^{(t)} \cdot \frac{3}{2} \sigma_n^2 d + p_{\mathrm{un}} N P \left( 25 \sqrt{\log\left(\frac{16NmP}{\delta}\right)} \hat{\rho}_{\max}^{(t)} \right) \cdot 2\sigma_n^2 \sqrt{d \log\left(\frac{16N^2P^2}{\delta}\right)}$$

$$+ N P \left( \frac{\sigma_0}{\sigma_n \sqrt{d}} + \frac{25 p_{\mathrm{un}} N \sigma_n^2 \sqrt{d} \log\left(\frac{16NmP}{\delta}\right)}{4 \log^{2/3} T} \left( \hat{\rho}_{\max}^{(t)} \right)^2 \right) \cdot 2\sigma_n^2 \sqrt{d \log\left(\frac{16N^2P^2}{\delta}\right)}. \tag{282}$$

We now show that the last three terms are negligible compared with the first leading term. We bound them one by one at the scale $\sqrt{\log\left(\frac{16NmP}{\delta}\right)} \hat{\rho}_{\max}^{(t)} \sigma_n^2 d$. First, by condition (C2) and condition (C7),

$$50 p_{\mathrm{un}} N P \sqrt{\log\left(\frac{16NmP}{\delta}\right)} \hat{\rho}_{\max}^{(t)} \sigma_n^2 \sqrt{d \log\left(\frac{16N^2P^2}{\delta}\right)} \leq \frac{1}{100} \sqrt{\log\left(\frac{16NmP}{\delta}\right)} \hat{\rho}_{\max}^{(t)} \sigma_n^2 d. \tag{283}$$

Second, since $\hat{\rho}_{\max}^{(t)} \geq \frac{1}{6}\frac{\sigma_0}{\sigma_n\sqrt{d}}$, we have

$$2NP\sigma_0\sigma_n\sqrt{\log\left(\frac{16N^2P^2}{\delta}\right)} \leq 12\frac{NP\sqrt{\log\left(\frac{16N^2P^2}{\delta}\right)}}{\sqrt{d}}\hat{\rho}_{\max}^{(t)}\sigma_n^2 d \leq \frac{1}{100}\sqrt{\log\left(\frac{16NmP}{\delta}\right)}\hat{\rho}_{\max}^{(t)}\sigma_n^2 d, \qquad (284)$$

where the last inequality follows from condition (C2). Finally, for the last term, using condition (C7) and the assumed upper bound $\hat{\rho}_{\max}^{(t)} \leq \frac{C_1}{(mP)^{1/3}\sigma_n^2 d}$, we get

$$\frac{25p_{\mathrm{un}}N^2P\sigma_n^4 d\log\left(\frac{16NmP}{\delta}\right)}{2\log^{2/3}T}\sqrt{\log\left(\frac{16N^2P^2}{\delta}\right)}\left(\hat{\rho}_{\max}^{(t)}\right)^2 \leq C\frac{NP\log\left(\frac{16NmP}{\delta}\right)\sqrt{\log\left(\frac{16N^2P^2}{\delta}\right)}}{(mP)^{1/3}d\log^{2/3}T}\hat{\rho}_{\max}^{(t)}\sigma_n^2 d$$
$$\leq \frac{1}{100}\sqrt{\log\left(\frac{16NmP}{\delta}\right)}\hat{\rho}_{\max}^{(t)}\sigma_n^2 d, \qquad (285)$$

where the last inequality follows from condition (C2). Therefore,

$$\left|\langle \mathbf{w}_r^{(t)}, \mathbf{x}_{i,j}\rangle\right| \leq 36\sqrt{\log\left(\frac{16NmP}{\delta}\right)}\hat{\rho}_{\max}^{(t)}\sigma_n^2 d + \frac{3}{100}\sqrt{\log\left(\frac{16NmP}{\delta}\right)}\hat{\rho}_{\max}^{(t)}\sigma_n^2 d \leq 40\sqrt{\log\left(\frac{16NmP}{\delta}\right)}\hat{\rho}_{\max}^{(t)}\sigma_n^2 d. \qquad (286)$$

Substituting this bound and $\psi\left(y_i f_{\mathbf{W}^{(t)}}(\tilde{\mathbf{X}}_i^{(t)})\right) \leq 1$ into the update rule yields

$$\rho_{i,j,r}^{(t+1)} - \rho_{i,j,r}^{(t)} \leq \frac{3\eta}{N}\left(40\sqrt{\log\left(\frac{16NmP}{\delta}\right)}\hat{\rho}_{\max}^{(t)}\sigma_n^2 d\right)^2 = \frac{4800\eta\sigma_n^4 d^2\log\left(\frac{16NmP}{\delta}\right)}{N}\left(\hat{\rho}_{\max}^{(t)}\right)^2. \qquad (287)$$

Taking the maximum over $(r, i, j)$ gives

$$\hat{\rho}_{\max}^{(t+1)} \leq \hat{\rho}_{\max}^{(t)} + \frac{4800\eta\sigma_n^4 d^2\log\left(\frac{16NmP}{\delta}\right)}{N}\left(\hat{\rho}_{\max}^{(t)}\right)^2. \qquad (288)$$

For the lower bound, using the maximizing triple $(r_t, i_t, j_t)$,

$$y_{i_t}\langle \mathbf{w}_{r_t}^{(t)}, \mathbf{x}_{i_t,j_t}\rangle = \hat{\rho}_{\max}^{(t)}\|\mathbf{x}_{i_t,j_t}\|_2^2 + \sum_{\substack{k\in\mathcal{S}_U,\, q\neq s(\mathbf{X}_k)\\(k,q)\neq(i_t,j_t)}} y_{i_t}y_k\,\rho_{k,q,r_t}^{(t)}\langle \mathbf{x}_{k,q}, \mathbf{x}_{i_t,j_t}\rangle + \sum_{k\in\mathcal{S}_L,\, q\neq s(\mathbf{X}_k)} y_{i_t}y_k\,\rho_{k,q,r_t}^{(t)}\langle \mathbf{x}_{k,q}, \mathbf{x}_{i_t,j_t}\rangle$$
$$\geq \hat{\rho}_{\max}^{(t)}\|\mathbf{x}_{i_t,j_t}\|_2^2 - \sum_{\substack{k\in\mathcal{S}_U,\, q\neq s(\mathbf{X}_k)\\(k,q)\neq(i_t,j_t)}} \rho_{k,q,r_t}^{(t)}|\langle \mathbf{x}_{k,q}, \mathbf{x}_{i_t,j_t}\rangle| - \sum_{k\in\mathcal{S}_L,\, q\neq s(\mathbf{X}_k)} \rho_{k,q,r_t}^{(t)}|\langle \mathbf{x}_{k,q}, \mathbf{x}_{i_t,j_t}\rangle|. \qquad (289)$$

Using the bounds established above for the unlearnable (280) and learnable (281) components, respectively, together with the lower bound (P1), which gives $\|\mathbf{x}_{i_t,j_t}\|_2^2 \geq \frac{1}{2}\sigma_n^2 d$, we obtain

$$y_{i_t}\langle \mathbf{w}_{r_t}^{(t)}, \mathbf{x}_{i_t,j_t}\rangle \geq \frac{1}{2}\hat{\rho}_{\max}^{(t)}\sigma_n^2 d - p_{\mathrm{un}}NP\left(25\sqrt{\log\left(\frac{16NmP}{\delta}\right)}\hat{\rho}_{\max}^{(t)}\right)\cdot 2\sigma_n^2\sqrt{d\log\left(\frac{16N^2P^2}{\delta}\right)}$$
$$- NP\left(\frac{\sigma_0}{\sigma_n\sqrt{d}} + \frac{25p_{\mathrm{un}}N\sigma_n^2\sqrt{d}\log\left(\frac{16NmP}{\delta}\right)}{4\log^{2/3}T}\left(\hat{\rho}_{\max}^{(t)}\right)^2\right)\cdot 2\sigma_n^2\sqrt{d\log\left(\frac{16N^2P^2}{\delta}\right)}. \qquad (290)$$

By the same absorption estimates used for the upper bound, the two negative cross-patch terms are together bounded by $\frac{1}{6}\hat{\rho}_{\max}^{(t)}\sigma_n^2 d$. Hence,

$$y_{i_t}\langle \mathbf{w}_{r_t}^{(t)}, \mathbf{x}_{i_t,j_t}\rangle \geq \frac{1}{3}\hat{\rho}_{\max}^{(t)}\sigma_n^2 d. \qquad (291)$$

Since $i_t \in \mathcal{S}_U$, Lemma G.3 gives

$$\psi\left(y_{i_t} f_{\mathbf{W}^{(t)}}(\tilde{\mathbf{X}}_{i_t}^{(t)})\right) \geq \frac{1}{1 + \exp(4C_1^3)}. \tag{292}$$

Therefore,

$$\rho_{i_t, j_t, r_t}^{(t+1)} - \rho_{i_t, j_t, r_t}^{(t)} \geq \frac{3\eta}{N} \cdot \frac{1}{1 + \exp(4C_1^3)} \left(\frac{1}{3}\hat{\rho}_{\max}^{(t)}\sigma_n^2 d\right)^2 = \frac{\eta\sigma_n^4 d^2}{3N\left(1 + \exp(4C_1^3)\right)} \left(\hat{\rho}_{\max}^{(t)}\right)^2. \tag{293}$$

Hence,

$$\hat{\rho}_{\max}^{(t+1)} \geq \hat{\rho}_{\max}^{(t)} + \frac{\eta\sigma_n^4 d^2}{3N\left(1 + \exp(4C_1^3)\right)} \left(\hat{\rho}_{\max}^{(t)}\right)^2. \tag{294}$$

This proves the stated dynamics for the maximum shifted noise coefficient. $\qquad \square$

**Lemma G.5** (Noise Memorization Time). We formalize the noise-memorization statement in Lemma 4.11. Let $T_1$ be the first hitting time at which the maximum shifted noise coefficient reaches the threshold $C_1/((mP)^{1/3}\sigma_n^2 d)$:

$$T_1 := \min\left\{t \geq 0 \,\middle|\, \hat{\rho}_{\max}^{(t)} > \frac{C_1}{(mP)^{1/3}\sigma_n^2 d}\right\}. \tag{295}$$

On the event $\mathcal{E}$ defined in Lemma E.1, this time satisfies

$$\frac{N}{76800 \log^{3/2}(16NmP/\delta)\eta\sigma_0\sigma_n^3 d^{3/2}} \leq T_1 \leq \frac{60N\left(1 + \exp(4C_1^3)\right)}{\eta\sigma_0\sigma_n^3 d^{3/2}}. \tag{296}$$

Thus, $T_0 < T_1$, where $T_0$ is defined in Lemma F.5.

Furthermore, there exist an unlearnable sample $i \in \mathcal{S}_U$, a noise patch $j \neq s(\mathbf{X}_i)$, and a filter $r \in [m]$ such that for every iteration $t$ with $T_1 \leq t \leq T$,

$$y_i \langle \mathbf{w}_r^{(t)}, \mathbf{x}_{i,j} \rangle \geq \frac{C_1}{3(mP)^{1/3}}. \tag{297}$$

*Proof of Lemma G.5.* From Lemma G.4, define

$$A := \frac{4800\eta\sigma_n^4 d^2 \log\left(\frac{16NmP}{\delta}\right)}{N}, \qquad B := \frac{\eta\sigma_n^4 d^2}{3N\left(1 + \exp(4C_1^3)\right)}. \tag{298}$$

Then the shifted maximum coefficient satisfies the recursive bounds

$$\hat{\rho}_{\max}^{(t+1)} \leq \hat{\rho}_{\max}^{(t)} + A\left(\hat{\rho}_{\max}^{(t)}\right)^2, \qquad \hat{\rho}_{\max}^{(t+1)} \geq \hat{\rho}_{\max}^{(t)} + B\left(\hat{\rho}_{\max}^{(t)}\right)^2. \tag{299}$$

We first bound the initialization scale of $\hat{\rho}_{\max}^{(0)}$. Since $\rho_{k,q,r}^{(0)} = 0$, we have

$$\hat{\rho}_{\max}^{(0)} = \max_{r \in [m], \, k \in \mathcal{S}_U, \, q \neq s(\mathbf{X}_k)} y_k \frac{\langle \mathbf{w}_r^{(0)}, \mathbf{x}_{k,q}\rangle}{\|\mathbf{x}_{k,q}\|_2^2}. \tag{300}$$

On the event $\mathcal{E}$, by the lower bound in (P8) and the upper bound in (P1),

$$\hat{\rho}_{\max}^{(0)} \geq \frac{\frac{1}{4}\sigma_0\sigma_n\sqrt{d}}{\frac{3}{2}\sigma_n^2 d} = \frac{\sigma_0}{6\sigma_n\sqrt{d}}. \tag{301}$$

Applying Lemma I.1 with target threshold $\frac{C_1}{(mP)^{1/3}\sigma_n^2 d}$, we obtain

$$T_1 \leq \frac{3}{B\hat{\rho}_{\max}^{(0)}} + \frac{8A}{B} \left\lceil \frac{\log\left(\frac{C_1}{(mP)^{1/3}\sigma_n^2 d \, \hat{\rho}_{\max}^{(0)}}\right)}{\log 2} \right\rceil. \tag{302}$$

Substituting $\hat{\rho}_{\max}^{(0)} \geq \frac{\sigma_0}{6\sigma_n\sqrt{d}}$ and the definitions of $A$ and $B$ yields

$$T_1 \leq \frac{54N\left(1 + \exp(4C_1^3)\right)}{\eta\sigma_0\sigma_n^3 d^{3/2}} + 115200\left(1 + \exp(4C_1^3)\right)\log\left(\frac{16NmP}{\delta}\right)\left\lceil\log_2\left(\frac{6C_1}{(mP)^{1/3}\sigma_n\sqrt{d}\,\sigma_0}\right)\right\rceil. \tag{303}$$

By conditions (C4) and (C5), the second term is dominated by the first term. Therefore,

$$T_1 \leq \frac{60N\left(1 + \exp(4C_1^3)\right)}{\eta\sigma_0\sigma_n^3 d^{3/2}}. \tag{304}$$

For the lower bound on the hitting time, let $T_{\mathrm{dbl}}$ denote the minimum time required for $\hat{\rho}_{\max}^{(t)}$ to double from its initial value. By condition (C4), the target threshold satisfies

$$\frac{C_1}{(mP)^{1/3}\sigma_n^2 d} \geq 2\hat{\rho}_{\max}^{(0)}. \tag{305}$$

Therefore, before hitting the threshold, the process must first double from its initial value, and hence $T_1 \geq T_{\mathrm{dbl}}$. For any $t < T_{\mathrm{dbl}}$, we have

$$\hat{\rho}_{\max}^{(t)} \leq 2\hat{\rho}_{\max}^{(0)}. \tag{306}$$

Therefore, to traverse the distance from $\hat{\rho}_{\max}^{(0)}$ to $2\hat{\rho}_{\max}^{(0)}$, the number of steps must satisfy

$$T_{\mathrm{dbl}} \geq \frac{\hat{\rho}_{\max}^{(0)}}{4A\left(\hat{\rho}_{\max}^{(0)}\right)^2} = \frac{1}{4A\hat{\rho}_{\max}^{(0)}}. \tag{307}$$

Since $T_1 \geq T_{\mathrm{dbl}}$, we obtain the same lower bound for $T_1$.

On the event $\mathcal{E}$, by (P6) and the lower bound in (P1),

$$\hat{\rho}_{\max}^{(0)} \leq \max_{r,k,q}\frac{|\langle\mathbf{w}_r^{(0)},\mathbf{x}_{k,q}\rangle|}{\|\mathbf{x}_{k,q}\|_2^2} \leq \frac{2\sigma_0\sigma_n\sqrt{d\log\left(\frac{16NmP}{\delta}\right)}}{\frac{1}{2}\sigma_n^2 d} = \frac{4\sigma_0\sqrt{\log\left(\frac{16NmP}{\delta}\right)}}{\sigma_n\sqrt{d}}. \tag{308}$$

Substituting this bound together with $A = \frac{4800\eta\sigma_n^4 d^2\log\left(\frac{16NmP}{\delta}\right)}{N}$ gives

$$T_1 \geq \frac{N}{76800\log^{3/2}(16NmP/\delta)\eta\sigma_0\sigma_n^3 d^{3/2}}. \tag{309}$$

Moreover, by Lemma F.5,

$$T_0 \leq \frac{5\exp(2C_0^3)}{\eta(\alpha - \epsilon)^3(1 - p_{\mathrm{un}})\sigma_0}. \tag{310}$$

Comparing this with the lower bound on $T_1$, it suffices that

$$(\alpha - \epsilon)^3 > \frac{384000\exp(2C_0^3)\sigma_n^3 d^{3/2}\log^{3/2}(16NmP/\delta)}{N(1 - p_{\mathrm{un}})}. \tag{311}$$

Under conditions (C3) and (C6), the above inequality holds. Hence $T_0 < T_1$.

It remains to convert the shifted-coefficient lower bound into a lower bound on the actual noise response. By the definition of $T_1$, there exist $i \in \mathcal{S}_U$, $j \neq s(\mathbf{X}_i)$, and $r \in [m]$ such that

$$\hat{\rho}_{i,j,r}^{(T_1)} \geq \frac{C_1}{(mP)^{1/3}\sigma_n^2 d}. \tag{312}$$

Since each coefficient is monotonically non-decreasing by Lemma F.2, and the initialization shift is constant over time, the shifted coefficient is also monotonically non-decreasing. Hence, for every $t$ with $T_1 \leq t \leq T$,

$$\hat{\rho}_{i,j,r}^{(t)} \geq \frac{C_1}{(mP)^{1/3}\sigma_n^2 d}. \tag{313}$$

By Lemma F.2,

$$y_i \langle \mathbf{w}_r^{(t)}, \mathbf{x}_{i,j} \rangle = \hat{\rho}_{i,j,r}^{(t)} \|\mathbf{x}_{i,j}\|_2^2 + \sum_{(k,q) \neq (i,j)} y_i y_k \, \rho_{k,q,r}^{(t)} \langle \mathbf{x}_{k,q}, \mathbf{x}_{i,j} \rangle. \tag{314}$$

Applying the triangle inequality, and using (P1), (P2), and Lemma F.9, we obtain

$$
\begin{aligned}
y_i \langle \mathbf{w}_r^{(t)}, \mathbf{x}_{i,j} \rangle &\geq \frac{C_1}{(mP)^{1/3} \sigma_n^2 d} \cdot \frac{1}{2} \sigma_n^2 d - NP \cdot \frac{10 \log^{1/3} T}{\sigma_n^2 d} \cdot 2\sigma_n^2 \sqrt{d \log \left( \frac{16 N^2 P^2}{\delta} \right)} \\
&= \frac{C_1}{2(mP)^{1/3}} - \frac{20 NP \log^{1/3} T}{\sqrt{d}} \sqrt{\log \left( \frac{16 N^2 P^2}{\delta} \right)}.
\end{aligned}
\tag{315}
$$

By condition (C2), the second term is at most $C_1/(6(mP)^{1/3})$. Therefore,

$$y_i \langle \mathbf{w}_r^{(t)}, \mathbf{x}_{i,j} \rangle \geq \frac{C_1}{3(mP)^{1/3}}. \tag{316}$$

This completes the proof. $\qquad\square$

## G.3. Adversarial Training with Unlearnable Samples

We now combine the noise-memorization analysis above with the signal-learning guarantees from the previous section. This yields the final result for adversarial training in the sparse-unlearnable regime.

**Theorem G.6** (Adversarial Training with Unlearnable Samples). *On the event $\mathcal{E}$ defined in Lemma E.1, assume that the requirements in Condition D.5 hold, with the dataset satisfying the sparse-unlearnable regime in (C7). Suppose we run AT for $T$ iterations on this dataset, in which a fraction $p_{\mathrm{un}}$ of the training samples is unlearnable. Then the learned weights satisfy:*

1. *There exists at least one filter $r \in [m]$ whose first coordinate is aligned with the learnable feature:*

$$w_{r,1}^{(T)} \geq \tilde{\Omega}(\alpha^{-1}). \tag{317}$$

2. *The network memorizes the noise features on at least one unlearnable sample. More precisely, there exists an index $i \in \mathcal{S}_U$ such that*

$$\max_{r \in [m], j \neq s(\mathbf{X}_i)} y_i \langle \mathbf{w}_r^{(T)}, \mathbf{x}_{i,j} \rangle \geq \tilde{\Omega}(1). \tag{318}$$

*Consequently, the robust training error is $o(1)$, while the robust test error is at least $\frac{1}{2} - o(1)$.*

*Proof of Theorem G.6.* **Step 1: Signal Alignment.** By Lemma F.5, after $T_0 \leq \frac{5 \exp(2C_0^3)}{\eta(\alpha - \epsilon)^3 (1 - p_{\mathrm{un}})\sigma_0}$ iterations, there exists at least one filter $r \in [m]$ such that $w_{r,1}^{(T_0)} \geq \frac{C_0}{\alpha m^{1/3}}$. Since the lower bound of the signal update in (53) is non-negative, $w_{r,1}^{(t)}$ is non-decreasing in $t$. Hence,

$$w_{r,1}^{(T)} \geq \frac{C_0}{\alpha m^{1/3}} \geq \tilde{\Omega}(\alpha^{-1}), \tag{319}$$

where the last inequality follows from condition (C8).

**Step 2: Noise Memorization.** By Lemma G.5, at the final iteration $T \geq T_1$, there exist an unlearnable sample $i \in \mathcal{S}_U$, a noise patch $j \neq s(\mathbf{X}_i)$, and a filter $r \in [m]$ such that

$$y_i \langle \mathbf{w}_r^{(T)}, \mathbf{x}_{i,j} \rangle \geq \frac{C_1}{3(mP)^{1/3}} \geq \tilde{\Omega}(1), \tag{320}$$

where the last inequality follows from condition (C8).

**Step 3: Training Error.** Decomposing the average over the learnable and unlearnable subsets, we have:

$$\frac{1}{N} \sum_{i \in [N]} \ell \left( y_i f_{\mathbf{W}^{(T)}}(\tilde{\mathbf{X}}_i^{(T)}) \right) = \frac{1}{N} \sum_{i \in \mathcal{S}_L} \ell \left( y_i f_{\mathbf{W}^{(T)}}(\tilde{\mathbf{X}}_i^{(T)}) \right) + \frac{1}{N} \sum_{i \in \mathcal{S}_U} \ell \left( y_i f_{\mathbf{W}^{(T)}}(\tilde{\mathbf{X}}_i^{(T)}) \right). \tag{321}$$

By Lemma F.8, the loss on the learnable set satisfies:

$$\frac{1}{N} \sum_{i \in \mathcal{S}_L} \ell \left( y_i f_{\mathbf{W}^{(T)}}(\tilde{\mathbf{X}}_i^{(T)}) \right) \leq \frac{10}{N} \sum_{i \in \mathcal{S}_L} \psi \left( y_i f_{\mathbf{W}^{(T)}}(\tilde{\mathbf{X}}_i^{(T)}) \right) \leq \frac{40 m^{2/3} \log^{1/3} T}{C_0^2 \left(1 - \frac{\epsilon}{\alpha}\right)^3 \eta \alpha^2 (T - T_0)}, \tag{322}$$

where the first inequality follows from Lemma I.2. Moreover, since the logistic loss is uniformly bounded above by a constant for all samples under the current parameter regime, we have:

$$\frac{1}{N} \sum_{i \in \mathcal{S}_U} \ell \left( y_i f_{\mathbf{W}^{(T)}}(\tilde{\mathbf{X}}_i^{(T)}) \right) \leq p_{\text{un}} \max_{i \in \mathcal{S}_U} \ell \left( y_i f_{\mathbf{W}^{(T)}}(\tilde{\mathbf{X}}_i^{(T)}) \right) \leq c \, p_{\text{un}}, \tag{323}$$

for some absolute constant $c > 0$. Therefore, the total robust training error is bounded by:

$$\frac{1}{N} \sum_{i \in [N]} \ell \left( y_i f_{\mathbf{W}^{(T)}}(\tilde{\mathbf{X}}_i^{(T)}) \right) \leq \frac{40 m^{2/3} \log^{1/3} T}{C_0^2 \left(1 - \frac{\epsilon}{\alpha}\right)^3 \eta \alpha^2 (T - T_0)} + c \, p_{\text{un}} \leq o(1). \tag{324}$$

where the last inequality follows from (C1), (C3) and (C7).

**Step 4: Test Error.**

**Step 4.1: Noise norm bound.** We decompose the final weight into the signal and noise components:

$$\mathbf{w}_r^{(T)} = w_{r,1}^{(T)} \mathbf{e}_1 + \mathbf{z}_r^{(T)}, \qquad r \in [m], \tag{325}$$

where $\mathbf{z}_r^{(T)} \in \text{span}(\mathbf{e}_1)^\perp$. By the update rule decomposition,

$$\mathbf{z}_r^{(T)} = \mathbf{z}_r^{(0)} + \sum_{i=1}^N \sum_{j \neq s(\mathbf{X}_i)} \rho_{i,j,r}^{(T)} \mathbf{x}_{i,j}. \tag{326}$$

Hence,

$$\|\mathbf{z}_r^{(T)}\|_2 \leq \|\mathbf{z}_r^{(0)}\|_2 + \sum_{i=1}^N \sum_{j \neq s(\mathbf{X}_i)} \rho_{i,j,r}^{(T)} \|\mathbf{x}_{i,j}\|_2. \tag{327}$$

On the event $\mathcal{E}$, using (P4), (P1), and Lemma F.9, we obtain

$$\begin{aligned}
\|\mathbf{z}_r^{(T)}\|_2 &\leq 2\sigma_0 \sqrt{d} + \sum_{i=1}^N \sum_{j \neq s(\mathbf{X}_i)} \frac{10 \log^{1/3} T}{\sigma_n^2 d} \|\mathbf{x}_{i,j}\|_2 \\
&\leq 2\sigma_0 \sqrt{d} + NP \cdot \frac{10 \log^{1/3} T}{\sigma_n^2 d} \cdot \sqrt{\frac{3}{2}} \sigma_n \sqrt{d} \\
&\leq 2\sigma_0 \sqrt{d} + 20 \log^{1/3} T \frac{NP}{\sigma_n \sqrt{d}}.
\end{aligned} \tag{328}$$

By condition (C2), the second term is bounded by $\frac{10 \log^{1/3} T}{\sigma_n}$, and by condition (C4), the first term is also bounded by $\frac{5 \log^{1/3} T}{\sigma_n}$. Therefore,

$$\|\mathbf{z}_r^{(T)}\|_2 \leq \frac{15 \log^{1/3} T}{\sigma_n}. \tag{329}$$

Moreover, by Lemma F.7 and (319),

$$\frac{C_0}{\alpha m^{1/3}} \leq w_{r,1}^{(T)} \leq \frac{3 \log^{1/3} T}{\alpha}. \tag{330}$$

**Step 4.2: Attack construction.** We now construct an admissible attack using the memorized vulnerable patch from (320). Let $(i, j, r)$ be a triple satisfying (320), and write $\mathbf{x}_{\text{vuln}} := \mathbf{x}_{i,j}$ and $y_{\text{vuln}} := y_i$. For every non-signal patch $j' \neq s(\mathbf{X})$ of a test sample $(\mathbf{X}, y) \sim \mathcal{S}_L$, define

$$\boldsymbol{\delta}_{j'}^* := -y \, y_{\text{vuln}} \cdot \frac{\epsilon}{\sigma_n \sqrt{2 \log \left( \frac{16 d N P}{\delta} \right)}} \mathbf{x}_{\text{vuln}}. \tag{331}$$

On the event $\mathcal{E}$, (P3) gives

$$\|\mathbf{x}_{\mathrm{vuln}}\|_\infty \leq \sigma_n \sqrt{2 \log \left( \frac{16dNP}{\delta} \right)}. \tag{332}$$

Therefore,

$$\|\boldsymbol{\delta}_{j'}^*\|_\infty \leq \frac{\epsilon}{\sigma_n \sqrt{2 \log \left( \frac{16dNP}{\delta} \right)}} \|\mathbf{x}_{\mathrm{vuln}}\|_\infty \leq \epsilon. \tag{333}$$

Thus $\boldsymbol{\delta}^*$ is an admissible $\ell_\infty$ perturbation.

Because the perturbation incorporates $-y$, we have

$$y \langle \mathbf{z}_{r'}^{(T)}, \boldsymbol{\delta}_{j'}^* \rangle^3 = - \left( \frac{\epsilon}{\sigma_n \sqrt{2 \log \left( \frac{16dNP}{\delta} \right)}} \right)^3 \left( y_{\mathrm{vuln}} \langle \mathbf{z}_{r'}^{(T)}, \mathbf{x}_{\mathrm{vuln}} \rangle \right)^3. \tag{334}$$

Since $\mathbf{x}_{\mathrm{vuln}}$ is a noise patch, it is orthogonal to $\mathbf{e}_1$, and hence $\langle \mathbf{w}_r^{(T)}, \mathbf{x}_{\mathrm{vuln}} \rangle = \langle \mathbf{z}_r^{(T)}, \mathbf{x}_{\mathrm{vuln}} \rangle$. Therefore, (320) yields for the vulnerable filter $r$:

$$y_{\mathrm{vuln}} \langle \mathbf{z}_r^{(T)}, \mathbf{x}_{\mathrm{vuln}} \rangle \geq \frac{C_1}{3(mP)^{1/3}}. \tag{335}$$

For any other filter $r' \neq r$, expanding the inner product via Lemma F.2 and using the non-negativity of $\rho$ gives

$$y_{\mathrm{vuln}} \langle \mathbf{z}_{r'}^{(T)}, \mathbf{x}_{\mathrm{vuln}} \rangle \geq - \left| \langle \mathbf{w}_{r'}^{(0)}, \mathbf{x}_{i,j} \rangle \right| - \sum_{(k,q) \neq (i,j)} \rho_{k,q,r'}^{(T)} \left| \langle \mathbf{x}_{k,q}, \mathbf{x}_{i,j} \rangle \right|. \tag{336}$$

Applying Lemma F.9 together with (P6) and (P2), we obtain

$$y_{\mathrm{vuln}} \langle \mathbf{z}_{r'}^{(T)}, \mathbf{x}_{\mathrm{vuln}} \rangle \geq - \left( 2\sigma_0 \sigma_n \sqrt{d \log \left( \frac{16NmP}{\delta} \right)} + \frac{20NP \log^{1/3} T}{\sqrt{d}} \sqrt{\log \left( \frac{16N^2P^2}{\delta} \right)} \right). \tag{337}$$

Using $(a+b)^3 \leq 4(a^3 + b^3)$, we bound the total negative cubic drift over the remaining filters:

$$\sum_{r' \neq r} \left| \min \left( 0, \, y_{\mathrm{vuln}} \langle \mathbf{z}_{r'}^{(T)}, \mathbf{x}_{\mathrm{vuln}} \rangle \right) \right|^3 \leq 32 \, m \, \sigma_0^3 \sigma_n^3 d^{3/2} \log^{3/2} \left( \frac{16NmP}{\delta} \right)$$
$$+ 32000 \, m \, \frac{N^3 P^3 \log T}{d^{3/2}} \log^{3/2} \left( \frac{16N^2P^2}{\delta} \right) \tag{338}$$
$$\leq \frac{C_1^3}{54mP},$$

where the last inequality follows from conditions (C2) and (C4). Hence,

$$\sum_{r' \in [m]} \left( y_{\mathrm{vuln}} \langle \mathbf{z}_{r'}^{(T)}, \mathbf{x}_{\mathrm{vuln}} \rangle \right)^3 \geq \left( \frac{C_1}{3(mP)^{1/3}} \right)^3 - \sum_{r' \neq r} \left| \min \left( 0, \, y_{\mathrm{vuln}} \langle \mathbf{z}_{r'}^{(T)}, \mathbf{x}_{\mathrm{vuln}} \rangle \right) \right|^3$$
$$\geq \frac{C_1^3}{27mP} - \frac{C_1^3}{54mP} = \frac{C_1^3}{54mP}. \tag{339}$$

Multiplying by the outer negative factor and summing over the $P-1$ non-signal patches yields

$$y \sum_{r' \in [m]} \sum_{j' \neq s(\mathbf{X})} \langle \mathbf{z}_{r'}^{(T)}, \boldsymbol{\delta}_{j'}^* \rangle^3 \leq -(P-1) \left( \frac{\epsilon}{\sigma_n \sqrt{2 \log \left( \frac{16dNP}{\delta} \right)}} \right)^3 \frac{C_1^3}{54mP}. \tag{340}$$

**Step 4.3: Probability bound.** We lower-bound the robust test error using the specific attack above. Since $\alpha^3 \sum_{r' \in [m]} (w_{r',1}^{(T)})^3 \leq 27m \log T$, we have

$$\mathcal{L}_{\mathcal{D}_L}^{\text{rob}}(f_{\mathbf{W}^{(T)}}) \geq \Pr_{(\mathbf{X},y) \sim \mathcal{D}_L} \left[ y \sum_{r \in [m]} \sum_{j \neq s(\mathbf{X})} \langle \mathbf{z}_r^{(T)}, \mathbf{x}_j + \boldsymbol{\delta}_j^* \rangle^3 \leq -27m \log T \right]. \tag{341}$$

Define

$$A(\mathbf{X}, y) := y \sum_{r \in [m]} \sum_{j \neq s(\mathbf{X})} \left( \langle \mathbf{z}_r^{(T)}, \mathbf{x}_j \rangle^3 + 3 \langle \mathbf{z}_r^{(T)}, \mathbf{x}_j \rangle \langle \mathbf{z}_r^{(T)}, \boldsymbol{\delta}_j^* \rangle^2 \right), \tag{342}$$

and

$$B(\mathbf{X}, y) := y \sum_{r \in [m]} \sum_{j \neq s(\mathbf{X})} \left( 3 \langle \mathbf{z}_r^{(T)}, \mathbf{x}_j \rangle^2 \langle \mathbf{z}_r^{(T)}, \boldsymbol{\delta}_j^* \rangle + \langle \mathbf{z}_r^{(T)}, \boldsymbol{\delta}_j^* \rangle^3 \right). \tag{343}$$

Then

$$\mathcal{L}_{\mathcal{D}_L}^{\text{rob}}(f_{\mathbf{W}^{(T)}}) \geq 1 - \Pr_{(\mathbf{X},y) \sim \mathcal{D}_L} [A(\mathbf{X}, y) > 0] - \Pr_{(\mathbf{X},y) \sim \mathcal{D}_L} [B(\mathbf{X}, y) > -27m \log T]. \tag{344}$$

We first control the odd part. Under the Gaussian data model, $\mathbf{X}$ and $-\mathbf{X}$ have the same distribution, while $\{\mathbf{z}_r^{(T)}\}_{r \in [m]}$ and $\{\boldsymbol{\delta}_j^*\}_{j \neq s(\mathbf{X})}$ are fixed. Moreover,

$$y \left( \left\langle \mathbf{z}_r^{(T)}, -\mathbf{x}_j \right\rangle^3 + 3 \left\langle \mathbf{z}_r^{(T)}, -\mathbf{x}_j \right\rangle \left\langle \mathbf{z}_r^{(T)}, \boldsymbol{\delta}_j^* \right\rangle^2 \right)$$
$$= -y \left( \left\langle \mathbf{z}_r^{(T)}, \mathbf{x}_j \right\rangle^3 + 3 \left\langle \mathbf{z}_r^{(T)}, \mathbf{x}_j \right\rangle \left\langle \mathbf{z}_r^{(T)}, \boldsymbol{\delta}_j^* \right\rangle^2 \right). \tag{345}$$

Hence $A(\mathbf{X}, y)$ is symmetric around zero, and therefore

$$\Pr_{(\mathbf{X},y) \sim \mathcal{D}_L} [A(\mathbf{X}, y) > 0] = \frac{1}{2}. \tag{346}$$

It remains to control the second term. By the deterministic bound on the cubic perturbation established above,

$$\Pr_{(\mathbf{X},y) \sim \mathcal{D}_L} [B(\mathbf{X}, y) > -27m \log T] \leq \Pr_{(\mathbf{X},y) \sim \mathcal{D}_L} \left[ y \sum_{r \in [m]} \sum_{j \neq s(\mathbf{X})} 3 \langle \mathbf{z}_r^{(T)}, \mathbf{x}_j \rangle^2 \langle \mathbf{z}_r^{(T)}, \boldsymbol{\delta}_j^* \rangle > \gamma \right], \tag{347}$$

where

$$\gamma := (P - 1) \frac{C_1^3 \epsilon^3}{54 m P \sigma_n^3 \left( 2 \log \left( \frac{16dNP}{\delta} \right) \right)^{3/2}} - 27m \log T. \tag{348}$$

By Lemma I.11, for each $(r, j)$,

$$\left\| \langle \mathbf{z}_r^{(T)}, \mathbf{x}_j \rangle^2 \right\|_{\Psi_{1,1}} \leq c \, \sigma_n^2 \| \mathbf{z}_r^{(T)} \|_2^2 \leq 225c \log^{2/3} T, \tag{349}$$

where we used $\| \mathbf{z}_r^{(T)} \|_2 \leq \frac{15 \log^{1/3} T}{\sigma_n}$. Moreover, we have

$$\left| \langle \mathbf{z}_r^{(T)}, \boldsymbol{\delta}_j^* \rangle \right| \leq \frac{\epsilon}{\sigma_n \sqrt{2 \log \left( \frac{16dNP}{\delta} \right)}} \left| \langle \mathbf{z}_r^{(T)}, \mathbf{x}_{\text{vuln}} \rangle \right|. \tag{350}$$

Using Lemma F.9 together with (P6), (P2), and (P1), we obtain

$$\left|\langle \mathbf{z}_r^{(T)}, \mathbf{x}_{\text{vuln}}\rangle\right| \leq \left|\langle \mathbf{w}_r^{(0)}, \mathbf{x}_{\text{vuln}}\rangle\right| + \rho_{i,j,r}^{(T)}\|\mathbf{x}_{\text{vuln}}\|_2^2 + \sum_{(k,q)\neq(i,j)} \rho_{k,q,r}^{(T)} \left|\langle \mathbf{x}_{k,q}, \mathbf{x}_{\text{vuln}}\rangle\right|$$

$$\leq 2\sigma_0\sigma_n\sqrt{d\log\left(\frac{16NmP}{\delta}\right)} + \frac{10\log^{1/3}T}{\sigma_n^2 d}\cdot\frac{3}{2}\sigma_n^2 d + NP\cdot\frac{10\log^{1/3}T}{\sigma_n^2 d}\cdot 2\sigma_n^2\sqrt{d\log\left(\frac{16N^2P^2}{\delta}\right)}$$

$$= 2\sigma_0\sigma_n\sqrt{d\log\left(\frac{16NmP}{\delta}\right)} + 15\log^{1/3}T + \frac{20NP\log^{1/3}T}{\sqrt{d}}\sqrt{\log\left(\frac{16N^2P^2}{\delta}\right)}.$$

$$(351)$$

Under conditions (C2) and (C4), the first and third terms are bounded by sufficiently small multiples of $\log^{1/3}T$. Hence,

$$\left|\langle \mathbf{z}_r^{(T)}, \mathbf{x}_{\text{vuln}}\rangle\right| \leq 20\log^{1/3}T. \tag{352}$$

Therefore,

$$\left|\langle \mathbf{z}_r^{(T)}, \boldsymbol{\delta}_j^*\rangle\right| \leq \frac{20\epsilon\log^{1/3}T}{\sigma_n\sqrt{2\log\left(\frac{16dNP}{\delta}\right)}}. \tag{353}$$

Applying Lemma I.10 with $Q_1 = 1$, we obtain

$$\left\|\sum_{r\in[m]}\sum_{j\neq s(\mathbf{X})} 3\langle \mathbf{z}_r^{(T)}, \mathbf{x}_j\rangle^2 \left(y\langle \mathbf{z}_r^{(T)}, \boldsymbol{\delta}_j^*\rangle\right)\right\|_{\Psi_{1,1}} \leq \sum_{r\in[m]}\sum_{j\neq s(\mathbf{X})} \left\|3\langle \mathbf{z}_r^{(T)}, \mathbf{x}_j\rangle^2 \left(y\langle \mathbf{z}_r^{(T)}, \boldsymbol{\delta}_j^*\rangle\right)\right\|_{\Psi_{1,1}}$$

$$\leq 13500\, c\, mP\, \frac{\epsilon\log T}{\sigma_n\sqrt{2\log\left(\frac{16dNP}{\delta}\right)}} := \kappa. \tag{354}$$

Now apply Lemma I.8 with $\alpha = 1$, $L = 1$, and the parameter $\kappa$:

$$\Pr_{(\mathbf{X},y)\sim\mathcal{D}_L}\left[y\sum_{r\in[m]}\sum_{j\neq s(\mathbf{X})} 3\langle \mathbf{z}_r^{(T)}, \mathbf{x}_j\rangle^2\langle \mathbf{z}_r^{(T)}, \boldsymbol{\delta}_j^*\rangle > \gamma\right] \leq 2\exp\left(-\min\left\{\left(\frac{\gamma}{2\kappa}\right)^2, \frac{\gamma}{2\kappa}\right\}\right). \tag{355}$$

Under condition (C6), the threshold $\gamma$ is positive and satisfies $\gamma \gg \kappa$, and hence the right-hand side is $o(1)$. Therefore,

$$\Pr_{(\mathbf{X},y)\sim\mathcal{D}_L}\left[B(\mathbf{X},y) > -27m\log T\right] \leq o(1). \tag{356}$$

Combining the two bounds, we conclude

$$\mathcal{L}_{\mathcal{D}_L}^{\text{rob}}(f_{\mathbf{W}^{(T)}}) \geq 1 - \frac{1}{2} - o(1) = \frac{1}{2} - o(1). \tag{357}$$

Combining this robust test error bound with the signal alignment, the noise memorization bound, and the robust training error estimate established above proves all claims of the theorem. $\qquad\square$

### G.4. Dichotomy of AT

Theorem 4.7 follows immediately from Theorem G.2 and Theorem G.6. In the learnable-only regime $p_{\text{un}} = 0$, adversarial training learns the robust signal while keeping the noise responses controlled, yielding vanishing robust training and test error. In the sparse-unlearnable regime, adversarial training still drives the robust training error to zero, but it also memorizes sample-specific noise on unlearnable samples, leading to a large robust test error.

# H. The Dichotomy of Adversarial Distillation

In this section, we analyze the dynamics of adversarial distillation under different teacher behaviors. We begin with a generic observation on learnable samples, which holds for both good and bad teachers. We then study the bad-teacher regime, where the student inherits the same harmful gradient structure as standard adversarial training, and finally the good-teacher regime, where the teacher suppresses noise growth on unlearnable samples.

**Lemma H.1** (Gradient Approximation on Learnable Samples). Consider Adversarial Distillation with a teacher satisfying the learnable-sample margin condition in Section D.4. For any learnable sample $i \in \mathcal{S}_L$, the AD and AT gradients differ by at most an exponentially small term:

$$\left| \frac{\partial \mathcal{L}_{AD}}{\partial f_{\mathbf{W}}} - \frac{\partial \mathcal{L}_{AT}}{\partial f_{\mathbf{W}}} \right| \leq e^{-Cd}. \tag{358}$$

Consequently, the local update inequalities for learnable samples used in the AT analysis remain valid for AD up to negligible terms.

*Proof of Lemma H.1.* For any learnable sample $i \in \mathcal{S}_L$, the teacher property in Section D.4 gives

$$y_i f_{\mathbf{W}_T}(\mathbf{X}_i) \geq \Gamma. \tag{359}$$

We compare the point-wise gradients of the AD and AT losses with respect to the student logit $f_{\mathbf{W}}$. By the definition of the margin-based AD loss in (32),

$$\frac{\partial \mathcal{L}_{AD}}{\partial f_{\mathbf{W}}} = -y_i \sigma(-y_i f_{\mathbf{W}}) + y_i \sigma(-y_i f_{\mathbf{W}_T}(\mathbf{X}_i)). \tag{360}$$

On the other hand, for the standard AT loss,

$$\frac{\partial \mathcal{L}_{AT}}{\partial f_{\mathbf{W}}} = -y_i \sigma(-y_i f_{\mathbf{W}}). \tag{361}$$

Subtracting the two gradients yields

$$\frac{\partial \mathcal{L}_{AD}}{\partial f_{\mathbf{W}}} - \frac{\partial \mathcal{L}_{AT}}{\partial f_{\mathbf{W}}} = y_i \sigma(-y_i f_{\mathbf{W}_T}(\mathbf{X}_i)). \tag{362}$$

Taking absolute values and using $\sigma(-z) \leq e^{-z}$ for $z \geq 0$, we obtain

$$\left| \frac{\partial \mathcal{L}_{AD}}{\partial f_{\mathbf{W}}} - \frac{\partial \mathcal{L}_{AT}}{\partial f_{\mathbf{W}}} \right| = \sigma(-y_i f_{\mathbf{W}_T}(\mathbf{X}_i)) \leq e^{-y_i f_{\mathbf{W}_T}(\mathbf{X}_i)} \leq e^{-\Gamma}. \tag{363}$$

By condition (C9), $\Gamma \geq Cd$, and hence

$$e^{-\Gamma} \leq e^{-Cd}. \tag{364}$$

Since $e^{-Cd}$ is negligible compared with the polynomial-scale quantities tracked in the analysis, the local update inequalities for learnable samples used in the AT analysis remain valid for AD up to negligible terms. $\square$

### H.1. Adversarial Distillation with Bad Teacher

We now consider the bad-teacher regime. On learnable samples, Lemma H.1 already shows that the distillation gradient matches the adversarial-training gradient up to an exponentially small per-step error. We next show that the same is true on unlearnable samples when the teacher is bad.

**Lemma H.2** (Gradient Approximation under a Bad Teacher. Formal statement of Lemma 4.13). Consider Adversarial Distillation under a Bad Teacher. For any unlearnable sample $i \in \mathcal{S}_U$, the AD and AT gradients differ by at most an exponentially small term:

$$\left| \frac{\partial \mathcal{L}_{AD}}{\partial f_{\mathbf{W}}} - \frac{\partial \mathcal{L}_{AT}}{\partial f_{\mathbf{W}}} \right| \leq e^{-Cd}. \tag{365}$$

Consequently, the local update inequalities for unlearnable samples used in the AT analysis remain valid for AD with a saturated Bad Teacher up to negligible terms.

*Proof of Lemma H.2.* For any unlearnable sample $i \in \mathcal{S}_U$, the teacher property in Definition 4.3 gives

$$y_i f_{\mathbf{W}_B}(\mathbf{X}_i) \geq \Gamma_{T_B}. \tag{366}$$

We compare the point-wise gradients of the AD and AT losses with respect to the student logit $f_{\mathbf{W}}$. By the definition of the margin-based AD loss in (32),

$$\frac{\partial \mathcal{L}_{\mathrm{AD}}}{\partial f_{\mathbf{W}}} = -y_i \sigma(-y_i f_{\mathbf{W}}) + y_i \sigma(-y_i f_{\mathbf{W}_{\mathrm{B}}}(\mathbf{X}_i)). \tag{367}$$

On the other hand, for the standard AT loss,

$$\frac{\partial \mathcal{L}_{\mathrm{AT}}}{\partial f_{\mathbf{W}}} = -y_i \sigma(-y_i f_{\mathbf{W}}). \tag{368}$$

Subtracting the two gradients yields

$$\frac{\partial \mathcal{L}_{\mathrm{AD}}}{\partial f_{\mathbf{W}}} - \frac{\partial \mathcal{L}_{\mathrm{AT}}}{\partial f_{\mathbf{W}}} = y_i \sigma(-y_i f_{\mathbf{W}_{\mathrm{B}}}(\mathbf{X}_i)). \tag{369}$$

Taking absolute values and using $\sigma(-z) \leq e^{-z}$ for $z \geq 0$, we obtain

$$\left| \frac{\partial \mathcal{L}_{\mathrm{AD}}}{\partial f_{\mathbf{W}}} - \frac{\partial \mathcal{L}_{\mathrm{AT}}}{\partial f_{\mathbf{W}}} \right| = \sigma(-y_i f_{\mathbf{W}_{\mathrm{B}}}(\mathbf{X}_i)) \leq e^{-y_i f_{\mathbf{W}_{\mathrm{B}}}(\mathbf{X}_i)} \leq e^{-\Gamma_{T_B}}. \tag{370}$$

By condition (C9), $\Gamma_{T_B} \geq Cd$, and hence

$$e^{-\Gamma_{T_B}} \leq e^{-Cd}. \tag{371}$$

Since $e^{-Cd}$ is negligible compared with the polynomial-scale quantities tracked in the analysis, the local update inequalities for unlearnable samples used in the AT analysis remain valid for AD with a saturated Bad Teacher up to negligible terms. □

**Theorem H.3** (Adversarial Distillation under a Bad Teacher). *On the event $\mathcal{E}$ defined in Lemma E.1, assume that the requirements in Condition D.5 hold, with the dataset satisfying the sparse-unlearnable regime in (C7). Suppose we run AD for $T$ iterations under a Bad Teacher. Then the learned weights satisfy:*

1. *There exists at least one filter $r \in [m]$ whose first coordinate is aligned with the learnable feature:*

$$w_{r,1}^{(T)} \geq \tilde{\Omega}(\alpha^{-1}). \tag{372}$$

2. *The network memorizes the noise features on at least one unlearnable sample. More precisely, there exists an index $i \in \mathcal{S}_U$ such that*

$$\max_{r \in [m], j \neq s(\mathbf{X}_i)} y_i \langle \mathbf{w}_r^{(T)}, \mathbf{x}_{i,j} \rangle \geq \tilde{\Omega}(1). \tag{373}$$

*Consequently, the robust training error is $o(1)$, while the robust test error is at least $\frac{1}{2} - o(1)$.*

*Proof of Theorem H.3.* By Lemma H.1, on learnable samples, the AD gradient matches the AT gradient up to a negligible error. By Lemma H.2, the same holds on unlearnable samples under a Bad Teacher. Since these errors are exponentially small, they do not affect the signal update bounds or the noise update bounds used in Theorem G.6. Therefore, the same argument as in Theorem G.6 gives the stated signal learning, unlearnable-noise memorization, robust training error, and robust test error bounds. □

### H.2. Adversarial Distillation with Good Teacher

We next consider the good-teacher regime. In this case, the teacher is uninformative on unlearnable samples, so the induced student updates no longer produce the persistent one-sided noise growth that appears in adversarial training and adversarial distillation under a bad teacher. Instead, once the unlearnable-sample margin approaches zero, the corresponding updates may enter a sign-changing regime.

To isolate the regime relevant for our analysis, we restrict attention to the finite horizon

$$T \leq T_{\max} := \frac{N}{1000 \, \eta \, mP \, \sigma_0^4 \sigma_n^6 d^3 \log^2 \left( \frac{16 NmP}{\delta} \right)}. \tag{374}$$

This horizon is already sufficient for the phenomenon of interest: over the whole interval $[0, T]$, the learnable-sample dynamics follow essentially the same signal-learning trajectory as in adversarial training.

Moreover, $T_{\max}$ is still substantially longer than the critical noise-fitting timescale $T_1$ from standard adversarial training. Indeed, Lemma G.5 gives $T_1 \leq \frac{60N\left(1+\exp(4C_1^3)\right)}{\eta\sigma_0\sigma_n^3 d^{3/2}}$. Under condition (C4), this upper bound is much smaller than $T_{\max}$:

$$\frac{60N\left(1+\exp(4C_1^3)\right)}{\eta\sigma_0\sigma_n^3 d^{3/2}} \ll T_{\max}. \tag{375}$$

Hence the admissible range $T \leq T_{\max}$ contains horizons beyond the noise memorization timescale of standard adversarial training or a bad teacher, while the good teacher prevents persistent unlearnable-noise growth throughout this interval.

**Hypothesis H.4** (Noise Stability under a Good Teacher). Consider Adversarial Distillation under a Good Teacher. On the event $\mathcal{E}$ defined in Lemma E.1, for every $t \leq T$ with $T$ satisfying (374), the following bounds hold.

For learnable samples,

$$\max_{i\in\mathcal{S}_L, j\neq s(\mathbf{X}_i), r\in[m]} \left|\rho_{i,j,r}^{(t)}\right| \leq 200\exp(2C_0^3)\log\left(\frac{16NmP}{\delta}\right)\cdot\frac{\sigma_0\sigma_n^2 d}{N(\alpha-\epsilon)^3}. \tag{376}$$

For unlearnable samples,

$$\max_{i\in\mathcal{S}_U, j\neq s(\mathbf{X}_i), r\in[m]} \left|\rho_{i,j,r}^{(t)}\right| \leq \frac{\sigma_0\sqrt{\log\left(\frac{16NmP}{\delta}\right)}}{\sigma_n\sqrt{d}}. \tag{377}$$

See Lemma H.7 for the verification of this hypothesis.

**Lemma H.5** (Inner Product Bound under a Good Teacher). Consider Adversarial Distillation under a Good Teacher. On the event $\mathcal{E}$ defined in Lemma E.1, suppose that Hypothesis H.4 holds at iteration $t$. Then, for every $r \in [m]$, $i \in \mathcal{S}_L$, and $j \neq s(\mathbf{X}_i)$,

$$\left|\langle\mathbf{w}_r^{(t)}, \mathbf{x}_{i,j}\rangle\right| \leq 3\sigma_0\sigma_n\sqrt{d\log\left(\frac{16NmP}{\delta}\right)}. \tag{378}$$

Consequently, the conclusion of Hypothesis F.3 holds at iteration $t$.

*Proof of Lemma H.5.* Fix any $r \in [m]$, $i \in \mathcal{S}_L$, and $j \neq s(\mathbf{X}_i)$. By the decomposition in Lemma F.2,

$$\langle\mathbf{w}_r^{(t)}, \mathbf{x}_{i,j}\rangle = \langle\mathbf{w}_r^{(0)}, \mathbf{x}_{i,j}\rangle + y_i\rho_{i,j,r}^{(t)}\|\mathbf{x}_{i,j}\|_2^2$$
$$+ \sum_{\substack{k\in\mathcal{S}_L,\, q\neq s(\mathbf{X}_k)\\(k,q)\neq(i,j)}} y_k\rho_{k,q,r}^{(t)}\langle\mathbf{x}_{k,q}, \mathbf{x}_{i,j}\rangle + \sum_{k\in\mathcal{S}_U,\, q\neq s(\mathbf{X}_k)} y_k\rho_{k,q,r}^{(t)}\langle\mathbf{x}_{k,q}, \mathbf{x}_{i,j}\rangle. \tag{379}$$

Therefore,

$$\left|\langle\mathbf{w}_r^{(t)}, \mathbf{x}_{i,j}\rangle\right| \leq \left|\langle\mathbf{w}_r^{(0)}, \mathbf{x}_{i,j}\rangle\right| + \left|\rho_{i,j,r}^{(t)}\right|\|\mathbf{x}_{i,j}\|_2^2$$
$$+ \sum_{\substack{k\in\mathcal{S}_L,\, q\neq s(\mathbf{X}_k)\\(k,q)\neq(i,j)}} \left|\rho_{k,q,r}^{(t)}\right||\langle\mathbf{x}_{k,q}, \mathbf{x}_{i,j}\rangle| + \sum_{k\in\mathcal{S}_U,\, q\neq s(\mathbf{X}_k)} \left|\rho_{k,q,r}^{(t)}\right||\langle\mathbf{x}_{k,q}, \mathbf{x}_{i,j}\rangle|. \tag{380}$$

Applying the bounds from Hypothesis H.4 at iteration $t$, along with (P6), (P1), and (P2), we obtain

$$\left|\langle\mathbf{w}_r^{(t)}, \mathbf{x}_{i,j}\rangle\right| \leq 2\sigma_0\sigma_n\sqrt{d\log\left(\frac{16NmP}{\delta}\right)} + 200\exp(2C_0^3)\log\left(\frac{16NmP}{\delta}\right)\cdot\frac{\sigma_0\sigma_n^2 d}{N(\alpha-\epsilon)^3}\cdot\frac{3}{2}\sigma_n^2 d$$

$$+ (1-p_{\mathrm{un}})NP\cdot 200\exp(2C_0^3)\log\left(\frac{16NmP}{\delta}\right)\cdot\frac{\sigma_0\sigma_n^2 d}{N(\alpha-\epsilon)^3}\cdot 2\sigma_n^2\sqrt{d\log\left(\frac{16N^2P^2}{\delta}\right)} \tag{381}$$

$$+ p_{\mathrm{un}}NP\cdot\frac{\sigma_0\sqrt{\log\left(\frac{16NmP}{\delta}\right)}}{\sigma_n\sqrt{d}}\cdot 2\sigma_n^2\sqrt{d\log\left(\frac{16N^2P^2}{\delta}\right)}.$$

By conditions (C3), (C4) and (C6), the second and third terms are strictly bounded by

$$\frac{1}{4}\sigma_0\sigma_n\sqrt{d\log\left(\frac{16NmP}{\delta}\right)}. \tag{382}$$

Moreover, by conditions (C2) and (C7), the fourth term is also strictly bounded by

$$\frac{1}{4}\sigma_0\sigma_n\sqrt{d\log\left(\frac{16NmP}{\delta}\right)}. \tag{383}$$

Hence,

$$\left|\langle\mathbf{w}_r^{(t)},\mathbf{x}_{i,j}\rangle\right| \leq 3\sigma_0\sigma_n\sqrt{d\log\left(\frac{16NmP}{\delta}\right)}, \tag{384}$$

which proves the claim. Since

$$3\sigma_0\sigma_n\sqrt{d\log\left(\frac{16NmP}{\delta}\right)} \leq 3\sigma_0\sigma_n\sqrt{d\log\left(\frac{16NmP}{\delta}\right)} + \frac{21p_{\mathrm{un}}NP\log^{1/3}T}{\sqrt{d}}\sqrt{\log\left(\frac{16N^2P^2}{\delta}\right)}, \tag{385}$$

the conclusion of Hypothesis F.3 holds at iteration $t$. $\qquad\square$

**Lemma H.6** (Inherited AT Estimates under Good Teacher). Consider Adversarial Distillation under a Good Teacher. Assume that Hypothesis H.4 holds for all iterations $t \leq T$. Then the following estimates continue to hold.

First, for every learnable sample $i \in \mathcal{S}_L$, if $w_{r,1}^{(t)} \leq \frac{C_0}{\alpha m^{1/3}}$ for all $r \in [m]$, then

$$\frac{1}{1+\exp(2C_0^3)} \leq \psi\left(y_i f_{\mathbf{W}^{(t)}}(\tilde{\mathbf{X}}_i^{(t)})\right) \leq 1. \tag{386}$$

Second, the hitting time $T_0 := \min\left\{t \geq 0 \;\middle|\; \max_{r\in[m]} w_{r,1}^{(t)} > \frac{C_0}{\alpha m^{1/3}}\right\}$ satisfies

$$\frac{1}{12\sqrt{2\log\left(\frac{16m}{\delta}\right)}\,\eta\alpha^3\sigma_0} \leq T_0 \leq \frac{5\exp(2C_0^3)}{\eta(\alpha-\epsilon)^3(1-p_{\mathrm{un}})\sigma_0}. \tag{387}$$

Third, for every $t \leq T_0$,

$$\max_{i\in\mathcal{S}_L,\,j\neq s(\mathbf{X}_i),\,r\in[m]}\left|\rho_{i,j,r}^{(t)}\right| \leq 100\exp(2C_0^3)\log\left(\frac{16NmP}{\delta}\right)\cdot\frac{\sigma_0\sigma_n^2 d}{N(\alpha-\epsilon)^3(1-p_{\mathrm{un}})}. \tag{388}$$

Fourth, for every $t \leq T$ and every $r \in [m]$,

$$w_{r,1}^{(t)} \leq \frac{3\log^{1/3}T}{\alpha}. \tag{389}$$

Finally, for every $t \in [T_0+1, T]$ and every learnable sample $i \in \mathcal{S}_L$,

$$\psi\left(y_i f_{\mathbf{W}^{(t)}}(\tilde{\mathbf{X}}_i^{(t)})\right) \leq \frac{4m^{2/3}\log^{1/3}T}{C_0^2\left(1-\frac{\epsilon}{\alpha}\right)^3\eta\alpha^2(t-T_0)}. \tag{390}$$

*Proof of Lemma H.6.* By Lemma H.5, the assumption Hypothesis H.4 implies that the conclusion of Hypothesis F.3 holds at every iteration $t \leq T$. Therefore, the learnable-sample inner-product bound required in the AT analysis is available throughout training.

On learnable samples, Lemma H.1 shows that the AD gradient differs from the AT gradient by at most an exponentially small term, independently of whether the teacher is good or bad. Hence every AT argument that depends only on the learnable-sample dynamics remains valid up to negligible terms.

In particular, the first claim follows from Lemma F.4, and the second claim follows from Lemma F.5, since unlearnable samples do not contribute directly to the signal-coordinate update. The fourth claim follows from Lemma F.7, and the final claim follows from Lemma F.8.

For the third claim, it suffices to repeat the threshold argument in the proof of Lemma F.6 for learnable samples. For $i \in \mathcal{S}_L$, we have

$$\left| \rho_{i,j,r}^{(t+1)} - \rho_{i,j,r}^{(t)} \right| \le \frac{3\eta}{N} \psi \left( y_i f_{\mathbf{W}^{(t)}}(\tilde{\mathbf{X}}_i^{(t)}) \right) \langle \mathbf{w}_r^{(t)}, \mathbf{x}_{i,j} \rangle^2. \tag{391}$$

Define

$$\rho_{\mathrm{th}} := 100 \exp(2C_0^3) \log\left( \frac{16NmP}{\delta} \right) \cdot \frac{\sigma_0 \sigma_n^2 d}{N(\alpha - \epsilon)^3 (1 - p_{\mathrm{un}})}. \tag{392}$$

Suppose for contradiction that there exists $t \le T_0$ such that $t$ is the first iteration satisfying

$$\max_{i \in \mathcal{S}_L,\, j \neq s(\mathbf{X}_i),\, r \in [m]} \left| \rho_{i,j,r}^{(t)} \right| > \rho_{\mathrm{th}}. \tag{393}$$

Then, by the minimality of $t$,

$$\max_{i \in \mathcal{S}_L,\, j \neq s(\mathbf{X}_i),\, r \in [m]} \left| \rho_{i,j,r}^{(\tau)} \right| \le \rho_{\mathrm{th}} \qquad \text{for all } \tau < t. \tag{394}$$

Under this bound, together with the unlearnable-sample coefficient bound in Hypothesis H.4, we estimate the learnable noise inner product as follows. By the decomposition identity in Lemma F.2, for every $\tau < t$, $i \in \mathcal{S}_L$, $j \neq s(\mathbf{X}_i)$, and $r \in [m]$,

$$\left| \langle \mathbf{w}_r^{(\tau)}, \mathbf{x}_{i,j} \rangle \right| \le \left| \langle \mathbf{w}_r^{(0)}, \mathbf{x}_{i,j} \rangle \right| + \left| \rho_{i,j,r}^{(\tau)} \right| \|\mathbf{x}_{i,j}\|_2^2$$
$$+ \sum_{\substack{k \in \mathcal{S}_L,\, q \neq s(\mathbf{X}_k) \\ (k,q) \neq (i,j)}} \left| \rho_{k,q,r}^{(\tau)} \right| |\langle \mathbf{x}_{k,q}, \mathbf{x}_{i,j} \rangle| + \sum_{k \in \mathcal{S}_U,\, q \neq s(\mathbf{X}_k)} \left| \rho_{k,q,r}^{(\tau)} \right| |\langle \mathbf{x}_{k,q}, \mathbf{x}_{i,j} \rangle|. \tag{395}$$

Using the minimality of $t$, the learnable coefficients are bounded by $\rho_{\mathrm{th}}$, while the unlearnable coefficients are bounded by Hypothesis H.4. Hence, together with (P6), (P1), and (P2),

$$\left| \langle \mathbf{w}_r^{(\tau)}, \mathbf{x}_{i,j} \rangle \right| \le 2\sigma_0 \sigma_n \sqrt{d \log\left( \frac{16NmP}{\delta} \right)} + \rho_{\mathrm{th}} \cdot \frac{3}{2} \sigma_n^2 d + (1 - p_{\mathrm{un}}) NP \cdot \rho_{\mathrm{th}} \cdot 2\sigma_n^2 \sqrt{d \log\left( \frac{16N^2P^2}{\delta} \right)}$$
$$+ p_{\mathrm{un}} NP \cdot \frac{\sigma_0 \sqrt{\log\left( \frac{16NmP}{\delta} \right)}}{\sigma_n \sqrt{d}} \cdot 2\sigma_n^2 \sqrt{d \log\left( \frac{16N^2P^2}{\delta} \right)}. \tag{396}$$

By conditions (C2), (C3), (C6) and (C7), the last three terms are bounded by a sufficiently small multiple of $\sigma_0 \sigma_n \sqrt{d \log\left( \frac{16NmP}{\delta} \right)}$. Therefore,

$$\left| \langle \mathbf{w}_r^{(\tau)}, \mathbf{x}_{i,j} \rangle \right| \le \sqrt{\frac{20}{3}} \sigma_0 \sigma_n \sqrt{d \log\left( \frac{16NmP}{\delta} \right)}. \tag{397}$$

Consequently,

$$\left| \rho_{i,j,r}^{(\tau+1)} - \rho_{i,j,r}^{(\tau)} \right| \le \frac{20\eta}{N} \sigma_0^2 \sigma_n^2 d \log\left( \frac{16NmP}{\delta} \right). \tag{398}$$

Summing over $\tau = 0, \ldots, t-1$ gives

$$\left| \rho_{i,j,r}^{(t)} \right| \le \sum_{\tau=0}^{t-1} \left| \rho_{i,j,r}^{(\tau+1)} - \rho_{i,j,r}^{(\tau)} \right| \le t \cdot \frac{20\eta}{N} \sigma_0^2 \sigma_n^2 d \log\left( \frac{16NmP}{\delta} \right) \le T_0 \cdot \frac{20\eta}{N} \sigma_0^2 \sigma_n^2 d \log\left( \frac{16NmP}{\delta} \right) \le \rho_{\mathrm{th}}, \tag{399}$$

where the last inequality uses the upper bound on $T_0$ from the second claim. This contradicts the choice of $t$. Therefore, for every $t \le T_0$,

$$\max_{i \in \mathcal{S}_L,\, j \neq s(\mathbf{X}_i),\, r \in [m]} \left| \rho_{i,j,r}^{(t)} \right| \le \rho_{\mathrm{th}}. \tag{400}$$

This proves the third claim. $\qquad\square$

**Lemma H.7** (Verification of Noise Stability under a Good Teacher. Verification of Hypothesis H.4)**.** Consider Adversarial Distillation under a Good Teacher. On the event $\mathcal{E}$ defined in Lemma E.1, suppose that (374) holds. Then both the learnable-sample and unlearnable-sample noise coefficient bounds in Hypothesis H.4 hold for all iterations $t \leq T$.

*Proof of Lemma H.7.* Assume for contradiction that there exists a first iteration $t \leq T$ at which at least one of the two bounds fails. Then for every $\tau < t$, both the learnable-sample and unlearnable-sample bounds in Hypothesis H.4 hold.

Hence, by Lemma H.5, the conclusion of Hypothesis F.3 holds at every iteration $\tau < t$. Therefore, by Lemma H.6, all inherited AT estimates remain valid on the whole interval $\{0, 1, \ldots, t-1\}$.

We first rule out failure on unlearnable samples. Fix any $i \in \mathcal{S}_U$, any $j \neq s(\mathbf{X}_i)$, and any $r \in [m]$. For every $\tau < t$, the decomposition in Lemma F.2 gives

$$
\begin{aligned}
\left| \langle \mathbf{w}_r^{(\tau)}, \mathbf{x}_{i,j} \rangle \right| &\leq \left| \langle \mathbf{w}_r^{(0)}, \mathbf{x}_{i,j} \rangle \right| + \left| \rho_{i,j,r}^{(\tau)} \right| \| \mathbf{x}_{i,j} \|_2^2 \\
&+ \sum_{k \in \mathcal{S}_L,\ q \neq s(\mathbf{X}_k)} \left| \rho_{k,q,r}^{(\tau)} \right| | \langle \mathbf{x}_{k,q}, \mathbf{x}_{i,j} \rangle | + \sum_{\substack{k \in \mathcal{S}_U,\ q \neq s(\mathbf{X}_k) \\ (k,q) \neq (i,j)}} \left| \rho_{k,q,r}^{(\tau)} \right| | \langle \mathbf{x}_{k,q}, \mathbf{x}_{i,j} \rangle | .
\end{aligned}
\tag{401}
$$

Applying the two bounds from Hypothesis H.4 together with (P6), (P1), and (P2), we obtain

$$
\begin{aligned}
\left| \langle \mathbf{w}_r^{(\tau)}, \mathbf{x}_{i,j} \rangle \right| &\leq 2\sigma_0 \sigma_n \sqrt{d \log \left( \frac{16NmP}{\delta} \right)} + \frac{3}{2} \sigma_n^2 d \cdot \frac{\sigma_0 \sqrt{\log \left( \frac{16NmP}{\delta} \right)}}{\sigma_n \sqrt{d}} \\
&+ (1 - p_{\mathrm{un}}) NP \cdot 200 \exp(2C_0^3) \log \left( \frac{16NmP}{\delta} \right) \frac{\sigma_0 \sigma_n^2 d}{N(\alpha - \epsilon)^3} \cdot 2\sigma_n^2 \sqrt{d \log \left( \frac{16N^2P^2}{\delta} \right)} \\
&+ p_{\mathrm{un}} NP \cdot \frac{\sigma_0 \sqrt{\log \left( \frac{16NmP}{\delta} \right)}}{\sigma_n \sqrt{d}} \cdot 2\sigma_n^2 \sqrt{d \log \left( \frac{16N^2P^2}{\delta} \right)} .
\end{aligned}
\tag{402}
$$

By conditions (C3), (C4) and (C6), the learnable cross term satisfies

$$
(1 - p_{\mathrm{un}}) NP \cdot 200 \exp(2C_0^3) \log \left( \frac{16NmP}{\delta} \right) \frac{\sigma_0 \sigma_n^2 d}{N(\alpha - \epsilon)^3} \cdot 2\sigma_n^2 \sqrt{d \log \left( \frac{16N^2P^2}{\delta} \right)} \leq \frac{1}{4} \sigma_0 \sigma_n \sqrt{d \log \left( \frac{16NmP}{\delta} \right)} .
\tag{403}
$$

Similarly, by conditions (C2), (C7) and (C8), the unlearnable cross term satisfies

$$
p_{\mathrm{un}} NP \cdot \frac{\sigma_0 \sqrt{\log \left( \frac{16NmP}{\delta} \right)}}{\sigma_n \sqrt{d}} \cdot 2\sigma_n^2 \sqrt{d \log \left( \frac{16N^2P^2}{\delta} \right)} \leq \frac{1}{4} \sigma_0 \sigma_n \sqrt{d \log \left( \frac{16NmP}{\delta} \right)} .
\tag{404}
$$

Combining this with the initialization term and the self term gives, for every $\tau < t$,

$$
\left| \langle \mathbf{w}_r^{(\tau)}, \mathbf{x}_{i,j} \rangle \right| \leq 4\sigma_0 \sigma_n \sqrt{d \log \left( \frac{16NmP}{\delta} \right)} .
\tag{405}
$$

Since $i \in \mathcal{S}_U$ and the teacher is good, we have, for every $\tau < t$,

$$
\rho_{i,j,r}^{(\tau+1)} - \rho_{i,j,r}^{(\tau)} = \frac{3\eta}{N} \left( \frac{1}{2} - \sigma \left( y_i f_{\mathbf{W}^{(\tau)}}(\tilde{\mathbf{X}}_i^{(\tau)}) \right) \right) \langle \mathbf{w}_r^{(\tau)}, \mathbf{x}_{i,j} \rangle^2 .
\tag{406}
$$

Moreover, since the student remains orthogonal to $\mathbf{v}$,

$$
y_i f_{\mathbf{W}^{(\tau)}}(\tilde{\mathbf{X}}_i^{(\tau)}) = \sum_{r'=1}^{m} \sum_{j' \neq s(\mathbf{X}_i)} y_i \langle \mathbf{w}_{r'}^{(\tau)}, \mathbf{x}_{i,j'} \rangle^3 .
\tag{407}
$$

Using (405), we obtain, for every $\tau < t$,

$$\left| y_i f_{\mathbf{W}^{(\tau)}}(\tilde{\mathbf{X}}_i^{(\tau)}) \right| \leq mP \left( 4\sigma_0 \sigma_n \sqrt{d \log\left(\frac{16NmP}{\delta}\right)} \right)^3 = 64\, mP\, \sigma_0^3 \sigma_n^3 d^{3/2} \log^{3/2}\left(\frac{16NmP}{\delta}\right). \tag{408}$$

Hence, using $\left|\frac{1}{2} - \sigma(z)\right| \leq \frac{1}{4}|z|$ together with (405) and (408), we obtain, for every $\tau < t$,

$$\begin{aligned} \left| \rho_{i,j,r}^{(\tau+1)} - \rho_{i,j,r}^{(\tau)} \right| &\leq \frac{3\eta}{4N} \left| y_i f_{\mathbf{W}^{(\tau)}}(\tilde{\mathbf{X}}_i^{(\tau)}) \right| \langle \mathbf{w}_r^{(\tau)}, \mathbf{x}_{i,j} \rangle^2 \\ &\leq \frac{3\eta}{4N} \cdot 64\, mP\, \sigma_0^3 \sigma_n^3 d^{3/2} \log^{3/2}\left(\frac{16NmP}{\delta}\right) \cdot 16\, \sigma_0^2 \sigma_n^2 d \log\left(\frac{16NmP}{\delta}\right) \\ &\leq \frac{1000\, \eta\, mP\, \sigma_0^5 \sigma_n^5 d^{5/2} \log^{5/2}\left(\frac{16NmP}{\delta}\right)}{N}. \end{aligned} \tag{409}$$

Summing from $\tau = 0$ to $\tau = t-1$ and using $\rho_{i,j,r}^{(0)} = 0$, we obtain

$$\left| \rho_{i,j,r}^{(t)} \right| \leq T \frac{1000\, \eta\, mP\, \sigma_0^5 \sigma_n^5 d^{5/2} \log^{5/2}\left(\frac{16NmP}{\delta}\right)}{N} \leq \frac{\sigma_0 \sqrt{\log\left(\frac{16NmP}{\delta}\right)}}{\sigma_n \sqrt{d}}, \tag{410}$$

where the last inequality follows from the assumed upper bound on $T$. Thus the unlearnable-sample bound cannot fail at time $t$.

We next rule out failure on learnable samples. Since both parts of Hypothesis H.4 hold for every $\tau < t$, Lemma H.5 yields the conclusion of Hypothesis F.3 for all $\tau < t$. Hence Lemma H.6 applies on $\{0, 1, \ldots, t-1\}$.

If $t \leq T_0$, then the third claim of Lemma H.6 gives

$$\max_{i \in \mathcal{S}_L,\, j \neq s(\mathbf{X}_i),\, r \in [m]} \left| \rho_{i,j,r}^{(t)} \right| \leq 100 \exp(2C_0^3) \log\left(\frac{16NmP}{\delta}\right) \cdot \frac{\sigma_0 \sigma_n^2 d}{N(\alpha - \epsilon)^3 (1 - p_{\mathrm{un}})}. \tag{411}$$

By condition (C7), this is bounded by the claimed learnable-sample threshold. Hence it remains to consider $t \in (T_0, T]$.

Define

$$\rho_{\mathrm{th}} := 200 \exp(2C_0^3) \log\left(\frac{16NmP}{\delta}\right) \cdot \frac{\sigma_0 \sigma_n^2 d}{N(\alpha - \epsilon)^3}. \tag{412}$$

Fix any $\tau < t$, any $r \in [m]$, any $i \in \mathcal{S}_L$, and any $j \neq s(\mathbf{X}_i)$. By the decomposition in Lemma F.2,

$$\begin{aligned} \left| \langle \mathbf{w}_r^{(\tau)}, \mathbf{x}_{i,j} \rangle \right| &\leq \left| \langle \mathbf{w}_r^{(0)}, \mathbf{x}_{i,j} \rangle \right| + \left| \rho_{i,j,r}^{(\tau)} \right| \|\mathbf{x}_{i,j}\|_2^2 \\ &+ \sum_{\substack{k \in \mathcal{S}_L,\, q \neq s(\mathbf{X}_k) \\ (k,q) \neq (i,j)}} \left| \rho_{k,q,r}^{(\tau)} \right| |\langle \mathbf{x}_{k,q}, \mathbf{x}_{i,j} \rangle| + \sum_{k \in \mathcal{S}_U,\, q \neq s(\mathbf{X}_k)} \left| \rho_{k,q,r}^{(\tau)} \right| |\langle \mathbf{x}_{k,q}, \mathbf{x}_{i,j} \rangle|. \end{aligned} \tag{413}$$

Applying the learnable-sample bound $\left| \rho_{k,q,r}^{(\tau)} \right| \leq \rho_{\mathrm{th}}$, the assumed unlearnable-sample bound in Hypothesis H.4, and the event $\mathcal{E}$, we obtain

$$\begin{aligned} \left| \langle \mathbf{w}_r^{(\tau)}, \mathbf{x}_{i,j} \rangle \right| &\leq 2\sigma_0 \sigma_n \sqrt{d \log\left(\frac{16NmP}{\delta}\right)} + \rho_{\mathrm{th}} \cdot \frac{3}{2} \sigma_n^2 d + (1 - p_{\mathrm{un}}) NP \cdot \rho_{\mathrm{th}} \cdot 2\sigma_n^2 \sqrt{d \log\left(\frac{16N^2 P^2}{\delta}\right)} \\ &+ p_{\mathrm{un}} NP \cdot \frac{\sigma_0 \sqrt{\log\left(\frac{16NmP}{\delta}\right)}}{\sigma_n \sqrt{d}} \cdot 2\sigma_n^2 \sqrt{d \log\left(\frac{16N^2 P^2}{\delta}\right)}. \end{aligned} \tag{414}$$

By conditions (C3), (C4) and (C6), the second and third terms are together bounded by

$$\frac{1}{2} \sigma_0 \sigma_n \sqrt{d \log\left(\frac{16NmP}{\delta}\right)}, \tag{415}$$

and by conditions (C2), (C7) and (C8), the last term is bounded by

$$\frac{1}{2}\sigma_0\sigma_n\sqrt{d\log\left(\frac{16NmP}{\delta}\right)}. \tag{416}$$

Hence,

$$\left|\langle \mathbf{w}_r^{(\tau)}, \mathbf{x}_{i,j}\rangle\right| \le 3\sigma_0\sigma_n\sqrt{d\log\left(\frac{16NmP}{\delta}\right)} \qquad \text{for all } \tau < t. \tag{417}$$

Now consider the learnable-sample noise update. For every $\tau \in [T_0, t-1]$, the local update inequality for learnable samples gives

$$\left|\rho_{i,j,r}^{(\tau+1)} - \rho_{i,j,r}^{(\tau)}\right| \le \frac{3\eta}{N}\psi\left(y_i f_{\mathbf{W}^{(\tau)}}(\tilde{\mathbf{X}}_i^{(\tau)})\right)\langle \mathbf{w}_r^{(\tau)}, \mathbf{x}_{i,j}\rangle^2. \tag{418}$$

Summing this from $\tau = T_0$ to $t-1$ and substituting the uniform inner product bound (417), we obtain

$$\left|\rho_{i,j,r}^{(t)}\right| \le \max_{i\in\mathcal{S}_L, j\neq s(\mathbf{X}_i), r\in[m]}\left|\rho_{i,j,r}^{(T_0)}\right| + \frac{27\eta\sigma_0^2\sigma_n^2 d\log\left(\frac{16NmP}{\delta}\right)}{N}\sum_{\tau=T_0}^{t-1}\psi\left(y_i f_{\mathbf{W}^{(\tau)}}(\tilde{\mathbf{X}}_i^{(\tau)})\right). \tag{419}$$

By the post-$T_0$ gradient decay estimate inherited from Lemma H.6,

$$\sum_{\tau=T_0}^{t-1}\psi\left(y_i f_{\mathbf{W}^{(\tau)}}(\tilde{\mathbf{X}}_i^{(\tau)})\right) \le \frac{4m^{2/3}\log^{1/3}T}{C_0^2\left(1-\frac{\epsilon}{\alpha}\right)^3\eta\alpha^2}. \tag{420}$$

Substituting this sum bound and the phase-one bound for $\left|\rho_{i,j,r}^{(T_0)}\right|$ yields

$$\begin{aligned}\left|\rho_{i,j,r}^{(t)}\right| &\le 100\exp(2C_0^3)\log\left(\frac{16NmP}{\delta}\right)\cdot\frac{\sigma_0\sigma_n^2 d}{N(\alpha-\epsilon)^3(1-p_{\text{un}})}\\ &\quad + \frac{108m^{2/3}\log^{1/3}T\cdot\sigma_0^2\sigma_n^2 d\log\left(\frac{16NmP}{\delta}\right)}{C_0^2\left(1-\frac{\epsilon}{\alpha}\right)^3 N\alpha^2}.\end{aligned} \tag{421}$$

Under conditions (C4) and (C7), the two terms on the right-hand side are together bounded by $\rho_{\text{th}}$. Therefore,

$$\left|\rho_{i,j,r}^{(t)}\right| \le \rho_{\text{th}}, \tag{422}$$

which contradicts the assumption that $t$ is the first iteration where the learnable-sample bound fails.

Hence the learnable-sample bound in Hypothesis H.4 holds for all $t \le T$, completing the proof. $\qquad\square$

**Theorem H.8** (Adversarial Distillation with Good Teacher). *On the event $\mathcal{E}$ defined in Lemma E.1, assume that the requirements of Condition D.5 are satisfied, with the dataset satisfying the sparse-unlearnable regime in (C7). Suppose we run AD under a Good Teacher for a training horizon $T$ satisfying*

$$T_0 + \frac{m^{2/3}\log(T_{\max})}{\eta\alpha^2} \le T \le T_{\max}. \tag{423}$$

*Then the learned weights satisfy:*

1. *There exists at least one filter $r \in [m]$ whose first coordinate is aligned with the learnable feature:*

$$w_{r,1}^{(T)} \ge \tilde{\Omega}(\alpha^{-1}). \tag{424}$$

2. *The network barely memorizes the noise features. More precisely, for every filter $r \in [m]$ and every noise patch $(i,j)$ with $i \in [N]$ and $j \neq s(\mathbf{X}_i)$,*

$$\left|\langle \mathbf{w}_r^{(T)}, \mathbf{x}_{i,j}\rangle\right| \le \tilde{O}(\sigma_0\sigma_n\sqrt{d}) \tag{425}$$

*Consequently, both the robust training error and robust test error are $o(1)$.*

*Proof of Theorem H.8.* We compare the present argument with the four-step proof of Theorem G.2. Under a Good Teacher, the dynamics on unlearnable samples are no longer identical to those of adversarial training without unlearnable samples. Nevertheless, the previous lemmas show that, within the finite horizon $t \leq T$, these differences are absorbed before the final structural estimates used in Theorem G.2.

**Step 1: Signal alignment.** By Lemma H.6, the learnable-sample signal dynamics satisfy the same lower-bound and monotonicity estimates as in Theorem G.2. Hence there exists a filter $r \in [m]$ such that

$$w_{r,1}^{(T)} \geq \frac{C_0}{\alpha m^{1/3}} = \tilde{\Omega}(\alpha^{-1}), \tag{426}$$

which proves the first claim of the theorem.

**Step 2: Noise inner-product bound.** Here the argument differs from Theorem G.2, because unlearnable samples are present. However, Lemma H.7 shows that, throughout the finite horizon, the noise coefficients remain uniformly bounded on both learnable and unlearnable samples. Using these coefficient bounds, the proof of Lemma H.5 yields

$$\left| \langle \mathbf{w}_r^{(T)}, \mathbf{x}_{i,j} \rangle \right| \leq 3\sigma_0 \sigma_n \sqrt{d \log \left( \frac{16NmP}{\delta} \right)} \qquad \text{for all } i \in \mathcal{S}_L, \ j \neq s(\mathbf{X}_i), \ r \in [m]. \tag{427}$$

On the other hand, applying the decomposition estimate leading to (405) in the proof of Lemma H.7, we obtain, for every unlearnable sample $i \in \mathcal{S}_U$,

$$\left| \langle \mathbf{w}_r^{(T)}, \mathbf{x}_{i,j} \rangle \right| \leq 4\sigma_0 \sigma_n \sqrt{d \log \left( \frac{16NmP}{\delta} \right)} \qquad \text{for all } j \neq s(\mathbf{X}_i), \ r \in [m]. \tag{428}$$

Therefore,

$$\left| \langle \mathbf{w}_r^{(T)}, \mathbf{x}_{i,j} \rangle \right| \leq \tilde{O}(\sigma_0 \sigma_n \sqrt{d}) \qquad \text{for all } i \in [N], \ j \neq s(\mathbf{X}_i), \ r \in [m], \tag{429}$$

which proves the second claim of the theorem.

**Step 3: Training Error.** We decompose the empirical robust logistic loss into the learnable and unlearnable parts:

$$\frac{1}{N} \sum_{i \in [N]} \ell\left(y_i f_{\mathbf{W}^{(T)}}(\tilde{\mathbf{X}}_i^{(T)})\right) = \frac{1}{N} \sum_{i \in \mathcal{S}_L} \ell\left(y_i f_{\mathbf{W}^{(T)}}(\tilde{\mathbf{X}}_i^{(T)})\right) + \frac{1}{N} \sum_{i \in \mathcal{S}_U} \ell\left(y_i f_{\mathbf{W}^{(T)}}(\tilde{\mathbf{X}}_i^{(T)})\right). \tag{430}$$

For the learnable part, Lemma H.6 gives the same post-$T_0$ decay estimate as in the pure learnable AT regime. Hence, by Lemma I.2,

$$\frac{1}{N} \sum_{i \in \mathcal{S}_L} \ell\left(y_i f_{\mathbf{W}^{(T)}}(\tilde{\mathbf{X}}_i^{(T)})\right) \leq \frac{10}{N} \sum_{i \in \mathcal{S}_L} \psi\left(y_i f_{\mathbf{W}^{(T)}}(\tilde{\mathbf{X}}_i^{(T)})\right)$$

$$\leq \frac{40 m^{2/3} \log^{1/3} T}{C_0^2 \left(1 - \frac{\epsilon}{\alpha}\right)^3 \eta \alpha^2 (T - T_0)}. \tag{431}$$

By (423), we have

$$\frac{m^{2/3} \log^{1/3} T}{\eta \alpha^2 (T - T_0)} \leq \frac{\log^{1/3} T}{\log(T_{\max})} \leq \frac{1}{\log^{2/3}(T_{\max})} = o(1). \tag{432}$$

For the unlearnable part, (408) from the proof of Lemma H.7 yields the uniform margin bound

$$\left| y_i f_{\mathbf{W}^{(T)}}(\tilde{\mathbf{X}}_i^{(T)}) \right| \leq 64 \, mP \, \sigma_0^3 \sigma_n^3 d^{3/2} \log^{3/2} \left( \frac{16NmP}{\delta} \right) \qquad \text{for all } i \in \mathcal{S}_U. \tag{433}$$

Therefore, under conditions (C4) and (C8), the logistic loss on each unlearnable sample is bounded by an absolute constant, so

$$\frac{1}{N} \sum_{i \in \mathcal{S}_U} \ell\left(y_i f_{\mathbf{W}^{(T)}}(\tilde{\mathbf{X}}_i^{(T)})\right) \leq C \, p_{\text{un}} \tag{434}$$

for some absolute constant $C > 0$.

Combining the two bounds, we obtain

$$\frac{1}{N} \sum_{i \in [N]} \ell\left(y_i f_{\mathbf{W}^{(T)}}(\tilde{\mathbf{X}}_i^{(T)})\right) \leq \frac{40 m^{2/3} \log^{1/3} T}{C_0^2 \left(1 - \frac{\epsilon}{\alpha}\right)^3 \eta \alpha^2 (T - T_0)} + C\, p_{\text{un}}. \tag{435}$$

By condition (C7), the robust training error converges to $o(1)$.

**Step 4: Robust test error.** Write

$$\mathbf{w}_r^{(T)} = w_{r,1}^{(T)} \mathbf{e}_1 + \mathbf{z}_r^{(T)}, \qquad \mathbf{z}_r^{(T)} \in \text{span}(\mathbf{e}_1)^{\perp}. \tag{436}$$

Using the decomposition of the noise component,

$$\mathbf{z}_r^{(T)} = \mathbf{z}_r^{(0)} + \sum_{i \in \mathcal{S}_L} \sum_{j \neq s(\mathbf{X}_i)} \rho_{i,j,r}^{(T)} y_i \mathbf{x}_{i,j} + \sum_{i \in \mathcal{S}_U} \sum_{j \neq s(\mathbf{X}_i)} \rho_{i,j,r}^{(T)} y_i \mathbf{x}_{i,j}, \tag{437}$$

we obtain

$$\|\mathbf{z}_r^{(T)}\|_2 \leq \|\mathbf{z}_r^{(0)}\|_2 + \sum_{i \in \mathcal{S}_L} \sum_{j \neq s(\mathbf{X}_i)} \left|\rho_{i,j,r}^{(T)}\right| \|\mathbf{x}_{i,j}\|_2 + \sum_{i \in \mathcal{S}_U} \sum_{j \neq s(\mathbf{X}_i)} \left|\rho_{i,j,r}^{(T)}\right| \|\mathbf{x}_{i,j}\|_2. \tag{438}$$

Applying (P4), (P1), and the two coefficient bounds from Lemma H.7, we get

$$\|\mathbf{z}_r^{(T)}\|_2 \leq 2\sigma_0 \sqrt{d} + (1 - p_{\text{un}}) N P \cdot 200 \exp(2 C_0^3) \log\left(\frac{16 N m P}{\delta}\right) \cdot \frac{\sigma_0 \sigma_n^2 d}{N (\alpha - \epsilon)^3} \cdot \sqrt{\frac{3}{2}} \sigma_n \sqrt{d}$$

$$+ p_{\text{un}} N P \cdot \frac{\sigma_0 \sqrt{\log\left(\frac{16 N m P}{\delta}\right)}}{\sigma_n \sqrt{d}} \cdot \sqrt{\frac{3}{2}} \sigma_n \sqrt{d}. \tag{439}$$

The learnable contribution is bounded as in Theorem G.2, and the unlearnable contribution is bounded by conditions (C2), (C7) and (C8). Therefore,

$$\|\mathbf{z}_r^{(T)}\|_2 \leq 3\sigma_0 \sqrt{d} \qquad \text{for all } r \in [m]. \tag{440}$$

At this point, the two structural conditions required in Steps 4.2 and 4.3 of Theorem G.2 are identical: the signal margin is positive, and the noise component satisfies the same $L_2$ bound. Therefore, the tail-reduction and sub-Weibull concentration arguments from Theorem G.2 apply without modification, and yield robust test error $o(1)$. $\square$

### H.3. Dichotomy of AD

Theorem 4.8 follows directly by combining Theorem H.3 and Theorem H.8. Adversarial Distillation succeeds when the teacher suppresses noise gradients (Good Teacher), whereas it fails when the student mimics spurious noise confidence (Bad Teacher).

## I. Auxiliary Lemmas

### I.1. Tensor Power Method

**Lemma I.1** (Tensor power growth - Lemma K.15 from Jelassi & Li 2022). Let $\{z^{(t)}\}_{t=0}^T$ be a positive sequence defined by

$$\begin{cases} z^{(t+1)} \leq z^{(t)} + A\left(z^{(t)}\right)^2, \\ z^{(t+1)} \geq z^{(t)} + B\left(z^{(t)}\right)^2, \end{cases} \tag{441}$$

where $z^{(0)} > 0$ is the initialization and $m, M > 0$. Let $v > 0$ be such that $z^{(0)} \leq v$. Then the time $t_0$ such that $z^{(t)} \geq v$ for all $t \geq t_0$ is

$$t_0 = \frac{3}{B z^{(0)}} + \frac{8A}{B} \left\lceil \frac{\log(v/z^{(0)})}{\log 2} \right\rceil. \tag{442}$$

**Lemma I.2** (Basic connection between derivative and loss - Lemma K.22 from Jelassi & Li 2022). Let $x > 0$. Then, the negative sigmoid function and the logistic loss satisfy:

$$0.1 \log(1 + \exp(-x)) \leq \frac{1}{1 + \exp(x)} \leq 10 \log(1 + \exp(-x)). \tag{443}$$

### I.2. Gaussian and Bernstein Bounds

The following bounds will be repeatedly used in the proof of Lemma E.1.

**Lemma I.3** (Gaussian tail bound – Proposition 2.1.2 in Vershynin 2018). For $z \sim \mathcal{N}(0, \sigma^2)$ and every $\gamma > 0$,

$$\Pr(|z| \geq \gamma) \leq 2 \exp\left(-\frac{\gamma^2}{2\sigma^2}\right). \tag{444}$$

*Proof.* By scaling, it is enough to consider the case $\sigma = 1$. Then $z \sim \mathcal{N}(0, 1)$, and Proposition 2.1.2 in Vershynin (2018) gives

$$\Pr(z \geq \gamma) \leq \frac{1}{\gamma\sqrt{2\pi}} e^{-\gamma^2/2}, \qquad \gamma > 0. \tag{445}$$

By symmetry,

$$\Pr(|z| \geq \gamma) = 2\Pr(z \geq \gamma) \leq \frac{2}{\gamma\sqrt{2\pi}} e^{-\gamma^2/2}. \tag{446}$$

If $\gamma \geq (2\pi)^{-1/2}$, then $(\gamma\sqrt{2\pi})^{-1} \leq 1$, so

$$\Pr(|z| \geq \gamma) \leq 2e^{-\gamma^2/2}. \tag{447}$$

If $0 < \gamma < (2\pi)^{-1/2}$, then trivially

$$\Pr(|z| \geq \gamma) \leq 1 \leq 2e^{-\gamma^2/2}. \tag{448}$$

$\square$

**Lemma I.4** (Bernstein bound for Gaussian squares – Lemma 2.8.5 and Corollary 2.9.2 in Vershynin 2018). For i.i.d. standard Gaussian random variables $g_1, \ldots, g_n$ and every $\gamma > 0$,

$$\Pr\left(\left|\sum_{\ell=1}^{n} g_\ell^2 - n\right| \geq \gamma\right) \leq 2 \exp\left(-c \min\left\{\frac{\gamma^2}{n}, \gamma\right\}\right) \tag{449}$$

for an absolute constant $c > 0$.

*Proof.* Since $\mathbb{E}[g_\ell^2] = 1$, the random variables $g_\ell^2 - 1$ are independent and centered. By Lemma 2.8.5 in Vershynin (2018), each $g_\ell^2 - 1$ is sub-exponential with sub-exponential norm bounded by an absolute constant. Applying Corollary 2.9.2 in Vershynin (2018) to the sum $\sum_{\ell=1}^{n}(g_\ell^2 - 1)$ yields

$$\Pr\left(\left|\sum_{\ell=1}^{n} g_\ell^2 - n\right| \geq \gamma\right) \leq 2 \exp\left(-c \min\left\{\frac{\gamma^2}{n}, \gamma\right\}\right) \tag{450}$$

after adjusting the absolute constant $c$.
$\square$

**Lemma I.5** (Bernstein bound for products of independent Gaussians). For two independent collections of i.i.d. standard Gaussian random variables $g_1, \ldots, g_n$ and $z_1, \ldots, z_n$, and every $\gamma > 0$,

$$\Pr\left(\left|\sum_{\ell=1}^{n} g_\ell z_\ell\right| \geq \gamma\right) \leq 2 \exp\left(-\frac{1}{4} \min\left\{\frac{\gamma^2}{n}, \gamma\right\}\right). \tag{451}$$

*Proof.* The random variables $g_\ell z_\ell$ are independent and centered, since $\mathbb{E}[g_\ell z_\ell] = \mathbb{E}[g_\ell]\mathbb{E}[z_\ell] = 0$. Moreover, for every $|\lambda| < 1$,

$$\mathbb{E}\left[\exp(\lambda g_\ell z_\ell)\right] = (1 - \lambda^2)^{-1/2}. \tag{452}$$

If $|\lambda| \leq 1/2$, then $-\frac{1}{2}\log(1 - \lambda^2) \leq \lambda^2$, and hence

$$\mathbb{E}\left[\exp\left(\lambda \sum_{\ell=1}^{n} g_\ell z_\ell\right)\right] \leq \exp(n\lambda^2). \tag{453}$$

Therefore, for every $\gamma > 0$ and every $0 < \lambda \le 1/2$, Chernoff's bound gives

$$\Pr\left(\sum_{\ell=1}^{n} g_\ell z_\ell \ge \gamma\right) \le \exp\left(-\lambda\gamma + n\lambda^2\right). \tag{454}$$

Choosing $\lambda = \gamma/(2n)$ when $\gamma \le n$, and $\lambda = 1/2$ when $\gamma > n$, we obtain

$$\Pr\left(\sum_{\ell=1}^{n} g_\ell z_\ell \ge \gamma\right) \le \exp\left(-\frac{1}{4}\min\left\{\frac{\gamma^2}{n}, \gamma\right\}\right). \tag{455}$$

Applying the same bound to $-\sum_{\ell=1}^{n} g_\ell z_\ell$ and taking a union bound yields

$$\Pr\left(\left|\sum_{\ell=1}^{n} g_\ell z_\ell\right| \ge \gamma\right) \le 2\exp\left(-\frac{1}{4}\min\left\{\frac{\gamma^2}{n}, \gamma\right\}\right). \tag{456}$$

$\square$

**Lemma I.6** (Lower tail of Gaussian maxima). For i.i.d. random variables $z_1, \ldots, z_n \sim \mathcal{N}(0, \sigma^2)$,

$$\Pr\left(\max_{r\in[n]} z_r \le \frac{1}{2}\sigma\right) \le e^{-cn} \tag{457}$$

for a sufficiently small absolute constant $c > 0$.

*Proof.* By independence,

$$\Pr\left(\max_{r\in[n]} z_r \le \frac{1}{2}\sigma\right) = \prod_{r=1}^{n} \Pr\left(z_r \le \frac{1}{2}\sigma\right) = \Phi\left(\frac{1}{2}\right)^n. \tag{458}$$

Since $\Phi(1/2) \in (0, 1)$ is an absolute constant, there exists a sufficiently small absolute constant $c > 0$ such that

$$\Phi\left(\frac{1}{2}\right)^n \le e^{-cn}. \tag{459}$$

$\square$

### I.3. Generalized Bernstein–Orlicz (GBO) quasi-norm

We follow Kuchibhotla & Chakrabortty (2022) and work with the Generalized Bernstein–Orlicz (GBO) quasi-norm, which captures a two-regime tail behavior (Gaussian for moderate deviations and Weibull of order $\alpha$ for large deviations).

**Definition I.7** (GBO quasi-norm - Definition 2.1 and 2.3 from Kuchibhotla & Chakrabortty 2022). Let $g : [0, \infty) \to [0, \infty)$ be non-decreasing with $g(0) = 0$. The $g$-Orlicz norm of a real-valued random variable $X$ is defined by

$$\|X\|_g \coloneqq \inf\left\{\eta > 0 : \mathbb{E}[g(|X|/\eta)] \le 1\right\}. \tag{460}$$

Fix $\alpha > 0$ and $L \ge 0$. Define $\Psi_{\alpha,L}$ via its inverse

$$\Psi_{\alpha,L}^{-1}(t) \coloneqq \sqrt{\log(1+t)} + L\left(\log(1+t)\right)^{1/\alpha}, \qquad t \ge 0. \tag{461}$$

The GBO quasi-norm of $X$ is $\|X\|_{\Psi_{\alpha,L}}$ obtained by taking $g = \Psi_{\alpha,L}$ in the above Orlicz definition.

**Lemma I.8** (Tail bound - Proposition A.3 from Kuchibhotla & Chakrabortty 2022). Let $\kappa = \|X\|_{\Psi_{\alpha,L}}$. Then, for every $t \ge 0$,

$$\Pr\left(|X| \ge \kappa\left(\sqrt{t} + Lt^{1/\alpha}\right)\right) \le 2e^{-t}. \tag{462}$$

Equivalently, for every $\gamma \ge 0$,

$$\Pr\left(|X| \ge \gamma\right) \le 2\exp\left(-\min\left\{\left(\frac{\gamma}{2\kappa}\right)^2, \left(\frac{\gamma}{2L\kappa}\right)^\alpha\right\}\right), \tag{463}$$

where for $L = 0$ we interpret $\left(\frac{\gamma}{2L\kappa}\right)^\alpha = +\infty$ so that the minimum reduces to the first term.

*Proof of Lemma I.8.* The first display is exactly Proposition A.3 from Kuchibhotla & Chakrabortty (2022). We show it implies the threshold form. Fix $\gamma \geq 0$. If $L = 0$, choose $t = (\gamma/(2\kappa))^2$. Then $\kappa\sqrt{t} = \gamma/2 \leq \gamma$, and hence

$$\Pr(|X| \geq \gamma) \leq \Pr\left(|X| \geq \kappa\sqrt{t}\right) \leq 2e^{-t} = 2\exp\left(-\left(\frac{\gamma}{2\kappa}\right)^2\right). \tag{464}$$

Now assume $L > 0$ and set

$$t := \min\left\{\left(\frac{\gamma}{2\kappa}\right)^2, \left(\frac{\gamma}{2L\kappa}\right)^\alpha\right\}. \tag{465}$$

By construction, $\kappa\sqrt{t} \leq \gamma/2$ and $\kappa L t^{1/\alpha} \leq \gamma/2$, so

$$\kappa\left(\sqrt{t} + Lt^{1/\alpha}\right) \leq \gamma. \tag{466}$$

Therefore, since $\{|X| \geq \gamma\} \subseteq \{|X| \geq \kappa(\sqrt{t} + Lt^{1/\alpha})\}$, we have

$$\Pr(|X| \geq \gamma) \leq \Pr\left(|X| \geq \kappa\left(\sqrt{t} + Lt^{1/\alpha}\right)\right) \leq 2e^{-t}. \tag{467}$$

Substituting the chosen $t$ yields the claimed bound. $\qquad\square$

**Lemma I.9** (Moment characterization of GBO norm - Proposition A.4 from Kuchibhotla & Chakrabortty 2022)**.** For any random variable $X$ and parameters $\alpha > 0, L \geq 0$, the GBO quasi-norm is equivalent to the growth rate of moments:

$$c^*(\alpha)\sup_{p \geq 1}\frac{\|X\|_p}{\sqrt{p} + Lp^{1/\alpha}} \leq \|X\|_{\Psi_{\alpha,L}} \leq c_*(\alpha)\sup_{p \geq 1}\frac{\|X\|_p}{\sqrt{p} + Lp^{1/\alpha}}, \tag{468}$$

where $\|X\|_p := (\mathbb{E}[|X|^p])^{1/p}$ is the $L_p$-norm, and the constants are defined as:

$$c^*(\alpha) := \frac{1}{2}\min\{1, \alpha^{1/\alpha}\} \quad \text{and} \quad c_*(\alpha) := e\max\{2, 4^{1/\alpha}\}. \tag{469}$$

**Lemma I.10** (Quasi-triangle inequality - Proposition A.5 from Kuchibhotla & Chakrabortty 2022)**.** For any (possibly dependent) random variables $X_1, \ldots, X_N$ and any $\alpha > 0, L \geq 0$,

$$\left\|\sum_{i=1}^N X_i\right\|_{\Psi_{\alpha,L}} \leq Q_\alpha \sum_{i=1}^N \|X_i\|_{\Psi_{\alpha,L}}, \tag{470}$$

where

$$Q_\alpha := \begin{cases} (2e^{4/\alpha})^{1/\alpha}, & 0 < \alpha < 1, \\ 1, & \alpha \geq 1. \end{cases} \tag{471}$$

**Lemma I.11** (Linear, quadratic, and cubic Gaussian bounds)**.** Let $Z \sim \mathcal{N}(0, \sigma^2)$. Then there exists an absolute constant $c > 0$ such that

$$\|Z\|_{\Psi_{2,1}} \leq c\sigma, \qquad \|Z^2\|_{\Psi_{1,1}} \leq c\sigma^2, \qquad \|Z^3\|_{\Psi_{2/3,1}} \leq c\sigma^3. \tag{472}$$

*Proof of Lemma I.11.* For all $p \geq 1$, Gaussian moments satisfy

$$\|Z\|_p \leq c_0\sigma\sqrt{p} \tag{473}$$

for an absolute constant $c_0 > 0$.

We first consider the linear case. Let $Y_1 = Z$. Then

$$\sup_{p \geq 1}\frac{\|Y_1\|_p}{\sqrt{p} + p^{1/2}} \leq c_0\sigma. \tag{474}$$

Applying Lemma I.9 with $\alpha = 2$ and $L = 1$ yields

$$\|Z\|_{\Psi_{2,1}} \leq c_*(2) \sup_{p \geq 1} \frac{\|Y_1\|_p}{\sqrt{p} + p^{1/2}} \leq c\,\sigma, \tag{475}$$

after absorbing constants.

Next, consider the quadratic case. Let $Y_2 = Z^2$. Then

$$\|Y_2\|_p = \|Z\|_{2p}^2 \leq (c_0 \sigma \sqrt{2p})^2 \leq C\,\sigma^2 p \tag{476}$$

for an absolute constant $C > 0$. Therefore,

$$\sup_{p \geq 1} \frac{\|Y_2\|_p}{\sqrt{p} + p} \leq C\,\sigma^2. \tag{477}$$

Applying Lemma I.9 with $\alpha = 1$ and $L = 1$ yields

$$\|Z^2\|_{\Psi_{1,1}} \leq c_*(1) \sup_{p \geq 1} \frac{\|Y_2\|_p}{\sqrt{p} + p} \leq c\,\sigma^2, \tag{478}$$

after absorbing constants.

Finally, consider the cubic case. Let $Y_3 = Z^3$. Then

$$\|Y_3\|_p = \|Z\|_{3p}^3 \leq (c_0 \sigma \sqrt{3p})^3 \leq C\,\sigma^3 p^{3/2} \tag{479}$$

for an absolute constant $C > 0$. Hence,

$$\sup_{p \geq 1} \frac{\|Y_3\|_p}{\sqrt{p} + p^{3/2}} \leq C\,\sigma^3. \tag{480}$$

Applying Lemma I.9 with $\alpha = 2/3$ and $L = 1$ gives

$$\|Z^3\|_{\Psi_{2/3,1}} \leq c_*(2/3) \sup_{p \geq 1} \frac{\|Y_3\|_p}{\sqrt{p} + p^{3/2}} \leq c\,\sigma^3, \tag{481}$$

after absorbing constants. $\qquad\square$

