# OpenReview forum: "Toward Understanding Adversarial Distillation: Why Robust Teachers Fail"
_ICML.cc/2026/Conference — ICML 2026 regular_

### Official Review · Reviewer_tQMT · 2026-03-11

**Soundness:** 3
**Presentation:** 4
**Significance:** 3
**Originality:** 4
**Overall Recommendation:** 4
**Confidence:** 4

**Summary:**

This paper investigates the inconsistent performance of Adversarial Distillation (AD), specifically addressing why more robust teacher models often fail to improve—or even actively degrade—the robust generalization of student models. The authors identify a "Robustly Unlearnable Set" (RUS) as the primary driver of this phenomenon, arguing that robust overfitting is caused by a misalignment between a teacher's high-confidence supervision on these samples and the student's inherent representational limits. Through a theoretical framework using a two-layer neural network, the study demonstrates a dichotomy: while confident labels on unlearnable samples force students to memorize noise, a teacher’s high uncertainty (high entropy) on such data suppresses noise memorization and enhances the learning of robust signals. Validated by both simulations and empirical experiments on real-image datasets, the paper concludes that a teacher's predictive entropy on the RUS serves as a reliable indicator of student robustness, offering a principled guideline for teacher selection in adversarial distillation.

**Compliance With Llm Reviewing Policy:**

Affirmed.

**Final Justification:**

I recommend a Weak Accept for this paper, as its strengths in technical soundness, originality, and significance (addressing AD’s inconsistent performance via the RUS concept) outweigh its manageable weaknesses (e.g., limited generalization to deep architectures, computational cost of RUS identification).

**Key Questions For Authors:**

1.Generalization to Modern Architectures: Your theoretical analysis is grounded in a two-layer neural network. While this provides excellent clarity, how do you expect the dynamics of the "Robustly Unlearnable Set" (RUS) to change in much deeper or non-convolutional architectures (e.g., WideResNets or Vision Transformers)? Specifically, does the increased capacity of these models significantly shrink the RUS, or does the "dichotomy" of teacher confidence remain equally prevalent?

Impact on Evaluation: A response clarifying this would help determine if the findings are universal properties of adversarial distillation or specific to certain model complexities.

2.Computational Overhead of Identifying the RUS: In a practical, large-scale training pipeline, what is the most efficient way to identify the Robustly Unlearnable Set? If identifying this set requires multiple training runs or expensive sensitivity analysis, does it offset the efficiency gains of using distillation over standard Adversarial Training?

Impact on Evaluation: This addresses the "Significance" and "Practical Utility" of the work. A low-cost method for identifying the RUS would significantly upgrade the paper's value for practitioners.

3.Distinction from "Hard Samples" and Noisy Labels: How does the RUS fundamentally differ from traditional "hard-to-learn" samples or samples with inherently noisy labels? Is the "unlearnability" a property of the data distribution itself under adversarial constraints, or is it strictly relative to the student's architecture?

Impact on Evaluation: This would clarify the "Originality" of the RUS concept. If the RUS is a distinct phenomenon unique to the adversarial min-max framework, it strengthens the paper’s claim to a novel discovery.

3.Sensitivity to Epsilon ($\epsilon$) Bounds: Your theory suggests that the RUS is a result of adversarial constraints. How sensitive is the size and composition of the RUS to the choice of the perturbation budget ($\epsilon$)? If $\epsilon$ is reduced, does the teacher's predictive entropy on the previous RUS still serve as a reliable indicator of student performance?

Impact on Evaluation: This would demonstrate the robustness of your proposed metric. If the metric holds across various $\epsilon$ levels, it confirms that the "predictive entropy" guideline is a stable and reliable tool for model selection.

**Limitations:**

Yes

**Strengths And Weaknesses:**

Strengths

Soundness: The paper is technically rigorous, combining theoretical analysis with empirical validation. The authors use a two-layer neural network model to provide a formal proof of the "dichotomy" in distillation outcomes, showing how teacher confidence on unlearnable samples directly impacts student memorization of noise. This theoretical foundation is well-supported by experiments on both synthetic data and real-image classification tasks, ensuring that the claims are not merely speculative but grounded in observable phenomena.

Significance: The work addresses a critical and well-documented bottleneck in adversarial training: the inconsistent performance of Adversarial Distillation (AD). By identifying the "Robustly Unlearnable Set" (RUS) as the root cause of teacher dependency, the paper provides practitioners with a principled guideline for teacher selection—specifically using predictive entropy on the RUS as a metric. This has high potential to influence future research in robust model compression and defense mechanisms.

Originality: The paper offers a novel perspective by shifting the focus from the teacher's overall robustness to its specific behavior on "unlearnable" data subsets. The introduction of the RUS concept and the theoretical demonstration of how high-confidence supervision on these samples drives robust overfitting are creative contributions that deepen the field's understanding of the teacher-student dynamic in adversarial settings.

Presentation: The submission is clearly structured and follows a logical narrative flow, moving from the identification of a problem (teacher dependency) to a theoretical explanation, and finally to empirical proof and practical solutions. It effectively positions itself within existing literature by acknowledging the limitations of standard Adversarial Training (AT) and prior AD methods.

Weaknesses

Soundness: While the two-layer neural network provides a clear theoretical framework, there is an inherent gap between such simplified models and the complex, deep architectures used in practice. The paper could be strengthened by further discussing how these theoretical insights translate to modern architectures like Transformers or very deep ResNets where feature learning dynamics may differ.
Significance: The practical utility of the proposed metric (predictive entropy on the RUS) depends on the ease of identifying the Robustly Unlearnable Set in new, unseen datasets. If the computational cost of identifying this set is high, the "principled guideline" for teacher selection might be difficult to implement in large-scale industrial pipelines.

Originality: While the "Robustly Unlearnable Set" is a compelling framing, the idea that some samples are harder to learn than others (e.g., "easy" vs. "hard" samples) is a recurring theme in machine learning. The paper would benefit from a more explicit discussion on how the RUS differs fundamentally from existing concepts like "noisy labels" or "out-of-distribution" samples in a robust context.


Presentation: The theoretical section involves dense mathematical proofs (as seen in the Appendix snippets). While rigorous, the main text could benefit from more intuitive visualizations or high-level summaries of these proofs to ensure the core insights are accessible to a broader audience who may not be experts in learning theory.

---

> ### Author Rebuttal · Authors · 2026-03-30
>
> **1. Generalization to Modern Architectures**
>
> **A.** Our two-layer analysis is designed as a minimal model to isolate the representational-mismatch mechanism. While increasing the student's capacity shrinks the robustly unlearnable set, the critical question is whether it completely disappears. If a non-zero subset remains, our theory implies that the teacher's confidence on these samples will still induce the same failure mode.
>
> This is consistent with our empirical results. While scaling the student to a WRN-34-10 reduces the unlearnable intersection to 1,559 samples (Table 1), our additional experiment below confirms that the same dichotomy from our main text (Table 2) still holds: the teacher's predictive entropy on this persisting subset remains strongly associated with robust generalization.
>
> | Teacher (Entropy) | Clean | PGD-20 | AA | GenGap | Degrad. |
> | :--- | :--- | :--- | :--- | :--- | :--- |
> | Chen (1.88) | 85.35 | 59.52 | 55.56 | 17.34 | 0.28 |
> | Gowal (0.40) | 86.33 | 47.04 | 45.45 | 51.83 | 2.86 |
>
> Taken together, these results suggest that the mechanism persists in larger CNN students as long as a nonzero unlearnable subset remains, while its extension to Vision Transformers remains open.
>
> **2. Computational Overhead of Identifying the RUS**
>
> **A.** We agree that our practical proxy in Section 5.3 is not fully training-free, as it requires training a baseline student (e.g., via standard PGD-AT) to identify the unlearnable set. However, this baseline is typically required to compare AD results, and its cost is minimal since it avoids the computational overhead of using a large-capacity teacher. By reusing this baseline to rank candidate teachers based on their predictive entropy, we avoid repeated full adversarial distillation runs. Empirically, this low-cost proxy strongly predicts final student robustness across various AD methods and teachers (Table 2, Fig. 4(b), Fig. 7). Thus, while not a zero-cost procedure, we view it as a highly practical, amortized teacher-selection strategy.
>
> **3&4. Distinction from “Hard Samples” and Noisy Labels & Sensitivity to Epsilon Bounds**
>
> **A.** We thank the reviewer for this insightful suggestion. The proposed $\epsilon$-sensitivity analysis helps clarify both (i) how the robustly unlearnable set (RUS) differs from conventional “hard” or noisy samples (Q3), and (ii) how stable the entropy-based criterion remains across perturbation budgets (Q4).
>
>
> We first examine how the RUS varies with $\epsilon$ using a ResNet-18 student (Algorithm 1, Section C.3). As $\epsilon$ increases, more samples become inaccessible under adversarial constraints, leading to an expansion of the unlearnable set:
>
> | Epsilon | 4/255 | 6/255 | 8/255 | 10/255 |
> | :--- | :--- | :--- | :--- | :--- |
> | Unlearnable Intersection | 642 | 2,518 | 5,217 | 7,865 |
>
> This trend shows that reducing $\epsilon$ allows the same student to learn samples that were previously unlearnable. Combined with Table 1—where increasing model capacity at fixed $\epsilon$ yields a similar effect—this indicates that the RUS is not an intrinsic property of the data alone, but a relative phenomenon determined by the interaction between the data, the model’s representational capacity, and the adversarial constraint. In this sense, the RUS is distinct from conventional “hard” samples or noisy-label examples: it arises specifically from the inability to access robust features under adversarial perturbations.
>
>
> To address Q4, we further evaluate whether teacher entropy computed on a reference RUS (identified at $\epsilon = 8/255$) remains predictive when distillation is performed at different perturbation budgets. Concretely, we fix the reference set at $8/255$, compute each teacher’s predictive entropy on this set, and then run adversarial distillation at varying $\epsilon$:
>
> | Epsilon | Chen (1.906) | Rebuffi (1.582) | Bartoldson (0.872) | Gowal (0.198) |
> | :--- | :--- | :--- | :--- | :--- |
> | 4/255 | 72.03 | 71.13 | 66.70 | 65.47 |
> | 6/255 | 63.37 | 62.12 | 55.05 | 52.34 |
> | 8/255 | 55.47 | 53.48 | 44.39 | 41.85 |
> | 10/255| 48.11 | 44.46 | 37.27 | 35.31 |
>
> Although the size and composition of the RUS vary with $\epsilon$, the relative ranking of teachers remains consistent across all evaluated budgets. This suggests that predictive entropy measured on a reference RUS provides a stable and transferable signal for teacher selection, even when the training perturbation budget differs from that used to define the set.
>
>
> **5. Presentation of Theoretical Results**
>
> **A.** To improve accessibility to a broader audience, we will add an 'Overview' subsection (with existing Table 10) at the end of the problem setup in the appendix. This will provide a high-level summary of our theoretical implications, making the core insights easily accessible without delving into the rigorous proofs.

---

> > ### Author Rebuttal · Reviewer_tQMT · 2026-04-02
> >
> > Thank you for the detailed and thoughtful rebuttal. The additional experiments and clarifications—particularly on ε-sensitivity, the distinction between RUS and hard/noisy samples, and the stability of the entropy-based metric—are helpful and strengthen the paper. The WRN results also provide supporting evidence that the proposed mechanism persists beyond the minimal model. However, I still find that the theoretical analysis remains somewhat limited in its direct applicability to modern architectures (e.g., deeper networks or Transformers), and the practical pipeline, while reasonable, is not yet fully lightweight for large-scale use. Overall, these limitations slightly constrain the impact.

---

### Official Review · Reviewer_sCF3 · 2026-03-11

**Soundness:** 3
**Presentation:** 3
**Significance:** 3
**Originality:** 3
**Overall Recommendation:** 5
**Confidence:** 4

**Summary:**

This paper proposes an insight regarding the mystery of adversarial distillation that often times a more robust teacher can harm the robust accuracy of the student. The insight is centered around the concept of "robustly unlearnable set", referring to samples whose robust features are beyond the representation capability of the small student. The more robust teacher, however, can assign high confidence to those samples, causing the student having to memorize them through spurious correlations and therefore yielding worse generalization. Theoretical analysis together with comprehensive experiments all validate the proposed hypothesis/insight, effectively demystifying the paradox of adversarial distillation.

**Compliance With Llm Reviewing Policy:**

Affirmed.

**Final Justification:**

I maintain being positive about this submission and keep my original rating, especially given the other reviewers' concerns are well-addressed as well.

**Key Questions For Authors:**

I don't have questions.

**Limitations:**

yes

**Strengths And Weaknesses:**

I don't see any major weakness in this submission. The logic chain is smooth and solid, from motivation (revealing teacher-dependent dynamics of robust overfitting) to the demonstration and visualization of a set of model-dependent unlearnable samples, to theoretical analysis, and finally to the validation through systematic experiments. Figure 4(b) is also valuable for choosing the optimal teacher for adversarial distillation.

---

> ### Author Rebuttal · Authors · 2026-03-30
>
> Thank you for the positive and encouraging assessment. We are glad that the main logic chain of the paper was clear, from the empirical teacher-dependent dynamics to the robustly unlearnable set, theory, and the teacher-selection implication.

---

> > ### Author Rebuttal · Reviewer_sCF3 · 2026-04-02
> >
> > I don't have much concerns.

---

### Official Review · Reviewer_DisW · 2026-03-26

**Soundness:** 3
**Presentation:** 4
**Significance:** 3
**Originality:** 3
**Overall Recommendation:** 4
**Confidence:** 3

**Summary:**

This paper studies the phenomenon of adversarial distillation when robust teachers fail. The authors show that, under their construction, distilling from a “robust” teacher can lead to worse adversarial robustness than distilling from a “non-robust” teacher. The work aims to provide a theoretical explanation for empirical observations that stronger (robust) teachers may not always yield better students under adversarial distillation.

**Compliance With Llm Reviewing Policy:**

Affirmed.

**Final Justification:**

My questions have been addressed.  I raise to 4.

**Key Questions For Authors:**

See weaknesses above.

**Limitations:**

yes

**Strengths And Weaknesses:**

### Strengths

1. **Nontrivial insight into adversarial distillation.**
   The work highlights an interesting and somewhat counterintuitive phenomenon: teacher quality (in terms of robustness) does not monotonically translate to student robustness under distillation, and attribute this to the fact that there are learnable and unlearnable samples in the dataset. This contributes to the broader understanding of when distillation can fail.


2. **Clear and tractable theoretical framework.**
   The paper proposes a clean, analytically tractable setup that isolates the interaction between teacher signals, adversarial training, and student representation constraints.


---

### Weaknesses

My main concern is the theoretical constructions, which I will detail below.

 The student is explicitly constrained so that its weights remain orthogonal to the feature direction \(v\) throughout training. This effectively hard-codes the inability to use the “unlearnable” feature, rather than deriving it from capacity, optimization, or data limitations. As a result, the main phenomenon is driven by a specifically constructed **projection constraint built into the model** for theory, which is quite different from the standard neural network training dynamics in practice.

2.  The adversarial training objective restricts perturbations to lie in the span of the signal patch, instead of the general l_infty or l_2 ball. However, the resulting conclusions are interpreted as failures of adversarial robustness more broadly (including vulnerability due to noise memorization). This creates a conceptual gap: the theorem analyzes a **restricted inner maximization**, while the conclusions are phrased in terms of general robustness.

3.   In the proof of the key robustness lower bound (showing \(L^{rob}_{S_L} \ge 1/2 - o(1)\)), the final step (on page 45) relies on ruling out an event involving a negative bound on a sum of squares, and then directly concludes the desired inequality. I don’t know how equation 165 is derived from equation 164. More explanation will be helpful.

4.    The paper’s conclusions rely heavily on assumptions hidden in Appendix D. The main results are said to hold under conditions “specified in (24),” plus Assumption D.4 and the induction hypothesis D.6, making the required assumptions hard to parse. In addition, combining these pieces, it seems the authors implicitly assume $\sigma_n \sqrt d \ll \alpha$, and $\sigma_0\sigma_n\sqrt d \ll 1$. These are too strong. More justification of these assumptions will be helpful.

---

> ### Author Rebuttal · Authors · 2026-03-30
>
> **1. On whether the orthogonality constraint hard-codes unlearnability.**
>
> **A.** Thank you for this important point. We agree that, at the theorem level, the student’s inability to use the feature direction is imposed through an explicit orthogonality constraint. Our intent, however, is not to claim that standard deep networks literally evolve under such a constraint, but to use it as a minimal abstraction of the sample-level representational mismatch suggested by Section 3: the same training sample can be robustly learnable for a larger model yet remain unlearnable for a smaller student. In this sense, the projection constraint is used to isolate the mechanism cleanly, not to exactly model practical training dynamics. Our theoretical claim is therefore that once such a mismatch exists, confident teacher supervision on student-inaccessible robust features can drive spurious fitting and robust overfitting. Section 5 complements this idealized analysis by showing that a corresponding teacher-dependent failure mode is also observed on real-image benchmarks under standard PGD-based training.
>
> **2. On the restricted inner maximization.**
>
> **A.** We agree that restricting the inner maximization to the signal subspace introduces a gap relative to fully general robustness, and we do not claim that our theorem provides a complete characterization of unrestricted adversarial training or distillation dynamics. Rather, our goal is to establish a mechanism-level sufficiency result. Specifically, we show that once confident supervision is imposed on unlearnable samples—either through hard labels or a Bad Teacher—it drives the student to memorize spurious noise directions, which in turn degrades robust test performance and induces adversarial vulnerability. In this sense, noise memorization is no longer benign for robustness.
>
> The structured perturbation set serves as an analytically tractable surrogate that allows us to rigorously isolate and prove this mechanism, rather than suggesting that the phenomenon depends on this restriction itself. This modeling choice is also aligned with prior theoretical works (Li, B. and Li, Y., ICML 2025), which similarly constrain perturbations to structured feature subspaces. Importantly, in Section 5 we demonstrate that the same teacher-dependent failure mode arises under standard PGD-based training on real-image benchmarks, confirming that the core mechanism identified under our restricted inner maximization also manifests in practice under more general AT training.
>
>
>
>
> **3. Derivation of Eq. 165**
>
> **A.** Eq. 165 is obtained by combining the decomposition before Eq. 162 with the two subsequent bounds. Eq. 162 shows that the first probability term is 1/2 by symmetry of the Gaussian data distribution. Eq. 164 shows that the second probability term is zero, since it would require a sum of squares to be strictly negative, which is impossible. Plugging these two facts into the preceding decomposition gives the stated lower bound. We will add an intermediate step to clarify Eq. 165.
>
> **4. Clarification on Assumptions and Hyperparameter Scaling**
>
> **A.** We clarify that the main results hold on the parameter conditions in (24), formalized in Assumption D.4. These are not unique choices; rather, any parameters satisfying the more explicit scaling conditions summarized in Table 10 are sufficient for our theoretical guarantees. Importantly, Induction Hypothesis D.6 is not an additional assumption, but an intermediate statement that is rigorously established in Lemma F.6.
>
> We now explain the role of the key scalings. Our analysis characterizes the learning dynamics of robust feature learning versus noise memorization. The condition $\sigma_n \sqrt{d} \ll \alpha$ ensures that the robust feature is learned significantly faster than noise, and can be interpreted as a per-sample signal-to-noise ratio condition. While this can be relaxed by considering aggregated signal strength (e.g., $N(1-p_{\mathrm{un}})\alpha$), controlling $N$ is also necessary to avoid cross-term effects that complicate the analysis of noise dynamics.
>
> The condition $\sigma_0 \sigma_n \sqrt{d} \ll 1$ ensures that unmemorized noise at initialization scale is sufficiently small, so that its contribution under test-time perturbations are negligible compared to that of the robust signal. Under these conditions, we show that noise memorization is suppressed unless there is confident supervision on unlearnable samples (e.g., from hard labels or a Bad Teacher). We will revise the manuscript to make these scaling assumptions more explicit and to clarify the range of regimes under which our results hold.

---

> > ### Author Rebuttal · Reviewer_DisW · 2026-04-02
> >
> > Thanks for your rebuttal. My questions have been addressed. Hope you can add these discussions to the revision.

---

### Decision · Program_Chairs · 2026-04-30

**Decision:**

Accept (regular)

**Comment:**

This paper provides a compelling and well-supported explanation for a widely observed phenomenon in adversarial distillation. While the theoretical model is simplified and relies on strong assumptions, it successfully isolates a key mechanism: the interaction between teacher confidence and student representational limits on unlearnable samples.